# Surface hole polaron site tuning governs charge carrier separation in BiVO$_4$ photoanodes

Houjiang Liu[1], Hongwei Cong[1], Guijun Yang[1], Chuangchuag Gong[1], Jiawei Ding[1], Yuan yuan Fu[1], Jin Cui[1], Kai Song[1] ✉, Biao Chen [1,2], Chunnian He [1,2], Naiqin Zhao[1], Jinhua Ye[3] & Fang He [1] ✉

The self-trapping of charge carriers, resulting in the formation of polarons, significantly restricts the separation and transport of charge carriers in photoelectrochemical systems. Herein, using bismuth vanadate as a model photoanode, we propose a surface-selective strategy to regulate hole polarons. Density functional theory calculations predict that substituting bismuth ions with indium ions suppresses hole polaron formation by weakening electron-phonon coupling. This substitution is achieved through a liquid-phase cation exchange method, enabling precise surface modification. The electron paramagnetic resonance, temperature-dependent photoluminescence spectroscopy, in situ irradiation X-ray photoelectron spectroscopy, and femtosecond time-resolved absorption spectroscopy all confirm the suppression of hole polaron formation. After loading co-catalyst, the optimized photoanode achieves a water-splitting photocurrent density of 6.46 mA cm$^{-2}$ at 1.23 V versus the reversible hydrogen electrode, with an applied bias photo-to-current efficiency of 2.19%. The unbiased tandem system exhibits a solar-to-hydrogen conversion efficiency of 6%. Here, we show that suppressing surface hole polaron formation facilitates hole carrier release, offering a pathway for enhancing photoelectrochemical performance.

Photoelectrochemical (PEC) water splitting represents a promising approach for converting intermittent solar energy into storable hydrogen fuel, thereby simultaneously addressing energy sustainability and environmental challenges[1,2]. Metal oxides such as BiVO$_4$[3–5], TiO$_2$[6–8], and Fe$_2$O$_3$[9,10] have emerged as promising photoanode materials due to their favorable optoelectronic properties. However, their practical implementation remains constrained by the low solar-to-hydrogen (STH) conversion efficiencies[11,12]. Effective charge carrier (electron-hole pair) separation and injection are critical in PEC systems, as these processes, coupled with efficient light harvesting, determine the population of carriers available for water oxidation[13]. Substantial surface recombination significantly reduces carrier utilization efficiency, rendering surface charge extraction a critical challenge.

Recent studies have highlighted the role of polarons in photocatalysis and PEC systems[14,15]. A small polaron is a quasiparticle formed through the self-trapping of excess charge via electron-phonon coupling[16,17]. The source of this additional charge is attributed to photogenerated carriers (electron-hole pairs), which can lead to the self-trapping of carriers and the formation of electron polaron (EP) and hole polaron (HP) in many

[1]School of Materials Science and Engineering, Tianjin Key Laboratory of Composite and Functional Materials, Key Laboratory of Advanced Ceramics and Machining Technology (Ministry of Education), Tianjin University, Tianjin, P. R. China. [2]School of Materials Science and Engineering, Advanced Catalytic Materials Research Center, Tianjin University, Tianjin, P. R. China. [3]National Industry-Education Platform of Energy Storage, Tianjin University, Tianjin, P. R. China. ✉e-mail: songk@tju.edu.cn; fanghe@tju.edu.cn

metal oxide photoanodes[14,18]. Their low mobility induces severe recombination, degrading photocatalytic activity. While strategies such as element doping[19,20] and oxygen vacancy engineering[21] enhance EPs hopping, the regulation of surface HPs remains unresolved.

Herein, using BiVO$_4$-which intrinsically hosts both EP (VO$_4$) and HP (BiO$_8$) sites-as a model system, we demonstrate electronic structure modulation of surface HP sites through in-situ selective cation exchange. By selectively suppressing the formation of HP sites, we aim to facilitate the release of hole carriers, thereby enhancing charge carrier separation efficiency. Density functional theory (DFT) calculations predict that isoelectronic substitution of In$^{3+}$ for Bi$^{3+}$ suppresses HPs formation by weakening electron-phonon coupling. This reduces the extent of charge localization under photoexcitation and increases the formation energy of HPs. A surface-selective liquid-phase cation exchange (LPCE) method is employed to achieve precise control over the selective substitution of Bi$^{3+}$ with In$^{3+}$. The suppression of HP formation has been experimentally verified using electron paramagnetic resonance (EPR), temperature-dependent photoluminescence (Td-PL) spectra, in situ irradiation XPS (ISI-XPS), and femtosecond transient absorption spectroscopy (fs-TAS). By effectively suppressing HPs formation and thereby releasing surface hole carriers, the optimized In/BiVO$_4$/FeOOH photoanode exhibits a water-splitting photocurrent density of 6.46 mA cm$^{-2}$ at 1.23 V$_{RHE}$ and an applied bias photo-to-current efficiency (ABPE) of 2.19% using a simple co-catalyst. The unbiased tandem device system constructed using In/BiVO$_4$/FeOOH photoanode achieves a solar-to-hydrogen (STH) conversion efficiency of 6%. Given that HPs typically hinder the performance of common metal oxide photoanodes, our strategy provides valuable insights into polaron suppression approaches that can be applied to a wide range of PEC processes to improve carrier release.

## Results

### Theoretical design for HPs suppression

During the transport of photogenerated carriers in metal oxides, carrier self-trapping inevitably occurs due to electron-phonon coupling induced by excess charge, leading to the formation of small polarons[22]. This polaronic state corresponds to a relatively low-energy configuration that confines carrier transport to thermally activated hopping mechanisms, resulting in intrinsically slow migration kinetics. Crucially, polaron-mediated recombination, especially that involving HPs, significantly compromises photoanode performance because surface oxidation reactions strongly depend on efficient hole utilization. To address this fundamental limitation, we employ BiVO$_4$ as a model system and propose a refined approach for the release of surface hole carriers. Theoretical and experimental evidence reveals that substituting Bi$^{3+}$ with higher electronegative In$^{3+}$ weakens the electron-phonon coupling, thereby inhibiting the formation of HPs (Fig. 1)[23]. Through precisely controlled LPCE method, In$^{3+}$ is selectively incorporated into surface Bi sites while preserving the bulk BiVO$_4$ integrity of the lattice (Fig.1). Both theoretical and experimental investigations reveal that the selective tuning of surface HP sites effectively weakens electron-phonon coupling, increases the formation energy of HPs, thereby facilitating the release of additional hole carriers.

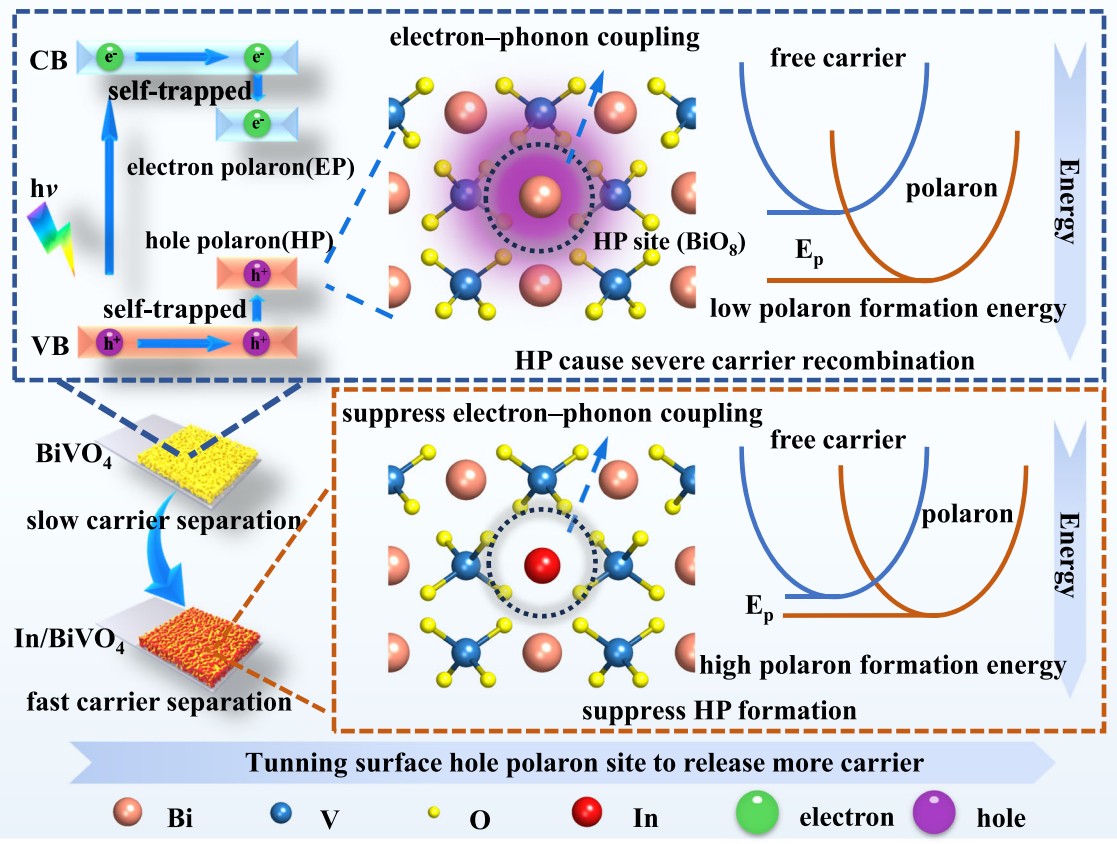

**Fig. 1 | Schematic diagram of the HPs suppression mechanism.** Figure 1 illustrates the formation and suppression mechanism of HPs. The orange, blue, yellow, and red spheres represent Bi, V, O, and In atoms, respectively. The green and purple spheres denote electrons and holes, respectively. In the BiVO$_4$ photoanode, the self-trapping of photogenerated carriers leads to the formation of EPs (VO$_4$ sites) and HPs (BiO$_8$ sites), with HPs significantly limiting the utilization of surface hole carriers (highlighted in the blue dashed box). By selectively substituting Bi$^{3+}$ with In$^{3+}$, the formation energy of polarons is increased, thereby suppressing the generation of HPs and releasing more hole carriers (highlighted in the orange dashed box).

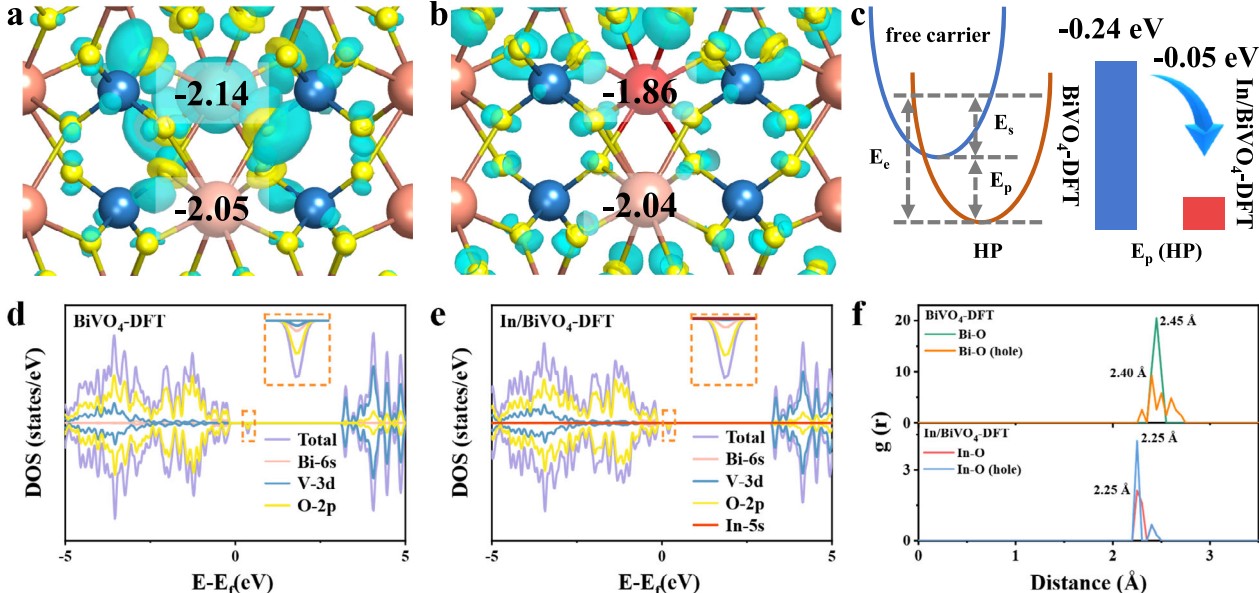

**Fig. 2 | DFT calculations analyze the suppressive effect of In substitution.** Charge density difference diagram of **a** BiVO₄-DFT and **b** In/BiVO₄-DFT with HP calculated by DFT. **c** The calculated polaronic stability of BiVO₄-DFT and In/BiVO₄-DFT, where $E_p$, $E_s$ and $E_e$ represent the contributions of polaron formation energy, lattice distortion energy, and electronic energy, respectively. PDOS of **d** BiVO₄-DFT and **e** In/BiVO₄-DFT calculated by DFT with HP (The dashed box shows an enlarged view of the curve). **f** Bi-O radial distribution functions in BiVO₄-DFT and In-O radial distribution functions in In/BiVO₄-DFT with and without HP. The orange, blue, yellow, and red spheres represent Bi, V, O, and In atoms, respectively.

## DFT-guided Strategies for the Suppression of HPs

To investigate the impact of surface polaron site regulation, we performed density functional theory (DFT) simulations to model the local electronic states associated with polaron formation under photoexcitation conditions. The BiVO₄ model and the In-substituted BiVO₄ model are denoted as BiVO₄-DFT and In/BiVO₄-DFT, respectively (Supplementary Figs. 1, 2). Differential charge density and corresponding Bader charge analysis reveal that, after substituting Bi³⁺ with In³⁺, the charge localization at the formed InO₈ site is significantly suppressed compared to that at the BiO₈ site (Fig. 2a, b). This suppression of charge localization is conducive to inhibiting electron-phonon coupling, thereby effectively suppressing the formation of HPs. We evaluated the ease of polaron formation using the polaron formation energy ($E_p$), defined as the energy difference between a supercell with frozen atomic positions in the polaronic state (without additional electrons) and a perfect bulk supercell, representing the energy cost required to accommodate a small polaron[24]. In contrast, the $E_p$(HP) of In/BiVO₄-DFT is −0.05 eV, significantly higher than that of BiVO₄-DFT (−0.24 eV), providing strong evidence for the suppressive effect of In substitution on polaron formation (Fig. 2c and Supplementary Table 1). Subsequently, we analyzed the electronic states using the Projected density of states (PDOS) method (Fig. 2d, e). The results indicate that in BiVO₄-DFT model, excess holes occupy the O and Bi sites, forming localized HPs. In contrast, in In/BiVO₄-DFT, there is no hole localization at the In sites. Instead, excess holes tend to accumulate at other Bi and O sites, suggesting that In sites are less susceptible to polaron formation. To further confirm the suppression of polaron formation, we compared the bond lengths of the In/BiVO₄-DFT and BiVO₄-DFT models before and after photoexcitation (Fig. 2f and Supplementary Fig. 3). It was observed that the Bi-O bond length significantly decreases from 2.45 Å to 2.40 Å, whereas the In-O bond length remains at 2.25 Å, further supporting the notion that polarons are less likely to form at InO₈ sites[25]. Correspondingly, we also investigated the effect of In substitution on EP behavior. Differential charge density, Bader charge analysis, $E_p$(EP), DOS, and bond length variation data collectively indicate that In substitution has negligible impact on

the properties of Eps (Supplementary Figs. 2, 3 and Supplementary Table 1).

Additionally, we characterized the properties of the BiVO₄-DFT and In/BiVO₄-DFT models under non-photoexcited conditions (Supplementary Figs. 4, 5 and Supplementary Tables 2, 3). The band structure and PDOS analyses indicate that their band gaps are 2.31 eV and 1.69 eV, respectively, which can be attributed to the contribution of the introduced In³⁺ to the conduction band minimum (Supplementary Fig. 4). Moreover, we calculated the effective carrier masses for both systems. BiVO₄-DFT has minimum effective masses of 0.864 for electrons and 1.144 for holes, while In/BiVO₄-DFT shows minimum effective masses of 2.013 for electrons and 0.712 for holes, respectively. A lower effective mass of charge carriers indicates a faster migration rate at the interface, which is particularly advantageous for photoanode[26]. Furthermore, the difference in effective masses between electrons and holes reflects the likelihood of carrier recombination; a larger disparity in effective mass suggests a reduced probability of recombination[26,27]. The effective mass ratio of 2.83 for the In/BiVO₄-DFT model indicates a lower tendency for carrier recombination. Although DFT simulations have inherent limitations in fully replicating the complexity of real catalytic environments, they nonetheless offer critical insights into the underlying electronic structure modifications.

## Synthesis and structural characterization

To investigate the role of In³⁺ in suppressing HP formation and enhancing carrier separation, surface In-substituted BiVO₄ photoanodes were synthesized via the LPCE method, labeled as InX/BiVO₄ (X = 1, 2, 5, 10) based on the substitution time (Supplementary Figs. 6, 7). PEC characterization revealed that the optimal performance was achieved at X = 2 h (designated as In/BiVO₄ hereafter, PEC Water Oxidation Performance section). A comprehensive characterization and analysis of BiVO₄ and In/BiVO₄ photoanodes were performed, including X-ray diffraction (XRD), Raman spectroscopy, scanning electron microscopy (SEM), transmission electron microscopy (TEM), and other analytical techniques (Supplementary Figs. 8–10). To accurately

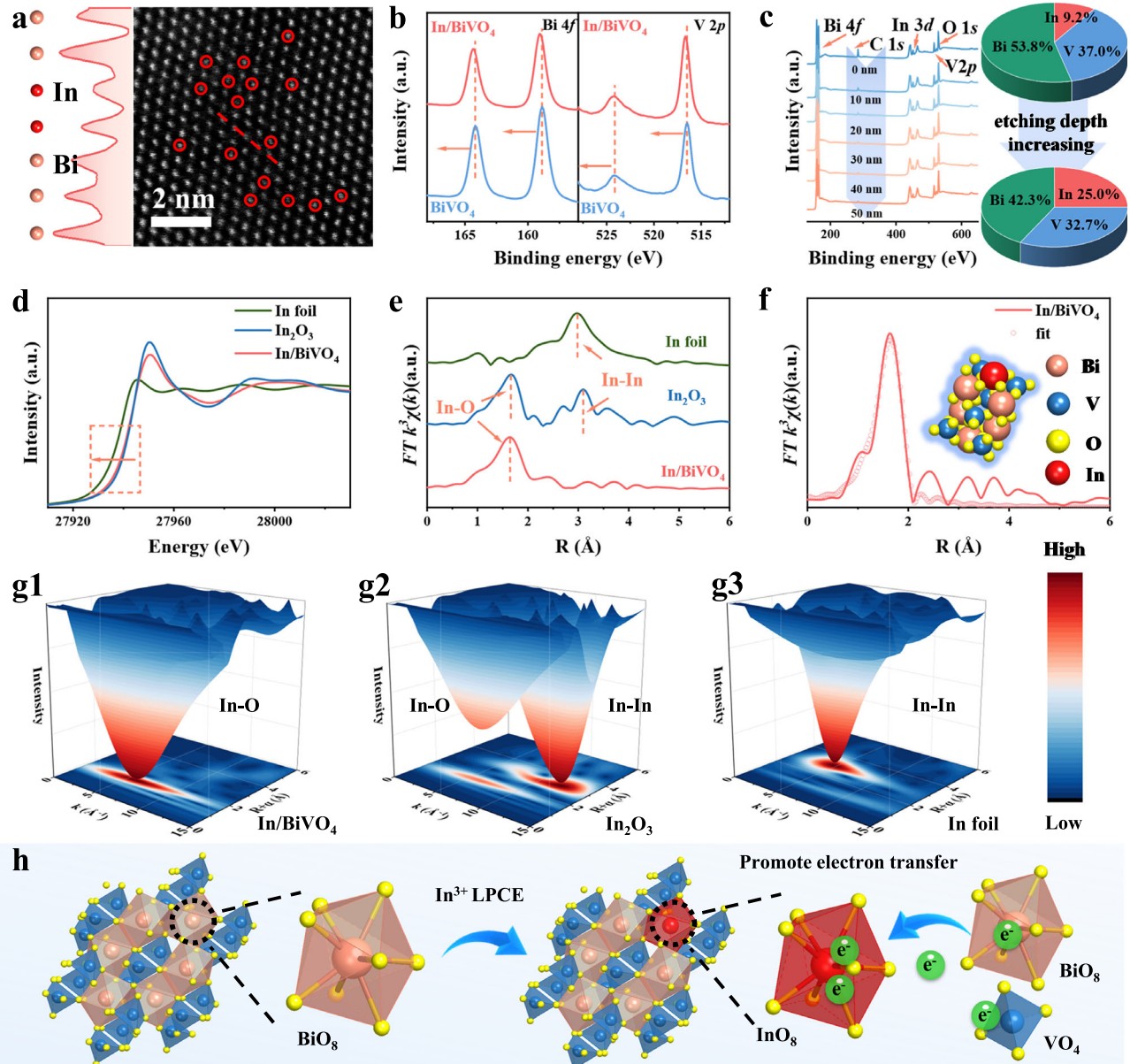

**Fig. 3 | Material characterization. a** AC HAADF-STEM images of In/BiVO$_4$ photo-anode (the red circles mark the dark spots, and the intensity profile extracted along the red dashed line is shown on the left). **b** XPS high-resolution spectroscopy of Bi 4$f$ and V 2$p$ in BiVO$_4$ and In/BiVO$_4$ photoanodes. **c** Etch-XPS spectroscopy of In/BiVO$_4$ photoanode (pie charts display the changes in elemental composition before and after etching). **d** In K-edge XANES spectroscopy of In foil, In$_2$O$_3$ and In/BiVO$_4$. **e** Fourier transformed k$^3$-weighted $\chi$(k) function of the EXAFS spectroscopy for In K-edge. **f** corresponding EXAFS fitting curve for In/BiVO$_4$. **g** WT-EXANES of In foil, In$_2$O$_3$ and In/BiVO$_4$. **h** LPCE process and the carrier transfer of different sites diagram (the dashed box shows the enlarged views of the BiO$_8$ and InO$_8$ sites). The orange, blue, yellow, and red spheres represent Bi, V, O, and In atoms, respectively.

determine the presence of In$^{3+}$, Spherical aberration corrected high-angle annular dark-field scanning transmission electron microscopy (AC HAADF-STEM) was employed. The isolated dark dots in Fig. 3a indicate the atomic dispersion of In atoms, rather than the formation of clusters or nanoparticles. Supplementary Fig. 11 presents an image of the (110) crystal plane and a schematic of atomic structure. These results indicate that surface In$^{3+}$ is incorporated into the BiVO$_4$ lattice by substituting Bi$^{3+}$, while the overall lattice structure remains intact, confirming the successful achievement of selective and uniform substitution through the LPCE method.

X-ray photoelectron spectroscopy (XPS) was employed to investigate the elemental states and chemical composition of the surface region. Quantitative elemental analyses via EDS and XPS revealed differences in the elemental composition between the bulk and surface

regions, indicating that LPCE initially occurs at the solid-liquid interface before extending inward (Supplementary Figs. 12, 13 and Supplementary Tables 4, 5). Furthermore, the significant reduction in surface Bi content suggests that In preferentially substitutes Bi rather than V. The high-resolution Bi 4$f$ spectrum exhibited symmetric peaks at 164.34 eV and 159.04 eV, corresponding to Bi 4$f_{5/2}$ and Bi 4$f_{7/2}$ of Bi$^{3+}$, respectively (Fig. 3b). Similarly, the high-resolution V 2$p$ XPS spectrum revealed two symmetric peaks at 524.33 eV and 516.63 eV, corresponding to V 2$p_{1/2}$ and V 2$p_{3/2}$ of V$^{5+}$ (Fig. 3b)[28]. Notably, the binding energies of the Bi 4$f$ and V 2$p$ peaks in In/BiVO$_4$ are shifted to higher values compared to pristine BiVO$_4$, suggesting an electron transfer from Bi and V sites to In sites after LPCE. Moreover, the potential effects induced by local strain and oxygen vacancies were systematically discussed and excluded (Supplementary Fig. 14). The In 3$d_{3/2}$

and In $3d_{5/2}$ peaks in the In $3d$ spectroscopy confirm the presence of $In^{3+}$, indicating that the valence state of In in the samples is consistent with that of the In source ($In(NO_3)_3$), which remained constant throughout LPCE process (Supplementary Fig. 13b)[29]. Further analysis of elemental composition changes and electron transfer was conducted using etching combined with XPS (etch-XPS). The results indicate that as the etching depth increases, the In content decreases while the Bi content significantly increases, with only a moderate increase in V content, suggesting that In primarily substitutes Bi sites (Fig. 3c and Supplementary Table 7). Additionally, the binding energy shifts further confirm the electron transfer from Bi and V sites to In sites (Supplementary Fig. 13d–f).

To further elucidate the electronic state and local coordination environment of In, X-ray absorption fine (XAFS) spectroscopy was employed. As shown in Fig. 3d, the K-edge X-ray absorption near-edge structure (XANES) spectroscopy of In/BiVO$_4$ closely resembles that of In$_2$O$_3$, implying that the oxidation state of In in In/BiVO$_4$ is predominantly trivalent. Notably, the absorption edge of In/BiVO$_4$ shifts toward lower binding energies compared to that of In$_2$O$_3$, consistent with the XPS results. This shift indicates a decrease in the oxidation state of In species, attributed to the higher electronegativity of In compared to Bi, resulting in electron transfer to the InO$_8$ sites formed via In substitution. Extended X-ray absorption fine structure (EXAFS) analysis was carried out to elucidate the local coordination environments of Bi and In species (Supplementary Figs. 15, 16). The local coordination environment of Bi species in BiVO$_4$ was carefully analyzed using the Bi L-edge X-ray absorption spectrum. The results revealed that in BiVO$_4$, the Bi species predominantly exhibit Bi-O coordination in the first coordination shell. By fitting the EXAFS data, a coordination number of approximately 7.4 for Bi-O was obtained, indicating the presence of BiO$_8$ sites (Supplementary Table 8)[30–32]. As shown in Fig. 3e, the R-space transformation of the EXAFS spectrum reveals a pronounced peak at 1.62 Å, corresponding to In-O bonding. The fitted EXAFS results show that in In/BiVO$_4$, the In-O bond length in the first coordination shell is 2.159 Å, with a coordination number of approximately 7.72, which is significantly different from that in In$_2$O$_3$ (6). The very similar coordination numbers between the Bi species in BiVO$_4$ and the In species in In/BiVO$_4$ suggest the presence of InO$_8$ sites in In/BiVO$_4$, analogous to BiO$_8$. This observation provides strong structural evidence in support of the selective replacement of Bi sites via the LPCE strategy. Additionally, wavelet transform (WT) simulations were carried out to analyze the radial distance resolution in K-space. As shown in Fig. 3g1-g3, the WT intensity maxima corresponding to In-O coordination near 5.0 Å$^{-1}$ are well-resolved between 1.0 and 2.0 Å, with no significant In-In coordination observed, indicating that In is predominantly coordinated with O atoms in the sample.

The ICP results also support the conclusion of the selective substitution of In for Bi (Supplementary Table 9). Moreover, we conducted DFT calculations to reveal the underlying mechanism of the selective substitution (Supplementary Fig. 17 and Supplementary Table 10). The significantly lower formation energy for In substituting the Bi site (0.03 eV) compared to that for substituting the V site (2.54 eV) indicates that the substitution of In for Bi is thermodynamically favorable, enabling selective substitution. Furthermore, DFT calculations elucidated the electron transfer behavior induced by the In substitution (Supplementary Fig. 5 and Supplementary Table 3). The differential charge density and Bader charge analysis on the (110) crystal plane reveal electron accumulation at the In sites, and these results are in good agreement with the XPS and synchrotron radiation results. Collectively, the analysis demonstrates that the InO$_8$ sites are formed after LPCE, which facilitates electron transfer from Bi and V sites to In sites. This selective regulation effectively modulates the electronic structure of specific sites and provides an accurate experimental framework

consistent with theoretical predictions, thereby elucidating the role of HP suppression in enhancing PEC performance.

## PEC water oxidation performance

The PEC performance of photoanodes was evaluated in a 0.2 M potassium borate electrolyte (KBi) using a standard three-electrode system under AM 1.5 G illumination (100 mW cm$^{-2}$). The optimal In2/BiVO$_4$ photoanode exhibited a photocurrent density of 3.19 mA cm$^{-2}$ (1.23 V$_{RHE}$) when the LPCE treatment time was 2 h, representing a 2.9-fold increase compared to pristine BiVO$_4$ (Fig. 4a and Supplementary Fig. 18). Additionally, the optimal In/BiVO$_4$/FeOOH photoanode demonstrated a photocurrent density of 6.46 mA cm$^{-2}$ at 1.23 V$_{RHE}$ upon the introduction of FeOOH as a co-catalyst (Fig. 4b and Supplementary Figs. 19–22). It is noteworthy that a comparison was made with the photocurrent density of the BiVO$_4$ photoanode with FeOOH (Supplementary Fig. 21b). The results demonstrate that LPCE treatment significantly enhanced the photocurrent density of both pristine BiVO$_4$ and BiVO$_4$ with FeOOH, indicating that surface In substitution plays a critical role in enhancing PEC activity. Notably, the Fe-substituted photoanode prepared using similar LPCE method also exhibits a significantly enhanced photocurrent density, indicating that our LPCE-regulated HP strategy is universally applicable (Supplementary Fig. 23).

To exclude the contribution of electrical energy and accurately evaluate the optical conversion efficiency, the ABPE was calculated. As shown in Fig. 4c, the maximum ABPE of In/BiVO$_4$ reached 0.84% at 0.75 V$_{RHE}$, which is 3.65-fold relative to pristine BiVO$_4$ (0.23%). The In/BiVO$_4$/FeOOH system achieved a maximum ABPE of 2.19% at 0.67 V$_{RHE}$. Furthermore, to investigate the effect of varying light wavelengths on photocurrent density, the incident photon-to-current conversion efficiency (IPCE) was measured at 1.23 V$_{RHE}$ (Fig. 4d and Supplementary Fig. 24). The absorption range of all photoanodes spanned approximately 350–520 nm, consistent with the light absorption profile of BiVO$_4$[33]. However, the IPCE values of In/BiVO$_4$ photoanode showed a significant increase, reaching a maximum of 48.7% at 380 nm. Furthermore, after the co-catalyst was introduced, the IPCE increased to 79.3% at 420 nm. Importantly, the IPCE response range remained largely unchanged, aligning with the optical bandgap assessment results (Supplementary Fig. 25), indicating that the observed enhancement in PEC performance primarily stems from improved charge carrier separation.

To evaluate carrier separation ($\eta_{separation}$) and injection ($\eta_{injection}$) efficiencies, photocurrent measurements were conducted in an electrolyte containing a sacrificial agent (0.2 M Na$_2$SO$_3$). The In/BiVO$_4$ photoanode demonstrated significant improvements in both $\eta_{separation}$ (65%) and $\eta_{injection}$ (73%) at 1.23 V$_{RHE}$, corresponding to 1.3-fold and 2.1-fold increases over pristine BiVO$_4$ (50% and 35%, respectively) (Fig. 4e, 4f). After the co-catalyst was introduced, the $\eta_{separation}$ and $\eta_{injection}$ of In/BiVO$_4$/FeOOH were further enhanced to 97% and 99%, attributed to enhanced surface oxidation capability.

To accurately evaluate the water-splitting efficiency of the In/BiVO$_4$/FeOOH photoanode, the Faradaic efficiencies for H$_2$ and O$_2$ production were calculated, yielding approximately 93%, indicating that the majority of photogenerated carriers are effectively utilized for water splitting (Supplementary Fig. 21c). Furthermore, as shown in Fig. 4g, the pristine BiVO$_4$ photoanode exhibits poor stability, with a significant decline in photocurrent within just 1 h. In contrast, In/BiVO$_4$ photoanode exhibited a moderate improvement in stability. Remarkably, the In/BiVO$_4$/FeOOH photoanode achieved prolonged stability exceeding 24 h, attributable to the protective effect on the surface effect of the co-catalyst (Supplementary Fig. 26). In particular, the PEC performance of the In/BiVO$_4$/FeOOH and In/BiVO$_4$ photoanodes in weak alkaline electrolytes is competitive with that of reported BiVO$_4$-based photoanodes (Supplementary Tables 11 and 12), underscoring

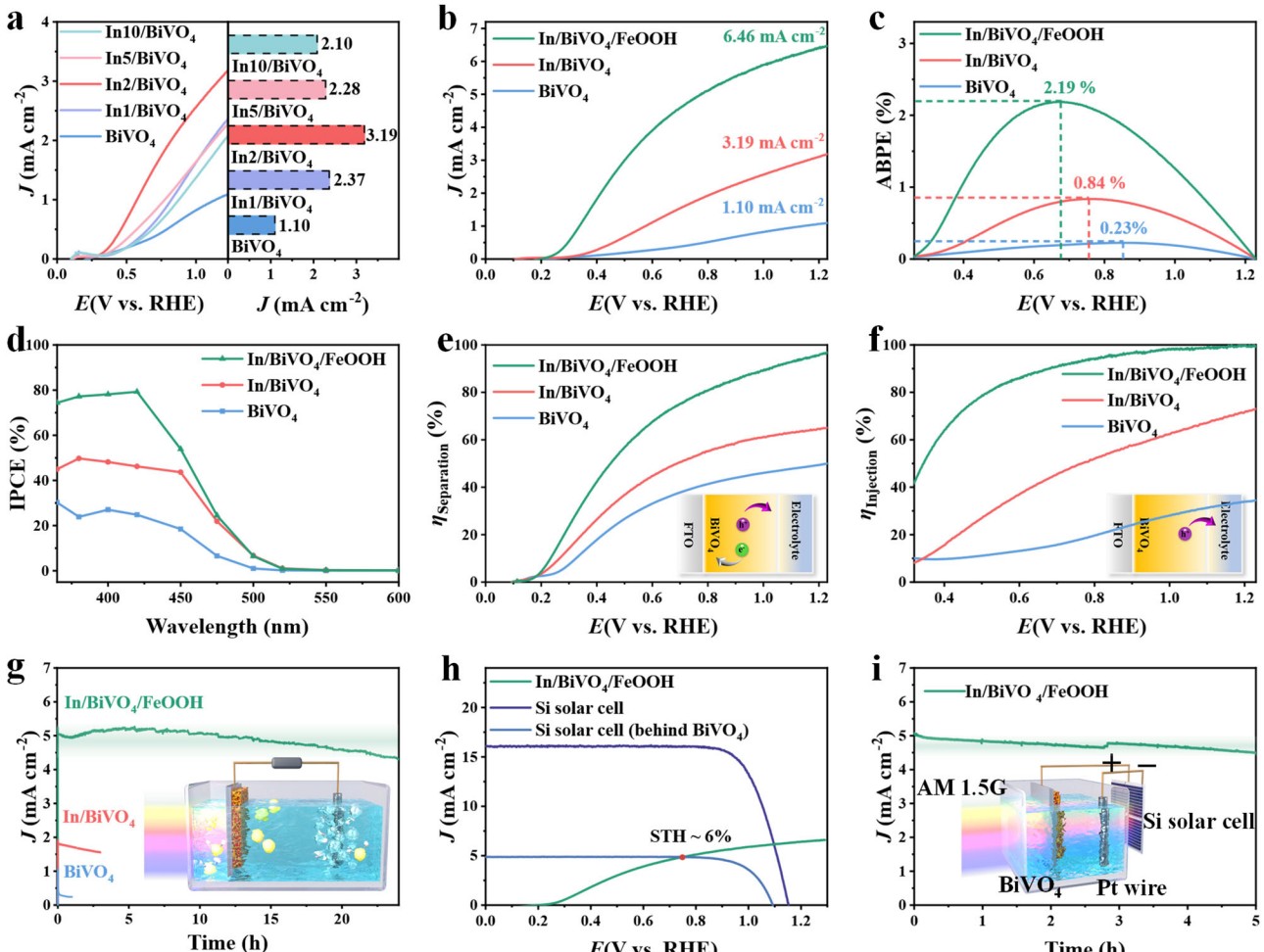

**Fig. 4 | PEC performance for water oxidation. a** LSV curves of $BiVO_4$ and $InX/BiVO_4$ ($X = 1, 2, 5, 10$) photoanodes in a 0.2 M KBi solution (pH = 8.4) under AM 1.5 G illumination (100 mW cm$^{-2}$). **b** LSV curves and chopped LSV curves, **c** ABPE and **d** IPCE of $BiVO_4$, $In/BiVO_4$ and $In/BiVO_4/FeOOH$ photoanodes in a 0.2 M KBi solution (pH = 8.4) under AM 1.5 G illumination (100 mW cm$^{-2}$). **e** Calculated charge separation efficiency of $BiVO_4$, $In/BiVO_4$ and $In/BiVO_4/FeOOH$ photoanodes (inset schematically illustrates the separation process of photogenerated electron-hole pairs). **f** Calculated charge injection efficiency of $BiVO_4$, $In/BiVO_4$ and $In/BiVO_4/FeOOH$ photoanodes (schematically depicts the hole injection process at the

electrode/electrolyte interface). **g** Photocurrent density versus time curves of $BiVO_4$, $In/BiVO_4$ and $In/BiVO_4/FeOOH$ photoanodes at 0.8 $V_{RHE}$ in a 0.2 M KBi solution (pH = 8.4) under AM 1.5 G illumination (inset shows the schematic of the PEC testing setup). **h** $J-V$ curves of the $In/BiVO_4/FeOOH$ photoanode and Si solar cell under AM 1.5 G illumination (100 mW cm$^{-2}$). **i** Photocurrent density versus time curves of the $In/BiVO_4/FeOOH$-Si solar cell tandem device under AM 1.5 G simulated solar radiation (100 mW cm$^{-2}$) (inset shows the schematic of the $In/BiVO_4/FeOOH$-Si solar cell tandem device). The green and purple spheres denote electrons and holes, respectively. The data are presented without iR correction.

the substantial enhancement in PEC performance attributed to surface In substitution.

Concurrently, a tandem device was developed for bias-free solar water splitting by integrating a commercial Si solar cell. The performance metrics of the Si solar cell are presented in Fig. 4h, where the intersection of the $J-V$ curves for the Si solar cell and $In/BiVO_4/FeOOH$ photoanode occurs at 4.85 mA cm$^{-2}$, yielding a calculated STH conversion efficiency of approximately 6%. Furthermore, the $In/BiVO_4/FeOOH$-Si solar cell tandem device demonstrated exceptional operational stability, sustaining unassisted water oxidation for over 5 h under unbiased conditions (Fig. 4i).

## Suppression of HPs and carrier property analysis

To elucidate the relationship between the enhancement of PEC activity and the suppression of HPs, EPR spectroscopy was performed. Under dark conditions, both samples exhibited an EPR signal at g = 2.003, which is attributed to free electrons (Supplementary Fig. 27a)[34]. Under illumination, additional EPR signals were observed in the $BiVO_4$ sample. Specifically, the peaks at g = 1.945 and g = 1.960 are attributed to the self-trapping of photo-

generated electrons, whereas the signal at g = 2.039 is assigned to the self-trapping of photo-generated holes[34-36]. These spectral features serve as clear evidence of the formation of electron and HPs, respectively. In contrast, the EPR signal for $In/BiVO_4$ was negligible, indicating that the generation of HPs under illumination was effectively suppressed in $In/BiVO_4$. Additionally, temperature-dependent photoluminescence (Td-PL) spectroscopy was employed to probe the underlying mechanism of electron-phonon coupling in the suppression of HPs. It was observed that the intensity of the PL peak decreased with increasing temperature, suggesting that the phonon-assisted recombination of photo-generated carriers is suppressed at lower temperatures (Fig. 5b and Supplementary Fig. 27b)[37]. The electron-phonon coupling strength was quantitatively evaluated using the Huang-Rhys factor (S), which was extracted by fitting the temperature-dependent full width at half maximum (FWHM) of the PL peak using Eq. 1[38,39].

$$FWHM = 2.36\sqrt{S}E_{phonon}\sqrt{\coth\frac{E_{phonon}}{2k_bT}} \qquad (1)$$

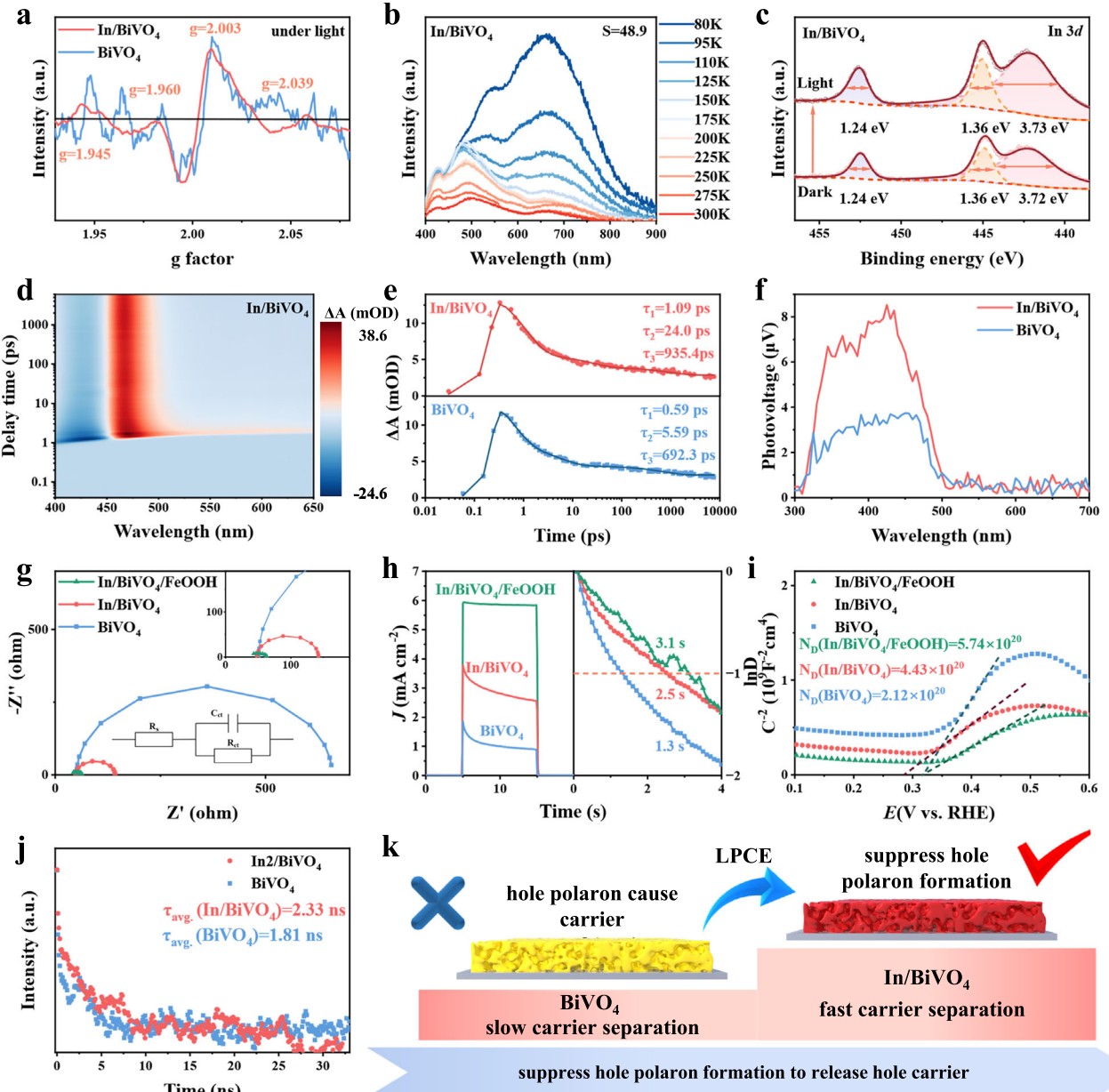

**Fig. 5 | Analysis of HPs and carrier properties. a** EPR spectroscopy of BiVO$_4$ and In/BiVO$_4$ photoanodes under illumination. **b** Td-PL spectroscopy of In/BiVO$_4$ photoanode. **c** ISI-XPS high-resolution spectra of In 3$d$ in In/BiVO$_4$ photoanode. **d** Transient absorption spectra of In/BiVO$_4$ photoanode. **e** The decay kinetics monitored at 515 nm of BiVO$_4$ and In/BiVO$_4$ photoanodes. **f** SPV spectroscopy of BiVO$_4$ and In/BiVO$_4$ photoanodes. **g** EIS spectroscopy, **h** chopped chronoamperometry curves and normalized transient current-time plots, **i** M-S spectroscopy of BiVO$_4$, In/BiVO$_4$ and In/BiVO$_4$/FeOOH photoanodes. **j** TRPL spectroscopy of BiVO$_4$ and In/BiVO$_4$ photoanodes. **k** Schematic diagram of hole carrier release through hole polaron suppression. The data are presented without iR correction.

where S and E$_{phonon}$ represent the electron-phonon coupling strength and the phonon energy, respectively. The S for In/BiVO$_4$ was calculated to be 48.9, whereas for BiVO$_4$, it was considerably higher at 96.5 (Supplementary Fig. 27). The significantly reduced S value for In/BiVO$_4$ indicates a marked suppression of carrier-phonon coupling. To reveal the changes in the strength of electron–phonon coupling at different polaron sites and thereby confirm the selective suppression of HPs, we conducted in situ irradiation XPS (ISI-XPS) measurements. During the self-trapping of photoexcited carriers to form polarons, the electron-phonon coupling leads to an increase in the peak broadening of XPS[40]. In the case of the BiVO$_4$ photoanode, the Bi 4$f$ and V 2$p$ spectra exhibit a significant increase in FWHM under illumination, indicating the localization of both HPs and EPs at the Bi and V sites, respectively

(Supplementary Fig. 28). In contrast, in the In/BiVO$_4$ photoanode, the In site shows no significant increase in FWHM, the Bi site exhibits a slight increase, and the V site still shows a clear broadening (Fig. 5c and Supplementary Fig. 28). This indicates that the In substitution weakens the electron-phonon coupling at HP sites, thus suppressing HP formation, while having a limited effect on EPs. Finally, we employed fs-TAS to provide time-resolved information on polaron dynamics and formation times. Both BiVO$_4$ and In/BiVO$_4$ show a negative absorption band in the 400–450 nm range, corresponding to ground-state bleaching (GSB), and a positive absorption band in the 450–500 nm range, corresponding to excited-state absorption (ESA) due to photo-induced carrier excitation (Fig. 5d, Supplementary Fig. 29)[41]. At the same time, both samples display a similar absorption decay process,

which gradually decreases after 500 fs, corresponding to the carrier recombination process. Notably, compared to $BiVO_4$, the stronger positive absorption signal in $In/BiVO_4$ indicates a higher carrier concentration and more efficient carrier separation[5]. Subsequently, the kinetic decay curves at 515 nm were fitted using a three-exponential model. As shown in Fig. 5e, both $BiVO_4$ and $In/BiVO_4$ exhibit three distinct decay time constants. The shortest time constant ($\tau_1$) is attributed to the formation of HPs, while $\tau_2$ and $\tau_3$ are associated with electron transfer and carrier recombination, respectively[42,43]. Notably, the $\tau_1$ of $In/BiVO_4$ is 1.09 ps, significantly longer than that of $BiVO_4$ (0.59 ps). This suggests that the self-trapping process of HPs is slowed down, indicating that HP formation is effectively suppressed. In addition, the longer $\tau_2$ and $\tau_3$ in $In/BiVO_4$ suggest a reduced carrier recombination rate and an extended carrier lifetime, which is beneficial for more active carriers to participate in the catalytic reaction[44]. In addition, the polaron hopping activation energy ($E_h$) indirectly indicates that In substitution has little effect on the hopping of EPs, suggesting that our strategy mainly modulates the HP sites and suppresses the formation of HPs (Supplementary Fig. 30). SPV spectroscopy results further support this conclusion, with a markedly enhanced SPV signal indicating an increased accumulation of positive charges at the surface of the $In/BiVO_4$ photoanode (Fig. 5f)[45]. This is attributed to the suppression of polaron formation, which facilitates the release of more hole carriers, resulting in a hole-enriched surface. The combined results from EPR, Td-PL, and SPV spectroscopy demonstrate that the surface substitution of In inhibits the formation of HPs by reducing electron-phonon coupling, thereby releasing more photogenerated holes. These findings are in good agreement with the predictions from the DFT calculations.

Following the suppression of HP formation and the subsequent release of additional photogenerated holes, the carrier properties of the photoanodes were significantly modified. These modifications were analyzed using electrochemical testing and spectroscopic characterization. Furthermore, the carrier properties and catalytic performance of the $In/BiVO_4/FeOOH$ photoanode were also compared to further elucidate the role of $In^{3+}$ substitution. The LSV curves under dark conditions, electrochemical double-layer capacitance ($C_{dl}$), and Tafel slopes were employed to reveal the changes in the catalytic activity of the photoanodes (Supplementary Fig. 31). The results indicate that the surface $In^{3+}$ substitution does not significantly enhance the kinetics of the water oxidation reaction, whereas the FeOOH co-catalyst, due to its efficient carrier transport properties and catalytic activity, significantly improves the catalytic performance of the photoanode.

The influence of $In^{3+}$ on the carrier properties of the photoanode was further investigated using electrochemical testing. Electrochemical impedance spectroscopy (EIS) results (Fig. 5g, Supplementary Table 13) show that the charge transfer resistance ($R_{ct}$) of $In/BiVO_4$ is 94.0 Ω, markedly lower than that of pristine $BiVO_4$ (607.6 Ω). This finding suggests that $In^{3+}$ substitution significantly reduces interfacial charge transfer resistance, thereby improving carrier transport kinetics. Furthermore, the loading of FeOOH further reduces the $R_{ct}$ to 19.9 Ω, demonstrating a significant improvement in carrier mobility and a reduction in recombination. By analyzing the current density distribution under chopped light and integrating the data, the decay rate ($\tau D$) of transient photocurrent was determined, which reflects carrier separation and transport dynamics (Fig. 5h). The $\tau D$ for pristine $BiVO_4$ is notably short (1.3 s), indicative of substantial carrier recombination and poor transport capability. Conversely, the $\tau D$ for the $In/BiVO_4$ photoanode increased significantly to 2.5 s, demonstrating that $In^{3+}$ substitution effectively enhances carrier transport capacity and inhibits recombination. With the addition of FeOOH, the $\tau D$ of $In/BiVO_4/FeOOH$ was extended to 3.1 s, reflecting improved carrier transport dynamics. Meanwhile, the increase in the ratio of steady-state to transient photocurrent density (from 0.48 to 0.67) after In

substitution further indicates the suppression of surface recombination and the efficient release of surface holes (Supplementary Fig. 32a). In addition, the higher open-circuit potential (OCP) of the $In/BiVO_4$ photoanode also demonstrates an enhanced hole transport and injection capability (Supplementary Fig. 32b). The positive slopes observed in the Mott-Schottky analysis (Fig. 5i) indicate that all photoanodes exhibit n-type semiconductor characteristics[46]. Notably, the $In/BiVO_4$ photoanode exhibits a lower slope than pristine $BiVO_4$, suggesting an increase in carrier density. The calculated donor densities ($N_D$) are consistent with this observation, revealing $BiVO_4(2.12 \times 10^{20}) < In/BiVO_4(4.43 \times 10^{20})$, confirming that $In^{3+}$ substitution significantly enhances carrier density[47]. Upon loading with FeOOH, the $N_D$ is further elevated to $5.74 \times 10^{20}$. To further investigate the role of $In^{3+}$ substitution in extending carrier lifetime, time-resolved photoluminescence (TRPL) spectroscopy was performed (Fig. 5j). The results revealed that the carrier lifetime of the $In/BiVO_4$ photoanode (2.33 ns) is significantly enhanced compared to that of the pristine $BiVO_4$ photoanode (1.81 ns), further corroborating that $In^{3+}$ substitution suppresses surface carrier recombination and prolongs carrier lifetime[48].

Collectively, the tests and analyses of carrier characteristics indicate that $In^{3+}$ substitution effectively inhibit surface carrier recombination, extend carrier lifetime, and promote carrier transport. This improvement is attributed to the precise modulation of HP sites (Fig. 5k). Moreover, FeOOH, due to its efficient catalytic activity, can significantly enhance the surface water oxidation capability of the material. It works synergistically with $In^{3+}$ modification to further promote carrier transport at the surface, thereby improving the PEC performance. In conclusion, through the precise modulation of specific sites to suppress electron-phonon coupling and inhibiting the formation of surface HPs, we effectively released a greater number of surface hole carriers, resulting in increased carrier concentration and lifetime. This improvement is critical for improving the overall carrier utilization efficiency.

## Discussion

In summary, we propose an innovative strategy to mitigate surface carrier recombination by selectively suppressing of hole polaron formation. Guided by DFT calculations, we precisely engineered the electronic structure at surface hole polaron sites through in-situ selective substitution $Bi^{3+}$ with isoelectronic but more electronegative $In^{3+}$. This substitution weakens the carrier-phonon coupling, thereby enhancing PEC activity. Comprehensive characterization techniques, including AC HAADF-STEM, etch-XPS, ICP and XAFS, confirmed the in-situ selective substitution of $Bi^{3+}$ with $In^{3+}$. EPR, Td-PL, ISI-XPS and fs-TAS provided clear evidence of the suppression of HPs, resulting in the release of more photogenerated hole carriers and a concomitant decrease in carrier recombination. The $In/BiVO_4/FeOOH$ photoanode, optimized through this polaron-suppression strategy, achieved a photocurrent density of 6.46 mA cm$^{-2}$ at 1.23 $V_{RHE}$, with an ABPE of 2.19%. The unbiased tandem device system constructed using the $In/BiVO_4/FeOOH$ photoanode achieved an STH conversion efficiency of approximately 6%. Our work may reshape the fundamental understanding of HPs and provides a promising strategy for the efficient utilization of holes through the modulation of surface HP sites to release photogenerated hole carriers.

## Methods
### Reagents and materials
Fluorine-doped tin oxide (FTO) coated glass (14 Ω sq$^{-1}$, thickness 2.2 mm) was used as the current collector and substrate, purchased from Suzhou Jieweiman New Energy Technology Co., Ltd (China). Vanadyl acetylacetonate (VO(acac)$_2$, 99.0%), bismuth nitrate pentahydrate (Bi(NO$_3$)$_3$·5H$_2$O, 99.0%), potassium iodide (KI, 99.0%), anhydrous ethanol (99.7%), p-benzoquinone (99.0%), and dimethyl sulfoxide

(DMSO, 99.0%) were obtained from Aladdin Reagent (Shanghai) Co., Ltd. Indium nitrate pentahydrate (In(NO$_3$)$_3$·5H$_2$O, 99.0%), choline chloride (98.0%), ethylene glycol (99.5%), and ferrous sulfate heptahydrate (FeSO$_4$·7H$_2$O, 98.0%) were purchased from Tianjin Heowns Opde Technology Co., Ltd. The chemical reagents used for electrolyte preparation in electrochemical tests, such as potassium tetraborate (99.5%) and sodium sulfite(99.0%), were purchased from Sinopharm Chemical Reagent Co., Ltd. All chemicals were used as received without further purification. Deionized water was self-prepared.

### Preparation of pristine BiVO$_4$ photoanodes

The fabrication of pristine BiVO$_4$ photoanodes involved a two-step process: electrodeposition of BiOI precursors followed by thermochemical conversion[28,49]. Initially, the electrolyte was prepared by dissolving Bi(NO$_3$)$_3$·5H$_2$O (0.5 g) and KI (3 g) in deionized water (50 mL), with the pH adjusted using HNO$_3$ (170 μL). This mixture was then combined with an anhydrous ethanol solution (20 mL) containing p-benzoquinone (0.5 g) under vigorous stirring for 3 min. Electrodeposition was conducted in a three-electrode system (FTO working electrode, 4 M KCl Ag/AgCl reference electrode, and Pt counter electrode) at a bias of −0.1 V vs. Ag/AgCl for 6 min. After rinsing and drying the deposited BiOI films, a vanadium precursor solution (40 μL, 2 M VO(acac)$_2$ in DMSO) was dropwise added onto the surface. The electrodes were pre-heated at 120 °C for 10 min to evaporate the solvent, followed by annealing in a muffle furnace at 450 °C for 30 min (ramp rate: 5 °C min$^{-1}$) to generate BiVO$_4$. Finally, the surface V$_2$O$_5$ impurities were dissolved by immersing the samples in 1 M NaOH for 10 min, followed by a final rinse with deionized water. The thickness of the BiVO$_4$ film was determined to be approximately 1.9 μm, as confirmed by cross-sectional SEM analysis (Supplementary Figs. 9, 10).

### Preparation of In$_X$/BiVO$_4$ (X = 1, 2, 5, 10)photoanodes

In$_X$/BiVO$_4$ (X = 1, 2, 5, 10) photoanodes were prepared via liquid-phase cation substitution. Briefly, 2.5 g of choline chloride and 0.25 g of In(NO$_3$)$_3$·5H$_2$O were dissolved in 2.4 g of ethylene glycol to form a homogeneous solvent. The pristine BiVO$_4$ photoanode was immersed in this solvent, heated to 80 °C on a heating plate, and maintained for X hours (X = 1, 2, 5, 10). The obtained samples were denoted as In$_X$/BiVO$_4$ (X = 1, 2, 5, 10).

### Preparation of In/BiVO$_4$/FeOOH and BiVO$_4$/FeOOH photoanodes

The FeOOH co-catalyst was loaded onto In/BiVO$_4$ and BiVO$_4$ photoanodes via photo-assisted electrodeposition. Specifically, the In/BiVO$_4$ or BiVO$_4$ photoelectrode was immersed in a 0.1 M FeSO$_4$ solution and electrodeposited for 12 minutes under AM 1.5 G illumination with a bias voltage of +0.25 V$_{Ag/AgCl}$. The resulting samples were denoted as In/BiVO$_4$/FeOOH and BiVO$_4$/FeOOH, respectively.

### Material characterization

X-ray diffraction (XRD) measurements were carried out using a Bruker D8 Advance X-ray diffractometer with a Cu Kα radiation source (λ = 0.1541 nm) to study the crystal structure in the 2θ range of 10˚−90˚. Raman analyses were performed on a Renishaw inVia Raman microscope equipped with a 532 nm laser to characterize the phase composition and chemical bonding properties of the samples. Scanning electron microscopy (SEM) measurements were performed using a Hitachi S-4800 microscope (Hitachi, Japan) operated at 5 kV. Transmission electron microscopy (TEM) measurements were conducted using a JEOL JEM-2100F (JEOL, Japan) microscope operated at 200 kV, combined with energy-dispersive X-ray spectroscopy (EDX) to examine the material morphology and elemental distribution. Spherical aberration-corrected high-angle annular dark-field scanning transmission electron microscopy (AC HAADF-STEM) measurements were

carried out using a JEOL JEM-ARM200F (JEOL, Japan) microscope. X-ray photoelectron spectroscopy (XPS) and ultraviolet photoelectron spectroscopy (UPS) were performed using Thermo Scientific K-Alpha (Thermo Scientific, America) and Thermo Fisher Scientific ESCALAB XI + (Thermo Scientific, America). Measurements of In K-edge and Bi L-edge XAFS were carried out using the BL14W beamline at SSRF (Shanghai, China), which is equipped with Si(111) crystal monochromators. Before the measurements, the samples were sealed within aluminum holders using Kapton film to ensure stability during data collection. XAFS data acquisition was performed at room temperature utilizing a 4-channel Silicon Drift Detector (SDD, Bruker 5040). Specifically, the extended X-ray absorption fine structure (EXAFS) spectra were collected in fluorescence mode for the In K-edge and in transmission mode for the Bi L-edge. All XAFS data were processed and analyzed using Athena and Artemis software[50]. Electron paramagnetic resonance (EPR) measurements were conducted using a Bruker EMXplus-6/1 spectrometer (Germany) utilizing a microwave frequency of 9.84 GHz with an incident power of 6.325 mW. Measurements were conducted under temperature-controlled conditions set to 295.00 K. Field modulation parameters included a frequency of 100.00 kHz with 4.000 G amplitude, employing first-derivative detection and 0.01 ms time constant. The inductively coupled plasma optical emission spectrometer (ICP-OES) measurements were conducted using a Agilent 5110 ICP-OES system with the following operating parameters: plasma gas flow rate of 12.0 L/min, nebulizer gas flow rate of 0.70 L/min, auxiliary gas flow rate of 1.0 L/min, pump rate of 60 r/min, RF power of 1250 W, stable time of 20 s, reading access time of 5 s, and sample flush time of 20 s. In-situ irradiation XPS measurements were performed using a ThermoFisher ESCALAB 250Xi X-ray photoelectron spectrometer. The system utilized a monochromatic Al-Kα X-ray source (1486.6 eV) for excitation, operating under ambient conditions with illumination provided by a 300 W xenon lamp (PLS-SXE300E, Beijing Perfectlight, China) emitting light in the wavelength range of 320−780 nm. Prior to conducting the irradiation experiments, XPS spectra of all elemental components in the samples were collected in the dark as reference data. The femtosecond transient absorption spectroscopy (fs-TAS) is performed using the Ultrafast Helios system, which is coupled with a Coherent Astrella laser (> 7 mJ, 800 nm, <100 fs, 1 kHz) and an OPerA-Solo OPA (240−2600 nm tunable), enabling high-sensitivity transient absorption measurements in the 320−1600 nm range with sub-14 fs time resolution. Surface photovoltage (SPV) measurements were performed using a CEL-SPS1000 system. Ultraviolet-visible diffuse reflectance spectroscopy (UV−vis) was carried out on a UV-3100 spectrometer with BaSO$_4$ as the reference. Temperature-dependent photoluminescence (Td-PL) spectroscopy and time-resolved photoluminescence (TRPL) spectroscopy were performed using an Edinburgh FLS920 fluorescence spectrophotometer, equipped with a Shimadzu RF-6000 fluorescence spectrometer.

### PEC measurements

PEC measurements were performed using an electrochemical workstation (CHI660E) in a standard three-electrode system. The light source was a 300 W Xe arc lamp (FX 300HU, Beijing PerfectLight Co., Ltd.) equipped with an AM 1.5 G filter, and the light intensity at the working electrode (WE) was calibrated to 100 mW cm$^{-2}$ using an optical radiometer (FZ-A, Beijing Normal University Photoelectric Instrument Factory). The spectrum of the light source was confirmed to match the standard AM 1.5 G spectrum using a portable spectrometer (labpatSS1, Zhipu Tech (Hefei) Co., Ltd.) (Supplementary Fig. 24). A 0.2 M KBi solution (pH = 8.4) was used as the electrolyte, and the test temperature was maintained at 20 °C. All PEC measurements were conducted under backside illumination through the FTO glass. Current-potential (J-V) characteristics were recorded by sweeping the potential in the positive direction at a scan rate of 20 mV s$^{-1}$. The cyclic voltammetry (CV) curves were measured under dark conditions at a potential range of 0.5−0.6 V$_{RHE}$. EIS measurements were conducted by

AC impedance spectroscopy under a bias of 1.9 $V_{RHE}$, in the frequency range from $10^{-1}$ to $10^4$ Hz, without illumination. Mott–Schottky (M-S) plots were measured at a frequency of $10^3$ Hz.

The Ag/AgCl reference electrode was employed as the reference for all electrochemical and photoelectrochemical measurements. The measured potentials vs. Ag/AgCl were converted to the reversible hydrogen electrode scale according to the Nernst equation:

$$E_{RHE} = E_{Ag/AgCl} + E_{Ag/AgCl}^{\Theta} + 0.059 \times pH \qquad (2)$$

$$E_{Ag/AgCl}^{\Theta} = 0.1976 V_{RHE} \qquad (3)$$

where $E_{Ag/AgCl}$ is the potential measured with respect to the reference electrode, and $E_{Ag/AgCl}^{\Theta}$ is the standard potential of the Ag/AgCl reference electrode in the solution.

IPCE were measured at 1.23 $V_{RHE}$ using the same three-electrode setup as for PEC photocurrent measurements. The IPCE was calculated using the equation

$$IPCE = 1240(V \cdot nm) \times I(mA\,cm^{-2})/(\lambda(nm) \times P_{light}(mW\,cm^{-2})) \times 100\% \qquad (4)$$

where 1240 represents a multiplication of Planck's constant ($h$) and the light speed ($c$), $I$ is the photocurrent density, $\lambda$ is the wavelength of incident light, and $P_{light}$ is the measured light power density at that wavelength.

Assuming 100% faradaic efficiency, ABPE was calculated using the following equation:

$$ABPE = I(mA\,cm^{-2}) \times (1.23 - V_{app})(V)/P_{light}(mW\,cm^{-2}) \qquad (5)$$

where $I$ is the photocurrent density, $V_{app}$ is the applied potential, and $P_{light}$ is the incident illumination power density (100 mW cm$^{-2}$).

$\eta_{separation}$ and $\eta_{injection}$ was calculated using the following equations:

$$\eta_{separation} = J_{sulfite}/J_{abs} \qquad (6)$$

$$\eta_{injection} = J_{water}/J_{sulfite} \qquad (7)$$

where $J_{sulfite}$ and $J_{water}$ are the photocurrent densities for PEC sulfite oxidation and water oxidation, respectively.

Gas evolution experiments ($H_2$ and $O_2$) were conducted in a gas-tight PEC cell containing 0.2 M KBi electrolyte. Under simulated solar irradiation (AM 1.5 G), the photocurrent was maintained at a constant bias of 1.23 $V_{RHE}$. An online gas chromatography system (GC-9790, Tianmei) was employed to quantitatively monitor the gas products. Consequently, the faradaic efficiency of the In/BiVO$_4$/FeOOH photoanode was determined by comparing the experimentally detected gas with that of their theoretical calculation.

The STH was calculated using the following equation:

$$STH = \frac{J_{OP} \times 1.23}{P_{light}} \times 100\% \qquad (8)$$

where $P_{light}$ is the power of the illuminating light, $J_{OP}$ is the photocurrent density at the intersection point.

Donor density ($N_D$) were calculated from the Mott-Schottky equations:

$$\frac{1}{C^2} = \left( \frac{2}{A^2 e_0 \varepsilon \varepsilon_0 N_d} \right) \left[ V - V_{fb} - \frac{K_B T}{e_0} \right] \qquad (9)$$

$$N_D = \left( \frac{2}{A^2 e_0 \varepsilon \varepsilon_0} \right) \left[ \frac{d(\frac{1}{c^2})}{dV} \right]^{-1} \qquad (10)$$

where C is capacitance, A is the area of the working electrode (1 × 1 cm$^2$), $N_d$ is carrier density, $e_0$ is electron charge ($1.6 \times 10^{-19}$ C), $\varepsilon$ is dielectric constant of BiVO$_4$ (69), $K_B$ is Boltzmann constant ($K_B = 1.38 \times 10^{-23}$ m$^2$ kg s$^{-2}$ K$^{-1}$), $\varepsilon 0$ is the permittivity of free space ($\varepsilon_0 = 8.85 \times 10^{-14}$ F m$^{-1}$), T is the absolute temperature (T = 298 K), V is applied bias, and $V_{fb}$ is flat-band potential.

The $i_0/i$ value was calculated from current density distribution under chopped light, using equation:

$$i_0/i = J_{ss}/J_{in} \qquad (11)$$

where $J_{ss}$ is steady-state photocurrent density and $J_{in}$ is initial photocurrent density.

A tandem device was designed based on an In/BiVO$_4$/FeOOH photoanode and a commercially available Si solar cell for unbiased solar water splitting. Specifically, the front side of the In/BiVO$_4$/FeOOH photoanode was aligned opposite to the positive terminal side of the Si solar cell. The assembly was illuminated from the rear side of the photoanode using an AM 1.5 G simulated solar xenon lamp, allowing transmitted light to irradiate the front surface of the Si solar cell, thus constructing an unbiased tandem device for solar water splitting.

## DFT computational methods

The reaction energetics were investigated using density functional theory (DFT) calculations performed with the Vienna Ab initio Simulation Package (VASP), employing the Perdew-Burke-Ernzerhof (PBE) functional within the generalized gradient approximation (GGA)[51,52]. A plane-wave basis set with a cut-off energy of 450 eV was employed to expand the electronic wave functions. Structural relaxation was carried out until the convergence criteria for energy and residual forces reached $10^{-5}$ eV and 0.02 eV/Å, respectively. Furthermore, to accurately simulate the formation of hole polarons, a $2 \times 2 \times 1$ supercell was adopted using the HSE06 hybrid functional, where the exchange and correlation energies are expressed as:

$$E_{XC}^{HSE} = \frac{1}{4} E_X^{SR}(\mu) + \frac{3}{4} E_X^{PBE,SR}(\mu) + E_X^{PBE,LR}(\mu) + E_C^{PBE} \qquad (12)$$

where $\mu = 0.207$ Å$^{-1}$, which is considered a reasonable value to satisfy both accuracy and cost[53]. For the HSE geometry optimization, we use the single Γ k-point approximation while simulating the hole polaron by removing an electron from the system. To locate the hole polaron, we extend the bond around the localization site by about 0.2 Å to break the local symmetry[54].

For the electronic structure calculations under non-photoexcited conditions, we used the DFT + U method to overcome the self-interaction error of the DFT with an effective Hubbard value of $U_{eff}$ (V) = 3 eV[55]. We adopted the original cell parameters taken from experiments (a = 5.092 Å, b = 5.195 Å, c = 11.701 Å, a = β = 90.0 ° and γ = 90.3 °) and used a 6 × 6 × 3 network of k-points for structural relaxation. At the same time, we compared the post-self-consistency energies of four different substitution sites to determine the optimal substitution model. The BiVO$_4$ surface model was constructed by cutting the block BiVO$_4$ 110 crystal face with the vacuum layer in the z-direction set to 15 Å to avoid interactions. For the calculation of the band structure and density of states, the VASPKIT code was used[56]. VESTA was used for the visualization of the crystal structure[57].

## Data availability

The data that support the findings of this study are available from the source data. The atomic coordinates generated in this study are available in Supplementary Data 1. Source data are provided with this paper.

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

## Acknowledgements

The authors acknowledge the financial support by the National Natural Science Foundation of China (Grant No. 52172222 and Grant No. U24A20202 to Fang He).

## Author contributions

H.L. conceived the original concept with discussion with F.H. and K.S.; H.L. carried out most of the preparations, characterizations, and wrote the first draft under the guidance of S.K.; H.C. and J.C. assisted the synthesis of photoanodes. J.D. and Y.F. assisted with characterization and testing. G.Y. and G.C. assisted with the DFT calculations. F.H. and J.Y. guided the entire research work. B.C., C.H. and N.Z. guided the revision of the manuscript. All authors discussed the results and contributed to the manuscript.

## Competing interests

The authors declare no competing interests.
