## [Transparent Peer Review file · Nature Communications]

Surface hole polaron site tuning governs charge carrier separation in BiVO₄ photoanodes

Corresponding Author: Professor Fang He

Version 0:

Reviewer comments:

Reviewer #1

(Remarks to the Author)

The use of bismuth vanadate (BiVO₄) as a photoanode is proposed to facilitate the regulation of surface hole polarons, which are known to hinder charge transport and reduce photoelectrochemical efficiency. The incorporation of an FeOOH co-catalyst is shown to enhance performance by mitigating polaron formation and improving surface charge transfer kinetics. To validate this mechanism, a combination of density functional theory (DFT) calculations and advanced characterization techniques including X-ray absorption near-edge structure (XANES), extended X-ray absorption fine structure (EXAFS), electron paramagnetic resonance (EPR), Raman spectroscopy, and scanning electron microscopy (SEM) is proposed. While various metal oxides have been explored as photoanodes, their reported solar-to-hydrogen (STH) conversion efficiencies remain limited. Therefore, achieving precise control over surface hole polaron sites is important for enhancing charge carrier dynamics and advancing the efficiency of current photoelectrochemical cells. However, there are severe limitations in this work and the reported results cannot currently be reproduced.

1. The characterization section lacks sufficient detail and requires significant improvement. More details are needed on all the separate and various techniques being used including SEM, TEM, EDX, XANES and EXAFS.
2. The EXAFS analysis section requires substantial clarification. First, the bond distance resolution of the EXAFS experiments is not specified. Additionally, the absence of k-space data prevents the reviewer from evaluating the quality of the raw EXAFS signal and the reliability of the Fourier-transformed results. The manuscript does not mention the specific EXAFS fitting equation used, nor does it detail which parameters were fitted versus those that were fixed during the modeling process. References for the software being used are also needed in the experimental section. As this is an experimental paper, those details are crucial and this is therefore a severe shortcoming. In Table S8 (Page 29), several parameters are elaborated but none of them are explained. It is also not clear to which reviewer how the coordination numbers were determined. Was a range of coordination numbers fitted and how did the reduced chi square values change?
3. Line 189-190, EXAFS section mentions that the coordination number of In-O in the first coordination sphere of In/BiVO₄ 8.24 close to that of BiVO₄ but no comparison is shown. The EXAFS of In/BiVO₄ has to be superimposed with that of BiVO₄ for comparisons when discussing this section.
4. The way in which the coordination number is determined have to be clarified. It is not clear from the Figure 3 e whether the coordination number of In-O in In₂O₃ is lower than that for BiVO₄. These 2 EXAFS spectra have to be once again superimposed for clarity.
5. It is not clear to this reviewer how the WT simulations were conducted?
6. It is not clear how the Bader charge analysis was carried out. The details in this section are quite limited.
7. The section on the EPR measurements is unclear. Line 301-202 mentions appearance of new peaks at $g = 1.96$ and 2.039 for BiVO₄, however several other peaks are shown which are not attributed, fitted, or explained.

8. The material section only mentions the user of a Bruker Instrument for the EPR measurements without any further details. What microwave frequency was used? At what temperature were the measurements conducted? What modulation frequency? What power?

Reviewer #2

(Remarks to the Author)

In order to improve photoelectrochemical (PEC) water-splitting performance, the manuscript presents a surface-engineering approach for BiVO₄ photoanodes using In³⁺ substitution to suppress hole polaron formation. The results are generally convincing, the work is well-written, and it is technically sound. To improve the manuscript, though, a few queries and explanations are required, especially regarding the mechanism of hole polaron suppression and pec performance as follows

1. In the DFT section the author claims that the bond length changes for the Bi-O while it remains same for the In-O case. What is the correlation of this physical phenomenon with the polaron formation. It needs more explanation for better understanding.
2. In the XPS analysis Shifts in Bi 4f and V 2p can also be influenced by local strain and oxygen vacancies?
3. Can the author provide the depth profiling with detailed discussion?
4. XANES analysis need more explanation.
5. About the PEC data, how reproducible this data set? how many devices were tested? it is advised to show different devices data set also provide error bars.
6. For the electrochemical testing did author calibrate the reference electrode? how this conversion happened to RHE. Please do the calibration and present in the revised version with detailed explanations.
7. Please provide the Tafel slope as well for better understanding of electrochemical performance with explanations.
8. Overall, the flow of the manuscript is good, please check again about minor grammatical mistakes and errors.

Reviewer #3

(Remarks to the Author)

In this study, guided by DFT calculations, the authors synthesized an In/BiVO₄ photoanode by substituting Bi³⁺ with less electronegative In³⁺. This substitution weakens the surface electron-phonon coupling in BiVO₄, thereby suppressing its PEC activity. The claim of successful substitution and the inhibition of polaron formation is supported by comprehensive characterization techniques, including AC-HAADF-STEM, XAFS, EPR, and Td-PL. Furthermore, the optimized In/BiVO₄/FeOOH photoanode achieves a water oxidation photocurrent density of 6.46 mA cm⁻² at 1.23 V RHE. This work presents a strategic approach to enhance hole utilization efficiency through the modulation of surface hole polaron sites to facilitate carrier release. Nevertheless, the following issues should be addressed before the manuscript can be considered for publication in this journal:

1. The photocurrent of this In/BiVO₄/FeOOH photoanode is relatively high. Therefore, the authors must provide IPCE spectra and irradiation data by wavelength from the solar simulator. Also, the integrated photocurrent density based on the IPCE and solar irradiation (AM1.5G) should be compared to the photocurrent density on the J-V curves. Regarding IPCE spectra of BiVO₄, it is recommended to refer to the following article: *Energy Environ. Sci.* 17, 2541-2553 (2024); *J. Am. Chem. Soc.* 2021, 143, 20657–20669.
2. After modification with FeOOH, the photocurrent density of the In/BiVO₄ photoanode reached approximately 6.5 mA/cm², whereas FeOOH/BiVO₄ and In/BiVO₄ alone exhibited photocurrent densities of only about 3.0 mA/cm². This performance enhancement appears consistent with the mechanism reported in previous studies (*Angew. Chem. Int. Ed.* 2023, e202307246; *J. Am. Chem. Soc.* 2021, 143, 20657–20669). For instance, in this manuscript, the enhanced PEC activity can be attributed to the presence of In on the FeOOH/In/BiVO₄ surface, which facilitated charge carrier migration of In/BiVO₄ toward FeOOH through the formation of In–O–Fe bonds.
3. The selective substitution of Bi³⁺ with In³⁺ on the surface of BiVO₄ reduces electron–phonon coupling. Theoretically, a higher In³⁺ substitution level at surface Bi sites should further suppress electron–phonon interactions, thereby improving PEC performance. However, experimental observations are inconsistent with this theoretical expectation. The authors should provide further explanation and discussion to clarify this discrepancy.
4. The authors suggest that the weaker electronegativity of In compared to Bi may contribute to the reduced electron–phonon coupling. It would be valuable to explore whether substituting Bi with other metals of lower electronegativity could similarly enhance the PEC performance of BiVO₄.
5. In the Figure 4h, the performance comparison of currently reported BiVO₄-based photoanodes appears incomplete. For instance, several recent studies have demonstrated BiVO₄-based photoanodes achieving photocurrent densities exceeding 6.5 mA·cm⁻², such as those reported in *Energy Environ. Sci.*, 2022, 15, 2867–2873; *Angew. Chem. Int. Ed.*, 2025, 64(4), e202416340; and *Nat. Commun.*, 2025, 16, 2792. The inclusion of these representative works would provide a more balanced and up-to-date context.
6. It is recommended to supplement the performance data of the In/BiVO₄/FeOOH photoanode missing in Figure 5 to further elucidate the specific role of In³⁺.

Reviewer #4

(Remarks to the Author)
Recommendation: Reject

The manuscript addresses the role of polarons in BiVO₄ photoanodes, particularly focusing on hole polarons. While the topic is of potential interest to the community working on photoelectrochemical catalysis, the current work suffers from significant conceptual, methodological, and presentation issues that preclude its publication in Nature Communications. My major concerns are as follows:

1. The hole polarons (HPs) in BiVO₄ are widely regarded as small polarons with relatively low activation energies compared to small electron polarons (EPs), which are often considered quasi-free carriers. The manuscript exclusively emphasizes HPs without any discussion or investigation of EPs. The lack of a balanced treatment weakens the overall scientific rationale.
2. The manuscript contains numerous typos and grammatical errors that must be carefully revised. The current language issues significantly hinder readability.
3. Electron and hole polarons in BiVO₄ are distinct species, and their different energetic positions within the bandgap need to be clearly discussed. The current Figure 1 is misleading, as it does not differentiate the locations of EPs and HPs. The authors should provide convincing evidence that HPs, rather than EPs, dominate charge recombination in BiVO₄.
4. Small polaron formation typically induces lattice expansion to accommodate the localized carrier. However, the theoretical results in this work indicate a reduced Bi–O bond length, which appears contradictory. The authors should provide a clear explanation of this discrepancy.
5. The claim that In³⁺ substitutes Bi³⁺ rather than V⁵⁺ lacks direct proof. Rigorous evidence combining DFT simulations with experimental characterization is required to substantiate this critical point.
6. In lines 308–309, the authors argue that phonon-assisted recombination of photo-generated carriers is enhanced at higher temperatures. This interpretation is inconsistent with the reported results, where one would instead expect recombination to be suppressed at elevated temperatures. This contradiction requires clarification.
7. The Td-PL results presented in the manuscript may suggest reduced polaron hopping, but they do not provide direct evidence that the observed effects stem from HPs rather than EPs. Additional experimental data are necessary to support the claim of enhanced HP hopping in contrast to EP hopping.

Version 1:

Reviewer comments:

Reviewer #1

(Remarks to the Author)
The authors have satisfactorily answered all of my comments and concerns.

Reviewer #2

(Remarks to the Author)
All issues addressed by this reviewer were well revised and explained.

Reviewer #3

(Remarks to the Author)
The authors provide detailed and satisfactory responses to the comments raised, which makes the manuscript more concise and clear. Therefore, I am in favor of its publication.

Reviewer #4

(Remarks to the Author)
I appreciate the authors' extensive effort in revising the manuscript and providing additional calculations and characterizations. However, after carefully evaluating the rebuttal and revised version, I find that the major scientific concerns I raised remain largely unresolved.

1. Most of the authors' new explanations, formation energies, charge density maps, Bader charge analyses, DOS, and bond-length changes, are purely theoretical. While these calculations are useful for conceptual support, they do not constitute direct experimental evidence for the proposed suppression of hole polarons (HPs) or the claimed enhancement of HP-mediated processes. The central mechanistic claim therefore remains insufficiently validated.
2. The rebuttal repeatedly states that separating hole vs. electron polarons experimentally is "difficult," yet the manuscript continues to make strong claims about HP-dominated recombination and HP-specific suppression. The added EIS-derived

activation energies do not experimentally isolate HP dynamics; they primarily reflect electron hopping. Thus, the key mechanistic assignment still lacks direct experimental grounding.

3. In BiVO₄, small electron polarons typically dominate bulk transport limitations, whereas hole polarons localize strongly at Bi–O motifs and are known to be less mobile. This fundamental understanding is well documented. The manuscript's central narrative that HPs are the primary limitation and require selective suppression does not convincingly introduce new physics. The responses emphasize literature consistency rather than presenting new experimental breakthroughs.

4. Although the authors provide extensive DFT evidence that In substitution increases HP formation energy, the manuscript still lacks direct observation of hole release from HP sites. As a result, the proposed "HP suppression" as the primary enhancement mechanism remains speculative.

5. While the topic is interesting and relevant to the BiVO₄ and polaron communities, the mechanistic claims are still not supported by sufficiently strong and direct experimental evidence. The revised manuscript appears to expand theoretical detail rather than resolve the scientific gaps.

In summary, I appreciate the authors' effort, but the concerns regarding conceptual solidity and experimental validation remain unresolved. I therefore maintain my previous recommendation.

Version 2:

Reviewer comments:

Reviewer #4

(Remarks to the Author)

The authors have successfully addressed my comments. The reviewer recommends it for a publication now.

Responses to the referees' comments

Dear Reviewers:

*Thank you for your thoughtful comments on our manuscript entitled “**Leveraging surface hole polaron site tuning to enhance carrier separation in BiVO₄ photoanodes**”*

We truly appreciate the time and effort you put into reviewing our work. Your feedback is invaluable and has significantly contributed to the improvement and revision of our paper. We have carefully addressed all of your comments and have made the necessary revisions, which we hope will meet your approval.

Thank you again for your constructive input.

Sincerely,

Fang He

Responses to Reviewer #1:

The use of bismuth vanadate (BiVO₄) as a photoanode is proposed to facilitate the regulation of surface hole polarons, which are known to hinder charge transport and reduce photoelectrochemical efficiency. The incorporation of an FeOOH co-catalyst is shown to enhance performance by mitigating polaron formation and improving surface charge transfer kinetics. To validate this mechanism, a combination of density functional theory (DFT) calculations and advanced characterization techniques including X-ray absorption near-edge structure (XANES), extended X-ray absorption fine structure (EXAFS), electron paramagnetic resonance (EPR), Raman spectroscopy, and scanning electron microscopy (SEM) is proposed. While various metal oxides have been explored as photoanodes, their reported solar-to-hydrogen (STH) conversion efficiencies remain limited. Therefore, achieving precise control over surface hole polaron sites is important for enhancing charge carrier dynamics and advancing the efficiency of current photoelectrochemical cells.

However, there are severe limitations in this work and the reported results cannot currently be reproduced.

Authors: We sincerely appreciate the time and effort you dedicated to evaluating our manuscript. We acknowledge the limitations of the current work and the reproducibility challenges associated with the reported results, and we recognize the importance of providing robust experimental validation and theoretical support for the proposed mechanism. To address the potential reproducibility issues, we have supplemented the manuscript with additional XAFS data to comprehensively substantiate our arguments. We have also thoroughly and meticulously added detailed parameters for various experimental conditions, including characterization parameters, DFT calculation settings, and some data analysis and processing procedures. We hope that these revisions will provide strong support for our reported findings, and we look forward to your further feedback. Once again, thank you for your patience and guidance. Below, we will provide a point-by-point discussion and response to your suggestions.

Comment 1: The characterization section lacks sufficient detail and requires significant improvement. More details are needed on all the separate and various techniques being used including SEM, TEM, EDX, XANES and EXAFS.

Response: We sincerely appreciate your thorough review and constructive suggestions. We agree with the comments that the detailed information regarding the various characterization techniques should be further supplemented and improved. In accordance with the reviewer's recommendations, we have added more comprehensive details to the manuscript, including the experimental parameters and conditions for XRD, Raman, SEM, TEM, EDX, AC HAADF-STEM, XPS, XANES, EXAFS, ICP-OES and EPR (**Material Characterization** in manuscript). In addition, some fitting parameters used in the EXAFS analysis have been added to the Supplementary Information (discussion below **Supplementary Figure 17** and **Supplementary Table 10**). We believe that the addition of more detailed characterization parameters and experimental conditions is of great importance to the scientific rigor and completeness of our manuscript. Once again, we sincerely thank you for your valuable suggestions.

Changes to the revised manuscript are shown below.

Main Manuscript (Methods-Material Characterization):

Page 19-20: X-ray diffraction (XRD) measurements were carried out using a Bruker D8 Advance X-ray diffractometer with a Cu K α radiation source ($\lambda = 0.1541$ nm) to study the crystal structure in the 2θ range of 10° - 90° . Raman analyses were performed on a Renishaw inVia Raman microscope equipped with a 532 nm laser to characterize the phase composition and chemical bonding properties of the samples. Scanning electron microscopy (SEM) measurements were performed using a Hitachi S-4800 microscope (Hitachi, Japan) operated at 5 kV. Transmission electron microscopy (TEM) measurements were conducted using a JEOL JEM-2100F (JEOL, Japan) microscope operated at 200 kV, combined with energy-dispersive X-ray spectroscopy (EDX) to examine the material morphology and elemental distribution. Spherical aberration-corrected high-angle annular dark-field scanning transmission electron microscopy (AC HAADF-STEM) measurements were carried out using a JEOL JEM-ARM200F (JEOL, Japan) microscope. X-ray photoelectron spectroscopy (XPS) and ultraviolet photoelectron spectroscopy (UPS) were performed using Thermo Scientific K-Alpha (Thermo Scientific, America) and Thermo Fisher Scientific ESCALAB XI+ (Thermo Scientific, America). In K-edge and Bi L-edge X-ray absorption fine structure (XAFS) analyses were conducted at the BL14W beamline at the Shanghai Synchrotron Radiation Facility (SSRF, Shanghai, China) using Si(111) crystal monochromators. Before the analysis at the beamline, samples were placed into aluminum sample holders and sealed using Kapton tape film. XAFS spectra were recorded at room temperature using a 4-channel Silicon Drift Detector (SDD, Bruker 5040). In K-edge and Bi L-edge extended X-ray absorption fine structure (EXAFS) spectroscopy were recorded in fluorescence mode and transmission mode respectively. All XAFS datas were processed and analyzed using Athena and Artemis software⁴⁴. Electron paramagnetic resonance (EPR) measurements were conducted using a Bruker EMXplus-6/1 spectrometer (Germany) utilizing a microwave frequency of 9.84 GHz with an incident power of 6.325 mW. Measurements were conducted under temperature-controlled conditions set to 295.00 K. Field modulation parameters included a frequency of 100.00 kHz with 4.000 G amplitude, employing first-derivative detection and 0.01 ms time constant. The inductively coupled plasma

optical emission spectrometer (ICP-OES) measurements were conducted using a Agilent 5110 ICP-OES system with the following operating parameters: plasma gas flow rate of 12.0 L/min, nebulizer gas flow rate of 0.70 L/min, auxiliary gas flow rate of 1.0 L/min, pump rate of 60 r/min, RF power of 1250 W, stable time of 20 s, reading access time of 5 s, and sample flush time of 20 s. Surface photovoltage (SPV) measurements were performed using a CEL-SPS1000 system. Ultraviolet-visible diffuse reflectance spectroscopy (UV-vis) was carried out on a UV-3100 spectrometer with BaSO₄ as the reference. Temperature-dependent photoluminescence (Td-PL) spectroscopy and time-resolved photoluminescence (TRPL) spectroscopy were performed using an Edinburgh FLS920 fluorescence spectrophotometer, equipped with a Shimadzu RF-6000 fluorescence spectrometer.

Supplementary Information:

Page 21-22 (discussion below Supplementary Figure 17): Wavelet transform analysis was performed using the commercial open-source software HAMA. The Morlet wavelet was selected for its capability to discriminate contributions from atoms at similar interatomic distances by resolving varying amplitudes and oscillating phase^{9,10}. This approach leverages the two-dimensional distribution of EXAFS signals in k-space and R-space, enabling effective separation of contributions from multiple scattering paths. The equation of Morlet wavelet is shown in equation 1⁹:

$$\Psi(t) = \frac{1}{\sqrt{2\pi}\sigma} \left[e^{ikt} - e^{-\frac{k^2}{2}} \right] e^{-\frac{t^2}{2\sigma^2}} \quad (\text{S2})$$

Wave number k was set to 10, corresponding to the frequency of sine/cosine basis functions. This free parameter determines the number of wave oscillations within the Gaussian envelope. Scale parameter σ was set to 1, defining the half-width of the Gaussian envelope that localizes the wavelet in time-frequency space.

Page 46 (discussion below Supplementary Table 10): The obtained XAFS data was processed in Athena for background, pre-edge line, and post-edge line calibrations¹³. The EXAFS data were processed according to the standard procedures using Artemis software¹³. To obtain the quantitative structural parameters around central atoms, least-squares curve parameter fitting was performed. The following EXAFS equation was used¹⁰:

$$\chi(k) = \sum_j \frac{N_j S_0^2 F_j(k)}{k R_j^2} e^{-2\sigma_j^2 k^2} e^{-2R_j/\lambda(k)} \sin[2kR_j + \delta_j(k)] \quad (\text{S3})$$

where k denotes the photoelectron wave vector, N_j is the coordination number for scattering path, S_0^2 represents the amplitude reduction factor, R_j is the average absorber-scatterer distance, σ_j^2 is the Debye-Waller factor quantifying atomic disorder, $F_j(k)$ stands for the k -dependent scattering amplitude, $\lambda(k)$ describes the photoelectron mean free path for inelastic losses, $\delta_j(k)$ is the phase shift function, The index j labels distinct scattering paths. The functions $F_j(k)$, λ and $\delta_j(k)$ were calculated with the ab initio code in Artemis software. The additional details for EXAFS simulations are given below.

By fixing the CN of oxide samples (In_2O_3 and Bi_2O_3) as the known crystallographic value, the S_0^2 were fixed to 0.89 (In K-edge) and 0.64 (Bi L-edge), respectively^{14,15,16}. The definitions of other fitting parameters are as follows:

CN: coordination numbers; R: bond distance; σ^2 : Debye-Waller factors; ΔE_0 : the inner potential correction; R factor: goodness of fit.

Fitting ranges: $2.0 \leq k (\text{\AA}^{-1}) \leq 12.0$, $1.0 \leq R (\text{\AA}) \leq 2.3$ (In_2O_3);

$3.0 \leq k (\text{\AA}^{-1}) \leq 12.0$, $1.0 \leq R (\text{\AA}) \leq 2.3$ (In/BiVO₄);

$2.0 \leq k (\text{\AA}^{-1}) \leq 12.0$, $1.1 \leq R (\text{\AA}) \leq 2.1$ (Bi_2O_3);

$3.0 \leq k (\text{\AA}^{-1}) \leq 12.0$, $1.0 \leq R (\text{\AA}) \leq 2.2$ (BiVO₄).

Comment 2: The EXAFS analysis section requires substantial clarification. First, the bond distance resolution of the EXAFS experiments is not specified. Additionally, the absence of k-space data prevents the reviewer from evaluating the quality of the raw EXAFS signal and the reliability of the Fourier-transformed results. The manuscript does not mention the specific EXAFS fitting equation used, nor does it detail which parameters were fitted versus those that were fixed during the modeling process. References for the software being used are also needed in the experimental section. As this is an experimental paper, those details are crucial and this is therefore a severe shortcoming. In Table S8(Page 29), several parameters are elaborated but none of them are explained. It is also not clear to which reviewer how the coordination numbers were

determined. Was a range of coordination numbers fitted and how did the reduced chi square values change?

Response: We appreciate the valuable advice provided and completely agree with your comments. And we sincerely apologize for the lack of substantial clarification in our EXAFS analysis section. As you pointed out, more detailed explanations of the EXAFS section can significantly enhance the scientific rigor and accuracy of our manuscript. By your suggestion, we have revised and supplemented the relevant information, as detailed below :

1. Supplementation of the bond distance resolution

As you mentioned, the bond distance resolution is crucial for EXAFS data. In response, we have supplemented the bond distance resolution data in the Supplementary Information (discussion below **Supplementary Table 10**). In our EXAFS analysis, the Maximum wavevector (k_{\max}) value used is 12.0 \AA^{-1} for all samples. Based on the bond distance resolution equation (Equation R1)¹:

$$\Delta R = \frac{\pi}{2k_{\max}} \quad (\text{R1})$$

the minimum bond distance (ΔR) resolution is estimated to be approximately 0.13 \AA . This resolution refers to the smallest difference in bond distances that can be reliably distinguished between two different atomic coordination shells.

In our data, for the BiVO_4 sample, we observe two distinct Bi-O bond distances of 2.199 \AA and 2.525 \AA , with a difference of 0.326 \AA , which is significantly larger than the minimum resolvable difference of 0.13 \AA . This indicates that our EXAFS data can accurately distinguish the difference between these two bond distances. For the other three samples, where only a single coordination shell is fitted in the EXAFS analysis, the data are still accurate and reliable, meeting the requirements for structural characterization.

Changes to the revised manuscript are shown below.

Supplementary Information:

Page 46 (discussion below Supplementary Table 10): Fitting ranges: $2.0 \leq k (\text{\AA}^{-1}) \leq 12.0$, $1.0 \leq R$

(\AA) ≤ 2.3 (In_2O_3);

$3.0 \leq k (\text{\AA}^{-1}) \leq 12.0$, $1.0 \leq R (\text{\AA}) \leq 2.3$ (In/BiVO_4);

$2.0 \leq k (\text{\AA}^{-1}) \leq 12.0$, $1.1 \leq R (\text{\AA}) \leq 2.1$ (Bi_2O_3);

$3.0 \leq k (\text{\AA}^{-1}) \leq 12.0$, $1.0 \leq R (\text{\AA}) \leq 2.2$ (BiVO_4).

2. Supplementation of k-space data

Following your suggestion, the k-space data for both the In K-edge and Bi L-edge have been added to the Supplementary Information. (**Supplementary Fig. 16c** and **Supplementary Fig. 17c**) These K-space data demonstrate the reliability of our EXAFS signals and their corresponding Fourier transform results.

Changes to the revised manuscript are shown below.

Supplementary Information:

Supplementary Fig. 16. In K-edge EXAFS data of In foil, In_2O_3 and In/BiVO_4

c k^3 -weighted EXAFS signal in k-space for In K-edge.

Supplementary Fig. 17. Bi L-edge XAFS data of Bi foil, Bi₂O₃ and BiVO₄
c k^3 -weighted EXAFS signal in k -space for Bi L-edge.

3. Supplementation of specific EXAFS fitting equation and parameters

We sincerely thank you for your professional and constructive feedback. We apologize for the absence of key equations and parameter explanations in the manuscript. To address this, we have added the specific EXAFS fitting equations and relevant parameters in the Supplementary Information, along with a clear explanation of the revisions made (discussion below **Supplementary Table 10**).

Changes to the revised manuscript are shown below.

Supplementary Information:

Page 46 (discussion below Supplementary Table 10): The obtained XAFS data was processed in Athena for background, pre-edge line, and post-edge line calibrations¹³. The EXAFS data were processed according to the standard procedures using Artemis software¹³. To obtain the quantitative structural parameters around central atoms, least-squares curve parameter fitting was performed. The following EXAFS equation was used¹⁰:

$$\chi(k) = \sum_j \frac{N_j S_0^2 F_j(k)}{k R_j^2} e^{-2\sigma_j^2 k^2} e^{-2R_j/\lambda(k)} \sin[2kR_j + \delta_j(k)] \quad (\text{S3})$$

where k denotes the photoelectron wave vector, N_j is the coordination number for scattering path, S_0^2 represents the amplitude reduction factor, R_j is the average absorber-scatterer distance, σ_j^2 is the Debye-Waller factor quantifying atomic disorder, $F_j(k)$ stands for the k -dependent scattering amplitude, $\lambda(k)$ describes the photoelectron mean free path for inelastic losses, $\delta_j(k)$ is the phase

shift function, The index j labels distinct scattering paths. The functions $F_j(k)$, λ and $\delta_j(k)$ were calculated with the ab initio code in Artemis software. The additional details for EXAFS simulations are given below.

By fixing the CN of oxide samples (In_2O_3 and Bi_2O_3) as the known crystallographic value, the S_0^2 were fixed to 0.89 (In K-edge) and 0.64 (Bi L-edge), respectively^{14, 15, 16}. The definitions of other fitting parameters are as follows:

CN: coordination numbers; R: bond distance; σ^2 : Debye-Waller factors; ΔE_0 : the inner potential correction; R factor: goodness of fit.

4. Supplementation of references for the software being used

Based on your valuable suggestion, we have supplemented the information regarding the software used for XAFS and EXAFS data processing. (**Material Characterization** in manuscript and discussion below **Supplementary Table 10**) Specifically, we performed background, pre-edge line, and post-edge line calibrations on the XAFS data using the Athena software². Subsequently, the EXAFS data were analyzed using Artemis². The corresponding descriptions and references have been added to both the manuscript and the Supplementary Information.

Changes to the revised manuscript are shown below.

Main Manuscript (Methods-Material Characterization):

Page 20: All XAFS datas were processed and analyzed using Athena and Artemis software⁴⁴.

Supplementary Information:

Page 46 (discussion below Supplementary Table 10): The obtained XAFS data was processed in Athena for background, pre-edge line, and post-edge line calibrations¹¹³. The EXAFS data were processed according to the standard procedures using Artemis software¹³.

5. Explanation for parameters in Supplementary Table 10.

We sincerely thank you for your suggestion. In response, we have added further explanations for the parameters in Supplementary Information (discussion below **Supplementary Table 10**).

Changes to the revised manuscript are shown below.

Supplementary Information:

Page 46 (discussion below Supplementary Table 10): The obtained XAFS data was processed in Athena for background, pre-edge line, and post-edge line calibrations¹³. The EXAFS data were processed according to the standard procedures using Artemis software¹³. To obtain the quantitative structural parameters around central atoms, least-squares curve parameter fitting was performed. The following EXAFS equation was used¹⁰:

$$\chi(k) = \sum_j \frac{N_j S_0^2 F_j(k)}{k R_j^2} e^{-2\sigma_j^2 k^2} e^{-2R_j/\lambda(k)} \sin[2kR_j + \delta_j(k)] \quad (\text{S3})$$

where k denotes the photoelectron wave vector, N_j is the coordination number for scattering path, S_0^2 represents the amplitude reduction factor, R_j is the average absorber-scatterer distance, σ_j^2 is the Debye-Waller factor quantifying atomic disorder, $F_j(k)$ stands for the k -dependent scattering amplitude, $\lambda(k)$ describes the photoelectron mean free path for inelastic losses, $\delta_j(k)$ is the phase shift function, The index j labels distinct scattering paths. The functions $F_j(k)$, λ and $\delta_j(k)$ were calculated with the ab initio code in Artemis software. The additional details for EXAFS simulations are given below.

By fixing the CN of oxide samples (In_2O_3 and Bi_2O_3) as the known crystallographic value, the S_0^2 were fixed to 0.89 (In K-edge) and 0.64 (Bi L-edge), respectively^{14, 15, 16}. The definitions of other fitting parameters are as follows:

CN: coordination numbers; R: bond distance; σ^2 : Debye-Waller factors; ΔE_0 : the inner potential correction; R factor: goodness of fit.

6. Determination of coordination number

We sincerely appreciate your valuable suggestion. As you pointed out, the determination of coordination numbers is a crucial part of the entire EXAFS fitting process. Therefore, we have added explanations for multiple parameters, including the coordination number (discussion below **Supplementary Table 10**). Additionally, it is necessary for us to describe the EXAFS fitting process in more detail here. Specifically, the S_0^2 parameter was first determined by fitting the corresponding oxide samples with known crystallographic values, which is essential for obtaining accurate coordination numbers. Subsequently, with S_0^2 fixed, we performed a simultaneous fitting of the other four variables. Based on this fitting approach, we obtained more accurate values for key parameters such as coordination number and bond length, which were then used in our subsequent analysis.

Changes to the revised manuscript are shown below.

Supplementary Information:

Page 46 (discussion below Supplementary Table 10): By fixing the CN of oxide samples (In_2O_3 and Bi_2O_3) as the known crystallographic value, the S_0^2 were fixed to 0.89 (In K-edge) and 0.64 (Bi L-edge), respectively^{14, 15, 16}. The definitions of other fitting parameters are as follows:

CN: coordination numbers; R: bond distance; σ^2 : Debye-Waller factors; ΔE_0 : the inner potential correction; R factor: goodness of fit.

In summary, through the additional explanations provided in the above six points, we have comprehensively improved the parameters and analysis in the EXAFS section. We believe that these supplements will enhance the credibility and scientific rigor of our subsequent analysis. We look forward to your further guidance and once again sincerely thank you for your valuable suggestions.

Comment 3: Line 189-190, EXAFS section mentions that the coordination number of In-O in the first coordination sphere of In/BiVO₄ 8.24 close to that of BiVO₄ but no comparison is shown. The EXAFS of In/BiVO₄ has to be superimposed with that of BiVO₄ for comparisons when discussing this section.

Response: We sincerely thank you for your detailed comments, which are crucial for us in determining the In-O coordination number and confirming the formation of the InO₈ site. Following your comments, we have supplemented the XAFS data of BiVO₄ in the Supplementary Information and conducted the corresponding analysis (**Supplementary Fig. 17** and **Supplementary Table 10**). Furthermore, we overlaid and compared the EXAFS data of In/BiVO₄ with that of BiVO₄. From the Fourier-transformed EXAFS spectrum of BiVO₄, a spectral peak at the first coordination shell can be observed, which is similar to that in Bi₂O₃. This suggests that a Bi-O coordination likely exists in the first shell of BiVO₄. Additionally, from the wavelet-transformed plots, a similar Bi-O bonding feature as in Bi₂O₃ is also identifiable. Subsequently, we performed a fitting analysis on the EXAFS data of BiVO₄ to further reveal the Bi-O CN. The results indicate that the Bi-O CN in BiVO₄ is approximately 7.4, confirming the presence of a BiO₈ site, which is consistent with previously reported results³. By comparing the fitted EXAFS data of In/BiVO₄ with that of BiVO₄, we observed that the In-O CN (7.72) is significantly different from that in In₂O₃ but is closer to the Bi-O CN. This suggests that the In species in In/BiVO₄ exists in a form similar to that of the Bi species in BiVO₄, with a comparable coordination environment (InO₈ site). This phenomenon indicates that our LPCE strategy enables In³⁺ to selectively substitute for Bi³⁺, resulting in the formation of a similar InO₈ site. Furthermore, by comparing the fitted bond lengths, we found that the In-O bond length (2.159 Å) is shorter than the Bi-O bond lengths (2.199 Å and 2.525 Å), which is in agreement with our DFT calculations. This finding further confirms the formation of the InO₈ site and is consistent with our analysis on the suppression of photoexcited polarons.

Changes to the revised manuscript are shown below.

Supplementary Information:

Supplementary Fig. 17. Bi L-edge XAFS data of Bi foil, Bi_2O_3 and BiVO_4

a Bi L-edge XANES spectroscopy of Bi foil, Bi_2O_3 and BiVO_4 . **b** Fourier transformed k^3 -weighted $\chi(k)$ function of the EXAFS spectroscopy for Bi L-edge. **c** k^3 -weighted EXAFS signal in k -space for Bi L-edge. **d, e** Corresponding EXAFS fitting curve for BiVO_4 and Bi_2O_3 . **f** Fourier transformed k^3 -weighted $\chi(k)$ function of the EXAFS spectroscopy of In/BiVO_4 and BiVO_4 . **g** WT-EXANES of Bi foil, Bi_2O_3 and BiVO_4 .

Page 22: From the Fourier-transformed EXAFS spectrum of BiVO_4 , a spectral peak corresponding to the first coordination shell can be observed, which exhibits features similar to those in Bi_2O_3 , but distinct from those in Bi foil. This indicates that Bi-O coordination is likely present in the first coordination shell of BiVO_4 . Moreover, the wavelet transform plots also reveal a Bi-O bonding feature in the first shell that is analogous to that in Bi_2O_3 . Subsequently, we performed a fitting analysis on the EXAFS data of BiVO_4 to further determine the Bi-O coordination number. The

results show that the Bi–O coordination number in BiVO₄ is approximately 7.4 (Supplementary Table 10).

Supplementary Table 10. Structural parameters of In/BiVO₄ and BiVO₄ extracted from the EXAFS fitting.

Sample	shell	CN	R (Å)	σ^2	ΔE_0	R factor
In ₂ O ₃	In-O	6*	2.165	0.004	3.33	0.006
In/BiVO ₄	In-O	7.72±0.9	2.159	0.015	-5.59	0.005
Bi ₂ O ₃	Bi-O	3*	2.133	0.005	-5.81	0.011
BiVO ₄	Bi-O	5.08±1.1	2.199	0.010	-0.18	0.004
		2.32±1.1	2.525		-0.04	

Comment 4: The way in which the coordination number is determined have to be clarified. It is not clear from the Figure 3 e whether the coordination number of In-O in In₂O₃ is lower than that for BiVO₄. These 2 EXAFS spectra have to be once again superimposed for clarity.

Response: We sincerely appreciate your valuable comments. Following your suggestion, we have overlaid these two EXAFS spectra (In₂O₃ and In/BiVO₄) separately in the Supplementary Information to reveals the difference in the In-O coordination numbers between In₂O₃ and In/BiVO₄ (**Supplementary Fig. 16a** and **Supplementary Table 10**). The comparison of the EXAFS data after Fourier transformation at the In K-edge shows that there is no significant difference in bond lengths within the range of 1 Å < R < 2 Å between the two samples. However, a peak corresponding to In-In bonding can only be observed in the In₂O₃ sample. The fitted parameters in **Supplementary Table 10** further explain the difference in coordination numbers. When the coordination number of In₂O₃ is fixed at 6, the fitted coordination number for the In/BiVO₄ sample is approximately 7.72. These results strongly confirm that the In-O coordination number in In₂O₃ is significantly lower than that in In/BiVO₄, indicating distinct local coordination environments.

Changes to the revised manuscript are shown below.

Supplementary Information:

Supplementary Fig. 16. In K-edge EXAFS data of In foil, In₂O₃ and In/BiVO₄

a Fourier transformed k³-weighted $\chi(k)$ function of the EXAFS spectroscopy for In K-edge.

Supplementary Table 10. Structural parameters of In/BiVO₄ and BiVO₄ extracted from the EXAFS fitting.

Sample	shell	CN	R (Å)	σ^2	ΔE_0	R factor
In ₂ O ₃	In-O	6*	2.165	0.004	3.33	0.006
In/BiVO ₄	In-O	7.72±0.9	2.159	0.015	-5.59	0.005
Bi ₂ O ₃	Bi-O	3*	2.133	0.005	-5.81	0.011
BiVO ₄	Bi-O	5.08±1.1	2.199	0.010	-0.18	0.004
		2.32±1.1	2.525		-0.04	

Comment 5: It is not clear to this reviewer how the WT simulations were conducted?

Response: We sincerely thank you for pointing out this important issue. In the manuscript, our description of the wavelet transform was indeed insufficient. In response, we have added a more detailed explanation of the principles and parameters used in our wavelet transform analysis (discussion below **Supplementary Fig. 17**). We believe that the addition of more detailed wavelet transform parameters will enhance the scientific rigor of our manuscript. Once again, we sincerely thank you for your professional insights.

Changes to the revised manuscript are shown below.

Supplementary Information:

Page 21-22 (discussion below Supplementary Figure 17): Wavelet transform analysis was performed using the commercial open-source software HAMA. The Morlet wavelet was selected for its capability to discriminate contributions from atoms at similar interatomic distances by resolving varying amplitudes and oscillating phase^{9,10}. This approach leverages the two-dimensional distribution of EXAFS signals in k-space and R-space, enabling effective separation of contributions from multiple scattering paths. The equation of Morlet wavelet is shown in equation 1⁹:

$$\psi(t) = \frac{1}{\sqrt{2\pi\sigma}} \left[e^{ikt} - e^{-\frac{k^2}{2}} \right] e^{-\frac{t^2}{2\sigma^2}} \quad (1)$$

Wave number k was set to 10, corresponding to the frequency of sine/cosine basis functions. This free parameter determines the number of wave oscillations within the Gaussian envelope. Scale parameter σ was set to 1, defining the half-width of the Gaussian envelope that localizes the wavelet in time-frequency space.

Comment 6: It is not clear how the Bader charge analysis was carried out. The details in this section are quite limited.

Response: We sincerely thank you for highlighting this important issue. Indeed, the description of the DFT calculations in the manuscript was not sufficiently detailed, and we sincerely apologize for this. To improve and complete the section on DFT calculation parameters, we have added detailed information covering charge density difference and Bader charge calculations (discussion below **Supplementary Fig. 5**). We believe these additions will enhance the authenticity and accuracy of our results. We once again thank you for your thoughtful comments.

Changes to the revised manuscript are shown below.

Supplementary Information:

1. Charge density difference details

Page 9 (discussion below Supplementary Figure 5): The differential charge is obtained by Equation 1:

$$\Delta\rho = \rho_{AB} - \rho_A - \rho_B \quad (1)$$

where ρ_{AB} represents the charge density of the system, ρ_A and ρ_B represent the charge density of a certain part, respectively. Specifically, in the model, ρ_A represents the charge density of the selected Bi or In atoms, and ρ_B represents the charge density of all the remaining atoms.

2. Bader charge analysis details

Page 9 (discussion below Supplementary Figure 5): Bader charge analysis was performed using the VASP built-in module and the Bader charge analysis program developed by Graeme Henkelman, et al³. The implementation procedure consisted of the following steps:

1. Core and valence electron densities were extracted from VASP output charge files (CHGCAR).
2. Bader charge partitioning was performed using the Bader charge analysis program. The Bader charges were specifically quantified for surface Bi, V, and In atoms in the modeled structure.

Comment 7: The section on the EPR measurements is unclear. Line 301-202 mentions appearance of new peaks at $g = 1.96$ and 2.039 for BiVO_4 , however several other peaks are shown which are not attributed, fitted, or explained.

Response: Thank you for raising this important issue. As you mentioned, our interpretation and analysis of the EPR spectra were indeed insufficient, and we sincerely apologize for this. Following your suggestion, we have carefully re-examined the EPR spectra and provided further explanations for the observed peaks in light of the literature (**Suppression of HPs and Carrier Property Analysis** in manuscript). The detailed analysis is as follows:

First, by comparing the EPR spectra under dark conditions, a peak at $g = 2.003$ is observed in both samples, which can be attributed to free electrons⁴. Under illumination, the BiVO_4 sample exhibits distinct EPR peaks at $g = 1.945$, $g = 1.960$, and $g = 2.039$. Specifically, the peaks at $g = 1.945$ and $g = 1.960$ can be attributed to the self-trapping

of photo-generated electrons, whereas the signal at $g = 2.039$ is assigned to the self-trapping of photo-generated holes^{4, 5, 6, 7, 8, 9}. In contrast, no significant new EPR peaks are observed in the In/BiVO₄ photoanode, indicating that the formation of polarons under illumination is effectively suppressed in this sample. It is worth noting that the weak and broad peak at $g = 1.955$ is a small background signal. Combining the results of EPR, Td-PL, and SPV, we can confirm that indium substitution effectively suppresses the formation of hole polarons, thereby releasing more hole charge carriers. We believe that this improved interpretation of the EPR data will help to further strengthen the manuscript. Once again, we sincerely thank you for your valuable comments.

Changes to the revised manuscript are shown below.

Main Manuscript (Results-Suppression of HPs and Carrier Property Analysis):

Page 14: To elucidate the relationship between the enhancement of PEC activity and the suppression of HPs, EPR spectroscopy was performed. Under dark conditions, both samples exhibited an EPR signal at $g = 2.003$, which is attributed to free electrons (Supplementary Fig. 27a)³⁴. Under illumination, additional EPR signals were observed in the BiVO₄ sample. Specifically, the peaks at $g = 1.945$ and $g = 1.960$ are attributed to the self-trapping of photo-generated electrons, whereas the signal at $g = 2.039$ is assigned to the self-trapping of photo-generated holes^{34,35,36}. These spectral features serve as clear evidence of the formation of electron and HPs, respectively. In contrast, the EPR signal for In/BiVO₄ was negligible, indicating that the generation of HPs under illumination was effectively suppressed in In/BiVO₄.

Comment 8: The material section only mentions the user of a Bruker Instrument for the EPR measurements without any further details. What microwave frequency was used? At what temperature were the measurements conducted? What modulation frequency? What power?

Response: We sincerely appreciate you pointing out this key issue and apologize for the insufficient description of the EPR instrument parameters in the original manuscript. Following your suggestion, we have added the detailed EPR measurement parameters in manuscript (please refer to the revised content below and **Material characterization** in manuscript). These supplementary details help enhance the transparency and reproducibility of our experimental methods. We are grateful for your valuable comments.

Changes to the revised manuscript are shown below.

Main Manuscript (Methods-Material Characterization):

Page 20: Electron paramagnetic resonance (EPR) measurements were conducted using a Bruker EMXplus-6/1 spectrometer (Germany) utilizing a microwave frequency of 9.84 GHz with an incident power of 6.325 mW. Measurements were conducted under temperature-controlled conditions set to 295.00 K. Field modulation parameters included a frequency of 100.00 kHz with 4.000 G amplitude, employing first-derivative detection and 0.01 ms time constant.

Responses to Reviewer #2:

In order to improve photoelectrochemical (PEC) water-splitting performance, the manuscript presents a surface-engineering approach for BiVO₄ photoanodes using In³⁺ substitution to suppress hole polaron formation. The results are generally convincing, the work is well-written, and it is technically sound. To improve the manuscript, though, a few queries and explanations are required, especially regarding the mechanism of hole polaron suppression and pec performance as follows

Authors: We sincerely appreciate your recognition of our work, as well as the decision and constructive comments you have provided on our manuscript. We agree with the reviewer's suggestion to include a more detailed explanation of the mechanism for suppressing hole polaron formation and its relationship to the enhancement of PEC

performance. In response, we have supplemented the manuscript and Supplementary Information with a variety of characterizations and tests, as well as DFT calculations. We have also discussed in depth, in combination with previously reported work, the relationship between polarons and PEC performance. We believe that these additions have enhanced the overall scientific rigor of the manuscript. Thank you once again for your valuable suggestions. Below, we discuss the specific suggestions you raised and provide point-by-point explanations.

Comment 1: In the DFT section the author claims that the bond length changes for the Bi-O while it remains same for the In-O case. What is the correlation of this physical phenomenon with the polaron formation. It needs more explanation for better understanding.

Response: We sincerely appreciate your valuable comments. We apologize for the limited explanation in our manuscript regarding the correlation between the metal-oxygen (M-O) bond length at polaron sites and polaron formation. To better clarify the physical relationship between bond length and polaron formation, we have added supplementary DFT calculations and discussed the connection between EPs and HPs and the bond lengths at their respective localization sites (**Fig. 2f** and **Supplementary Fig. 3**). Additionally, we compared the changes in bond lengths before and after light excitation to demonstrate the suppression of HP formation. In general, in metal oxides, shorter M-O bond lengths at polaron sites are less favorable for polaron formation. This is because the self-trapping of excess electrons (holes) to form EPs (HPs) typically accompanies local lattice expansion (contraction). In such cases, shorter M-O bonds indicate stronger bond energy and are less likely to be stretched or compressed, which makes polaron formation energetically less favorable. The specific discussion based on our DFT results is as follows:

We have added bond length change data for BiVO₄-DFT and In/BiVO₄-DFT with EP in **Supplementary Fig. 3**. It is evident that the M-O bonds at the localization sites elongate during EP formation and shorten during HP formation. This observation

highlights the localization of polarons at specific sites and confirms that our model accurately reflects the behavior of polaron localization. Notably, for EP formation, both BiVO₄-DFT and In/BiVO₄-DFT models show a V-O bond length increase of about 0.1 Å after excitation, indicating that In³⁺ substitution has little effect on EP formation. However, for HP formation, the In-O bond length at the substituted site remains almost unchanged, while the original Bi-O bond shortens by approximately 0.05 Å. This suggests that In³⁺ substitution at the Bi site effectively suppresses HP formation, which is consistent with our findings on the formation energy. To more clearly reveal the local bond length changes at polaron sites, we presented the local structures and M-O bond lengths for both models with EP and HP. As shown in **Supplementary Fig. 3b**, the formation of EP is consistently accompanied by an increase in V-O bond length of about 0.1 Å. In contrast, HP formation at the BiO₈ site leads to a reduction in Bi-O bond length of about 0.05 Å, while the In-O bond at the InO₈ site remains nearly unchanged. Combining the bond length statistics with the local structural analysis, we clearly described the changes in M-O bond lengths associated with EP and HP formation and demonstrated that In substitution at the Bi site helps to suppress HP localization. Finally, we summarized the bond length changes associated with polaron formation in a schematic diagram shown in **Supplementary Fig. 3c**. It is worth noting that the relationship between polaron formation and bond length changes has been reported in the literature. For example, Wiktor et al. observed that EPs localize around vanadium atoms, causing the V-O bond length to increase from about 1.72 Å to 1.81 Å, while HPs distribute among a bismuth atom and eight oxygen atoms, leading to a shortening of the Bi-O bond from 2.48 Å to 2.42 Å⁷. Similarly, Liu et al. pointed out that when HPs localize in the BiO₈ unit, the Bi-O bond length decreases by approximately 0.03 Å. These findings are in good agreement with our conclusions¹⁰.

We hope that the additional DFT calculations and the literature-supported analysis provide a more detailed explanation of the relationship between bond length changes and polaron formation.

Changes to the revised manuscript are shown below.

Main Manuscript (Results-DFT-guided Strategies for the Suppression of HPs):

Fig. 2. DFT calculations analyze the suppressive effect of In substitution

f Bi-O radial distribution functions in BiVO₄-DFT and In-O radial distribution functions in In/BiVO₄-DFT with and without HP.

Page 5-6: To further confirm the suppression of polaron formation, we compared the bond lengths of the In/BiVO₄-DFT and BiVO₄-DFT models before and after photoexcitation (Fig. 2f and Supplementary Fig. 3). It was observed that the Bi-O bond length significantly decreases from 2.45 Å to 2.40 Å, whereas the In-O bond length remains at 2.25 Å, further supporting the notion that polarons are less likely to form at InO₈ sites²⁵. Correspondingly, we also investigated the effect of In substitution on EP behavior. Differential charge density, Bader charge analysis, E_p(EP), DOS, and bond length variation data collectively indicate that In substitution has negligible impact on the properties of Eps (Supplementary Fig. 2, 3 and Supplementary Table 1).

Supplementary Information:

Supplementary Fig. 3. Relationship between polarons and bond length changes.

a V-O radial distribution functions and **b** variations in bond length within the local model in BiVO₄-DFT and In/BiVO₄-DFT with and without the hole polaron. **c** Schematic illustration of polaron formation.

Page 7: As shown in Supplementary Fig. 3 a, the V-O bond length is observed to increase after photoexcitation. In combination with Figure 2 f from the manuscript, it can be seen that the substitution of In significantly suppresses the shortening of the In-O bond under photoexcitation, but has little effect on the elongation of the V-O bond. A comparison of the models before and after photoexcitation reveals that in both BiVO₄-DFT and In/BiVO₄-DFT models, the V-O bond length increases by approximately 0.1 Å after photoexcitation. This can be attributed to the local lattice expansion induced by the formation of the electron polaron (EP). In contrast, the hole polaron (HP) site exhibits different behavior. In BiVO₄-DFT model, the Bi-O bond length decreases by about 0.05 Å after photoexcitation, while in In/BiVO₄-DFT model, the In-O bond shows almost no shortening. This indicates that the formation of the In-O site effectively suppresses the generation of hole polarons. Schematic illustration of polaron formation demonstrate the relationship between HP and EP formation and bond length changes. Specifically, the formation of EP requires local lattice expansion, while the formation of HP is associated with local lattice contraction.

Comment 2: In the XPS analysis Shifts in Bi 4f and V 2p can also be influenced by local strain and oxygen vacancies?

Response: We sincerely appreciate the insightful and well-considered comments provided by the reviewers. Your thoughtful attention has significantly contributed to the improvement of our work. In response to your suggestions, we conducted additional tests and analyses to investigate whether local strain and oxygen vacancies could cause the shifts in the XPS spectra (**Supplementary Fig. 15**). The results indicate that the influence of these two factors is relatively minor, and the spectral shifts are primarily attributed to the electronic transfer induced by the formation of InO₈ sites after LPCE process. The specific results are presented as follows:

1. The influence of local strain

To evaluate whether our LPCE process induces local strain that could lead to shifts in the XPS spectral peaks, we supplemented TEM measurements for In/BiVO₄ and BiVO₄ photoanodes and performed geometric phase analysis (GPA). As shown in **Supplementary Fig. 15a and b**, no significant strain is observed in either the xx or yy directions for these two samples. This indicates that LPCE process does not introduce excessive local strain, suggesting that the shifts in the Bi 4f and V 2p peaks are not caused by local strain. Instead, they are more likely attributed to the formation of InO₈ sites, which facilitates electronic transfer from BiO₈ and VO₄ sites to InO₈ sites.

2. The influence of oxygen vacancies

After ruling out the influence of local strain through GPA, we further investigated whether oxygen vacancies could affect the XPS spectral shifts. Our previous XPS data demonstrated that the proportion of oxygen vacancies in In/BiVO₄ did not exhibit significant changes after the LPCE process (**Supplementary Table 6**). It is important to note that due to a typographical error, the labeling of the O_v and O_c tables was accidentally reversed in the earlier version. We sincerely apologize for this mistake. However, this did not affect our analysis or conclusions of the O 1s XPS spectrum, and the error has been corrected in the revised manuscript.

Moreover, to further confirm that our LPCE method does not introduce a large number of oxygen vacancies into the material, we conducted additional XPS tests. Using a similar solvent immersion method (SIM, using the same amount of ethylene glycol and choline chloride, but without the addition of $\text{In}(\text{NO}_3)_3$), we prepared a control photoanode that was immersed only, named SIM/ BiVO_4 , and performed XPS measurements on it. As shown in **Supplementary Fig.15c and Supplementary Table 6**, the proportion of oxygen vacancies in SIM/ BiVO_4 remains largely unchanged compared to BiVO_4 , indicating that our LPCE process is unlikely to introduce significant oxygen vacancies. Furthermore, the comparison of the Bi 4f and V 2p spectra in **Supplementary Fig.15d and e** reveals that the SIM itself does not induce shifts in the XPS spectral peaks.

In summary, the additional GPA and XPS analyses confirm that the LPCE process does not introduce substantial local strain or oxygen vacancies. Therefore, the observed XPS spectral shifts are not attributable to these factors, but are instead more likely caused by the electronic transfer associated with the formation of InO_8 sites. This conclusion is further supported by our XAFS and DFT results. Based on your professional suggestions, we have added an analysis of local strain and oxygen vacancies, which provides a more accurate interpretation of the XPS spectral shifts and further supports our conclusions. We sincerely appreciate your valuable insights.

Changes to the revised manuscript are shown below.

Main Manuscript (Results-Synthesis and Structural Characterization):

Page 8: Moreover, the potential effects induced by local strain and oxygen vacancies were systematically discussed and excluded (Supplementary Fig. 15).

Supplementary Information:

Supplementary Fig. 15. XPS spectrum analysis.

TEM image and GPA strain map along ϵ_{xx} , ϵ_{yy} of **a** In/BiVO₄ and **b** BiVO₄ photoanode. XPS high-resolution spectra of **c** O 1s, **d** Bi 4f and **e** V 2p in In/BiVO₄ photoanode.

Page 15: The GPA results indicate that no significant strain is observed in either the xx or yy directions for both In/BiVO₄ and BiVO₄ photoanodes, suggesting that the LPCE process introduces minimal local strain. To further investigate this, we prepared a control photoanode using a similar solvent immersion method (SIM, with the same amount of ethylene glycol and choline chloride but without the addition of In(NO₃)₃), and named it SIM/BiVO₄. By comparing the high-resolution XPS spectra of SIM/BiVO₄ and BiVO₄, and in conjunction with Supplementary Table 6, it can be observed that the ratio of oxygen vacancies in SIM/BiVO₄ remains nearly unchanged compared to BiVO₄. Moreover, no significant shifts are observed in the Bi 4f and V 2p spectra. These results demonstrate that our LPCE treatment does not introduce a large amount of local strain or oxygen vacancies, and that the solvent immersion method does not cause significant shifts in the XPS spectral peaks. Combined with XAFS and DFT data, it is further confirmed that the observed shifts in the XPS peaks are primarily attributed to the formation of InO₈ sites.

Supplementary Table 6. Atomic ratios of O_L , O_V and O_C calculated from the XPS spectrum $BiVO_4$ and $In/BiVO_4$.

Sample	O_L (at. %)	O_V (at. %)	O_C (at. %)
$BiVO_4$	67.4	10.9	21.7
$In/BiVO_4$	65.4	13.4	21.2
$SIM/BiVO_4$	67.5	13.4	19.1

Comment 3: Can the author provide the depth profiling with detailed discussion?

Response: We sincerely appreciate your insightful comments, which are of great significance for the in-depth interpretation of the depth-profile analysis in our study. As you pointed out, the explanation and analysis of the depth profile figures in our manuscript were insufficient and unclear, and we sincerely apologize for that. In response to your suggestion, we have added detailed depth-etching XPS (etch-XPS) high-resolution spectra of $In/BiVO_4$, specifically including the Bi 4f, V 2p, and In 3d spectra (**Supplementary Fig. 14d-f**). Our analysis of these spectra reveals that, with increasing etching depth, the binding energies of the Bi and V spectral peaks shift toward lower values, whereas the In 3d spectral peaks exhibit a shift toward higher binding energies. These observations provide further experimental evidence supporting our conclusion that electrons are transferred from the Bi and V sites to the In sites. This finding is in good agreement with the results obtained from DFT calculations and XAFS analysis. The detailed discussion and analysis are as follows.

First, we have supplemented and analyzed the high-resolution etch-XPS spectra of $In/BiVO_4$, aiming to offer a more thorough interpretation of the depth-profile data. According to the elemental content analysis presented in manuscript, the In content decreases with increasing etching depth, while the Bi content increases markedly and the V content increases slightly (**Fig. 3** in manuscript). These trends confirm the selective substitution of In for Bi. Furthermore, we have conducted a systematic investigation of the spectral shifts. In the original XPS data, we observed that the

binding energies of the Bi and V spectra shifted to higher values following In substitution, indicating electron depletion at the Bi and V sites and potentially facilitating electron accumulation at the In sites (**Fig. 3b** in manuscript). The etch-XPS results offer even more compelling evidence for this electron transfer mechanism. From the Bi and V XPS spectra, it can be observed that with increasing etching depth, the binding energies of the corresponding peaks shift toward lower binding energy. This suggests that with the increase in In content, the Bi and V sites experience a greater loss of electrons, and these electrons are likely to be transferred to the In sites. This interpretation is further corroborated by the analysis of the In 3d high-resolution etch-XPS spectra. As shown in **Supplementary Fig. 14f**, the In 3d_{3/2} and In 3d_{5/2} peaks shift toward higher binding energies as the etching depth increases, which is consistent with the accumulation of electrons at the In site. In contrast, the Bi 4d_{5/2} peak shifts toward lower binding energies, indicating an electron loss at the Bi site. The opposite trends in binding energy shifts between In and Bi provide clear evidence for the electron transfer from Bi and V to In induced by In substitution. This electron transfer conclusion is consistent with the results from XAFS and DFT calculations, and it is beneficial for suppressing hole localization during the PEC process. We are confident that the additional analysis of the etch-XPS data enhances the experimental support for the proposed electron transfer mechanism and significantly improves the scientific rigor and clarity of our manuscript. We are grateful for your constructive feedback and have incorporated the suggested revisions as detailed below.

Changes to the revised manuscript are shown below.

Main Manuscript (Results-Synthesis and Structural Characterization):

Fig. 3: Material characterization.

b XPS high-resolution spectroscopy of Bi 4f and V 2p in BiVO₄ and In/BiVO₄ photoanodes. **c** etch-XPS spectroscopy of In/BiVO₄ photoanode.

Page 8: Further analysis of elemental composition changes and electron transfer was conducted using etching combined with XPS (etch-XPS). The results indicate that as the etching depth increases, the In content decreases while the Bi content significantly increases, with only a moderate increase in V content, suggesting that In primarily substitutes Bi sites (Fig. 3c). Additionally, the binding energy shifts further confirm the electron transfer from Bi and V sites to In sites (Supplementary Fig. 14d-f).

Supplementary Information:

Supplementary Fig. 14. Etch-XPS high-resolution spectra of **a** Bi 4f, **b** V 2p and **c** In 3d in In/BiVO₄ photoanode.

Page 18: With increasing etching depth, the peaks in the etch-XPS high-resolution spectra of Bi and V both shift toward lower binding energies, indicating that the Bi and V sites lose more electrons as the In content increases. In the etch-XPS high-resolution spectra of In 3d, the peaks corresponding to In 3d_{3/2} and In 3d_{5/2} shift toward higher binding energies, while the peak corresponding to Bi 4d_{5/2} shifts toward lower binding energies. This suggests that, with the increase in In content, the In sites gain more electrons. The results of these three spectra collectively indicate that the substitution of In facilitates the transfer of electrons from the Bi and V sites to the In sites.

Comment 4: XANES analysis need more explanation.

Response: We sincerely appreciate your thoughtful comments. As both you and the first reviewer pointed out, the XAFS analysis in the manuscript was not sufficiently detailed. We apologize for this oversight and have therefore thoroughly revised and expanded the XAFS analysis, adding comprehensive parameter information and including the XAFS data of BiVO₄ for comparison (**Fig. 3d-g** in manuscript, **Supplementary Fig. 17** and **Supplementary Table 10**). We believe the XAFS analysis section has been significantly enhanced based on these additions. Overall, our XAFS data indicate that the CN of Bi-O in BiVO₄ is close to 8, suggesting the presence of BiO₈ sites, which is consistent with previous reports in the literature^{3, 11, 12}. After LPCE, the In-O CN in In/BiVO₄ is also found to be approximately 8, further supporting the notion that In³⁺ selectively replaces Bi³⁺, thereby forming InO₈ sites. This observation confirms the selectivity of our LPCE approach. In combination with the shift in the absorption edge and the XPS results, we have also demonstrated a reduction in the oxidation state of In species, indicating that In sites have received additional electrons. Below is a detailed analysis of the supplementary data:

First, since our LPCE method involves the selective substitution of In³⁺ for Bi³⁺ in BiVO₄, it is essential to analyze the local structure of the Bi species in the pristine BiVO₄. The newly added XAFS data of BiVO₄ show that the main spectral peak in the first coordination shell is attributed to Bi-O bonding, suggesting the presence of BiO_x sites (**Supplementary Fig. 1**). To further clarify the coordination environment of Bi, we performed EXAFS fitting, which yielded a Bi-O coordination number close to 8, suggesting the presence of BiO₈ sites, which is consistent with the literature³ (**Supplementary Table 10**). Next, we analyzed the XAFS data of In/BiVO₄ to investigate the coordination environment of In. The K-edge XANES spectrum of In/BiVO₄ closely resembles that of In₂O₃, indicating that the predominant oxidation state of In is +3. Notably, the absorption edge of In/BiVO₄ shifts toward lower binding energies compared to that of In₂O₃, in agreement with the XPS results. This shift suggests a reduction in the valence state of In, likely due to the higher electronegativity of In relative to Bi, which facilitates electron transfer to the InO₈ sites formed by In substitution. To confirm the bonding configuration resulting from In substitution, we

compared the R-space transformed EXAFS data. A prominent In-O peak appears at approximately 1.62 Å, indicating that In primarily coordinates with oxygen atoms. EXAFS fitting further reveals an In-O CN of about 7.72 in the first shell, which is close to that of Bi-O in BiVO₄ and significantly higher than that in In₂O₃. The similarity in CN suggests that InO₈ sites are formed in In/BiVO₄ after substitution, demonstrating the selectivity of our LPCE strategy.

In summary, by comparing the XAFS data of BiVO₄ and In/BiVO₄, we have confirmed the existence of BiO₈ sites and shown that In³⁺ selectively substitutes Bi³⁺ to form InO₈ sites. These InO₈ sites are expected to facilitate electron transfer from neighboring sites. We hope that the revised and more detailed XAFS analysis will provide clearer support for our conclusions and strengthen the scientific rigor and completeness of the manuscript. Thank you once again for your professional and constructive feedback.

Changes to the revised manuscript are shown below.

Main Manuscript (Results-Synthesis and Structural Characterization):

Fig. 3: Material characterization.

d In K-edge XANES spectroscopy of In foil, In₂O₃ and In/BiVO₄. **e** Fourier transformed k³-weighted χ(k) function of the EXAFS spectroscopy for In K-edge. **f** corresponding EXAFS fitting curve for In/BiVO₄. **g** WT-EXANES of In foil, In₂O₃ and In/BiVO₄.

Page 8-9: To further analyze the electronic state and coordination environment of In, X-ray absorption fine spectroscopy (XAFS) was utilized. As shown in Fig. 3d, the K-edge X-ray absorption near-edge structure (XANES) spectroscopy of In/BiVO₄ closely resemble that of In₂O₃, implying that the oxidation state of In in In/BiVO₄ is predominantly trivalent. Notably, the absorption edge of In/BiVO₄ shifts toward lower binding energies compared to that of In₂O₃, consistent with the XPS results. This shift indicates a decrease in the valence state of In species, attributed to the greater electronegativity of In compared to Bi, leading to electron transfer to the InO₈ sites formed by In substitution. Extended X-ray absorption fine structure (EXAFS) analysis was performed to elucidate the local coordination environment of Bi and In species. The coordination environment of Bi species in BiVO₄ was carefully analyzed using the Bi L-edge absorption spectrum (Supplementary Fig. 17). We found that in BiVO₄, the Bi species primarily exhibit Bi-O coordination in the first coordination shell. By fitting the EXAFS data, a coordination number of approximately 7.4 for Bi-O was obtained, indicating the presence of BiO₈ sites (Supplementary Table 10)^{30,31,32}. Fig. 3e depicts the R-space transition of the EXAFS, where the pronounced peak at 1.62 Å corresponds to In-O bonding. The fitted EXAFS results show that in In/BiVO₄, the In-O bond length in the first coordination shell is 2.159 Å, with a coordination number of about 7.72, which is significantly different from that in In₂O₃ (6). The very similar coordination numbers between the Bi species in BiVO₄ and the In species in In/BiVO₄ suggest that InO₈ sites are present in In/BiVO₄, analogous to BiO₈. This observation provides strong structural evidence supporting our strategy of LPCE selectively replacing Bi sites. Additionally, wavelet transform (WT) simulations were conducted to analyze the radial distance resolution map in K-space. As presented in Fig. 3g1-g3, the WT intensity maxima corresponding to In-O coordination near 5.0 Å⁻¹ are well-resolved between 1.0 and 2.0 Å, with no significant In-In coordination observed, indicating that In is predominantly bonded to O in the sample.

Supplementary Information:

Supplementary Fig. 17. Bi L-edge XAFS data of Bi foil, Bi_2O_3 and BiVO_4

a Bi L-edge XANES spectroscopy of Bi foil, Bi_2O_3 and BiVO_4 . **b** Fourier transformed k^3 -weighted $\chi(k)$ function of the EXAFS spectroscopy for Bi L-edge. **c** k^3 -weighted EXAFS signal in k -space for Bi L-edge. **d, e** Corresponding EXAFS fitting curve for BiVO_4 and Bi_2O_3 . **f** Fourier transformed k^3 -weighted $\chi(k)$ function of the EXAFS spectroscopy of In/BiVO_4 and BiVO_4 . **g** WT-EXANES of Bi foil, Bi_2O_3 and BiVO_4 .

Page 22: From the Fourier-transformed EXAFS spectrum of BiVO_4 , a spectral peak corresponding to the first coordination shell can be observed, which exhibits features similar to those in Bi_2O_3 , but distinct from those in Bi foil. This indicates that Bi-O coordination is likely present in the first coordination shell of BiVO_4 . Moreover, the wavelet transform plots also reveal a Bi-O bonding feature in the first shell that is analogous to that in Bi_2O_3 . Subsequently, we performed a fitting analysis on the EXAFS data of BiVO_4 to further determine the Bi-O coordination number. The results show that the Bi-O coordination number in BiVO_4 is approximately 7.4 (Supplementary Table 10).

Supplementary Table 10. Structural parameters of In/BiVO₄ and BiVO₄ extracted from the EXAFS fitting.

Sample	shell	CN	R (Å)	σ^2	ΔE_0	R factor
In ₂ O ₃	In-O	6*	2.165	0.004	3.33	0.006
In/BiVO ₄	In-O	7.72±0.9	2.159	0.015	-5.59	0.005
Bi ₂ O ₃	Bi-O	3*	2.133	0.005	-5.81	0.011
BiVO ₄	Bi-O	5.08±1.1	2.199	0.010	-0.18	0.004
		2.32±1.1	2.525		-0.04	

Comment 5: About the PEC data, how reproducible this data set? how many devices were tested? it is advised to show different devices data set also provide error bars.

Response: Thank you for your valuable comments. As you have rightly pointed out, the reproducibility of photocurrent density curves is of critical importance in PEC systems. Following your suggestion, we conducted LSV tests on 10 independent samples and provided the average curve with error bars to illustrate the variability. The results show that our data exhibit good reproducibility, with the current density stabilizing at 6.464 mA cm⁻². We have supplemented the revised **Supplementary Fig. 21** with the LSV curves obtained from the tests, as well as the curves with error bars.

Changes to the revised manuscript are shown below.

Supplementary Information:

Supplementary Fig. 21. Repeated LSV tests of photoanodes.

a Average LSV curve with associated error bars of In/BiVO₄/FeOOH photoanodes. **b** Repeated LSV

tests of In/BiVO₄/FeOOH photoanodes (Ten independent samples were tested).

Page 26: To better demonstrate the reproducibility of our PEC data, we conducted LSV tests on ten independent samples and provided the average curve with error bars. The results show that our PEC data exhibit good reproducibility, with an average current density of 6.464 mA cm⁻².

Comment 6: For the electrochemical testing did author calibrate the reference electrode? how this conversion happened to RHE. Please do the calibration and present in the revised version with detailed explanations.

Response: Thank you for your professional comments. The calibration of the reference electrode is of great importance in photoelectrochemical and electrochemical measurements. In all tests, we used an Ag/AgCl reference electrode, and converted the reference electrode potential to the Reversible Hydrogen Electrode (RHE) scale based on the Nernst equation. The specific equation is as follows:

$$E_{\text{RHE}} = E_{\text{Ag/AgCl}} + E_{\text{Ag/AgCl}}^{\ominus} + 0.059 \times \text{pH} \quad (\text{R2})$$

$$E_{\text{Ag/AgCl}}^{\ominus} = 0.1976 \text{ V}_{\text{RHE}} \quad (\text{R3})$$

In which $E_{\text{Ag/AgCl}}$ is the potential measured with respect to the reference electrode, and $E_{\text{Ag/AgCl}}^{\ominus}$ is the standard potential of the Ag/AgCl reference electrode in the solution. We have also provided detailed calibration procedures in the revised manuscript.

Changes to the revised manuscript are shown below.

Main Manuscript (Methods-PEC Measurements):

Page 21: The Ag/AgCl reference electrode was employed as the reference for all electrochemical and photoelectrochemical measurements. The measured potentials vs. Ag/AgCl were converted to the reversible hydrogen electrode (RHE) scale according to the Nernst equation:

$$E_{\text{RHE}} = E_{\text{Ag/AgCl}} + E_{\text{Ag/AgCl}}^{\ominus} + 0.059 \times \text{pH} \quad (2)$$

$$E_{\text{Ag/AgCl}}^{\ominus}=0.1976 \text{ V}_{\text{RHE}} \quad (3)$$

where $E_{\text{Ag/AgCl}}$ is the potential measured with respect to the reference electrode, and $E_{\text{Ag/AgCl}}^{\ominus}$ is the standard potential of the Ag/AgCl reference electrode in the solution.

Comment 7: Please provide the Tafel slope as well for better understanding of electrochemical performance with explanations.

Response: We sincerely thank you for your constructive comments. The Tafel slope data is crucial for a better understanding of the intrinsic electrochemical activity of the photoanode. Following your suggestion, we have supplemented the Tafel slope data for BiVO_4 , In/BiVO_4 , and $\text{In/BiVO}_4/\text{FeOOH}$, and conducted a comparative analysis (please refer to **Supplementary Fig. 29c**). As shown in the figure, the Tafel slope of the In/BiVO_4 photoanode (233 mV dec^{-1}) does not exhibit a significant decrease compared to that of BiVO_4 (260 mV dec^{-1}), indicating that the In surface substitution has a limited effect on the intrinsic electrochemical activity of the material. In contrast, the loading of FeOOH significantly reduces the Tafel slope to 76 mV dec^{-1} , demonstrating a substantial improvement in the intrinsic electrochemical activity of the photoanode. In summary, the In^{3+} surface substitution does not markedly enhance the electrochemical activity of the material. As supported by the data in the manuscript, In^{3+} primarily improves the PEC performance of the photoanode by suppressing the formation of HPs, thereby releasing more holes. On the other hand, FeOOH, due to its excellent catalytic activity, significantly enhances the intrinsic electrochemical activity and the utilization efficiency of charge carriers after being loaded as a co-catalyst. By modifying the HP sites with In^{3+} to release more holes and further improving the surface catalytic activity with the co-catalyst, we have effectively enhanced the PEC performance of the photoanode. We believe that the supplementation of the Tafel slope is of great significance for improving the completeness of the electrochemical performance explanation in our manuscript. We would like to express our sincere gratitude once again for your valuable suggestion.

Changes to the revised manuscript are shown below.

Main Manuscript (Results-Suppression of HPs and Carrier Property Analysis.):

Page 16: The LSV curves under dark conditions, electrochemical double-layer capacitance (C_{dl}), and Tafel slopes were employed to reveal the changes in the catalytic activity of the photoanodes (Supplementary Fig. 29).

Supplementary Information:

Supplementary Fig. 29. Electrochemical testing of photoanodes.

c Tafel slope curves of BiVO_4 , In/BiVO_4 and $\text{In/BiVO}_4/\text{FeOOH}$ photoanodes.

Page 35-56: The Tafel slope is used to reveal the intrinsic electrochemical activity of the photoanode. It can be observed that the Tafel slope of the In/BiVO_4 photoanode (233 mV dec^{-1}) does not show a significant decrease compared to BiVO_4 (260 mV dec^{-1}), indicating that the In^{3+} surface substitution has limited effect on improving the electrochemical activity. In contrast, the Tafel slope of $\text{In/BiVO}_4/\text{FeOOH}$ is reduced to 76 mV dec^{-1} , demonstrating that FeOOH significantly enhances the intrinsic electrochemical activity of the photoanode.

Comment 8: Overall, the flow of the manuscript is good, please check again about minor grammatical mistakes and errors.

Response: Thank you for your valuable comments and suggestions. We have made every effort to revise the manuscript accordingly. The changes we have implemented do not affect the content or the overall structure of the paper. For your convenience, the revised version is highlighted in **green** to indicate the modifications. We sincerely appreciate the time and effort you have devoted to reviewing our work.

Responses to Reviewer #3:

In this study, guided by DFT calculations, the authors synthesized an In/BiVO₄ photoanode by substituting Bi³⁺ with less electronegative In³⁺. This substitution weakens the surface electron-phonon coupling in BiVO₄, thereby suppressing its PEC activity. The claim of successful substitution and the inhibition of polaron formation is supported by comprehensive characterization techniques, including AC-HAADF-STEM, XAFS, EPR, and Td-PL. Furthermore, the optimized In/BiVO₄/FeOOH photoanode achieves a water oxidation photocurrent density of 6.46 mA cm⁻² at 1.23 V RHE. This work presents a strategic approach to enhance hole utilization efficiency through the modulation of surface hole polaron sites to facilitate carrier release. Nevertheless, the following issues should be addressed before the manuscript can be considered for publication in this journal:

Authors: We sincerely appreciate the insightful and constructive comments on our manuscript. Your suggestions are of great importance in improving our understanding and explanation of the mechanism by which In substitution enhances the PEC performance of the photoanode. In response to your recommendations, we have carefully revised the manuscript in detail, added relevant experimental data, and conducted an in-depth analysis in conjunction with previously reported literature. We believe that these revisions have significantly improved the quality, scientific rigor, and completeness of our work. Thank you once again for your valuable feedback, and we

look forward to receiving further guidance from you. Below are our detailed responses to each of your comments.

Comment 1: The photocurrent of this In/BiVO₄/FeOOH photoanode is relatively high. Therefore, the authors must provide IPCE spectra and irradiation data by wavelength from the solar simulator. Also, the integrated photocurrent density based on the IPCE and solar irradiation (AM1.5G) should be compared to the photocurrent density on the J-V curves. Regarding IPCE spectra of BiVO₄, it is recommended to refer to the following article: Energy Environ. Sci. 17, 2541-2553 (2024); J. Am. Chem. Soc. 2021, 143, 20657–20669

Response: We sincerely appreciate your valuable comments regarding the IPCE results. We agree with your suggestion that a more comprehensive comparative analysis of the IPCE data is necessary. Based on your advice and the referenced literature, we have integrated our IPCE data (**Fig. 3d** in manuscript) with the AM 1.5G solar spectrum to calculate the integrated photocurrent density. This value was then compared with the data derived from the Linear Sweep Voltammetry (LSV, current-voltage) curves and with the results reported in the literature. We believe that this further analysis of the IPCE will significantly enhance the scientific validity of the PEC performance presented in our manuscript. The detailed modifications are as follows:

We have supplemented the irradiance data from the solar simulator and compared it with the standard AM 1.5G solar spectrum (**Supplementary Fig. 22**). The results show that the solar irradiance provided by our xenon lamp source is very close to that of the AM 1.5G spectrum, which demonstrates the scientific validity and accuracy of our PEC data. Subsequently, we integrated the IPCE data with the AM 1.5G spectrum based on the reported Equation R4 to calculate the integrated photocurrent density (J_{int})^{13, 14, 15}. The specific equation is:

$$J_{\text{int}} = \int_{350 \text{ nm}}^{600 \text{ nm}} \frac{\lambda \times E(\lambda) \text{ AM 1.5G} \times \text{IPCE}(\lambda)}{1240} d\lambda \quad (\text{R4})$$

where λ represent the light wavelength (nm), $E(\lambda)$ AM 1.5G represent the corresponding power density (mW cm^{-2}). It should be noted that the original IPCE curve consists of data points measured at approximately 20 nm intervals, which is significantly coarser

than the 1 nm resolution of the AM 1.5G spectrum. To perform the integration, we applied an interpolation and extrapolation method to estimate the IPCE values at 1 nm intervals. A comparison between the interpolated IPCE curve and the original data revealed a near-perfect match, indicating that the interpolation is a reasonable approximation. Subsequently, the 1 nm-interval IPCE data were integrated with the 1 nm-interval AM1.5G spectrum using equation R4 to obtain J_{int} . However, it is important to highlight that, although this integration method effectively accounts for the spectral overlap between the IPCE and the AM1.5G spectrum at a fine wavelength resolution, the resulting J_{int} still differs from the actual photocurrent density measured under the LSV curve at 1.23 V (J_{LSV}). This discrepancy can be attributed to the following four factors:

1. Discrepancy introduced by the integration method

Due to the difference in wavelength intervals between the IPCE and the AM1.5G spectrum, we used interpolation and extrapolation in the integration process, which is essentially an approximation of the IPCE curve. Therefore, this approximation inevitably leads to discrepancies between J_{int} and J_{LSV} . In addition, we only integrated the IPCE values in the wavelength range of 350 nm-600 nm with the AM 1.5G spectrum, which caused the J_{int} to be approximately 0.4 mA cm⁻² lower. It should be noted that the theoretical photocurrent density calculated from the UV-vis spectrum and the AM 1.5G spectrum shows good agreement with the LSV results (Supplementary Fig. 26). Therefore, our IPCE integration results can reflect the photoanode's ability to convert incident photons into electrons to some extent, but cannot fully match the actual photocurrent density obtained from the LSV measurement. Furthermore, we have compiled the IPCE curves from some recently reported works and calculated J_{int} according to our method based on Equation R4. The results show that, although J_{int} obtained by our integration method are all lower than J_{LSV} , this result is still sufficient to evaluate the photoanode's ability to convert incident photons into electrons. We have compiled the IPCE curves, J_{int} calculated according to our method based on Equation

R4, J_{LSV} (1.23V), and some reported J_{int} from these works (**Fig.R1** and **Table R1**). We hope that this comparison can demonstrate the rationality of our photoanode PEC data.

Figures and Tables in response to the reviewers' comments are shown below.

Fig. R1 a-m IPCE curves extracted from reported works and the original IPCE curves. **n** IPCE curves after interpolation and extrapolation. **o** Integrated photocurrent density.

Table R1 Comparison of maximum IPCE value ($IPCE_{max}$), J_{int} calculated according to our method, J_{LSV} (1.23V), and reported J_{int} (J_{int} -reputed).

Number	$IPCE_{max}$	J_{int}	J_{LSV}	J_{int} -reputed	Source of data
1	79	3.11	6.46	-	This Work
2	72	3.57	4.70	~5.3	Adv. Funct. Mater. ¹⁶
3	90	2.61	5.80	-	Angew. Chem. Int. Ed. ¹⁷
4	80	3.39	5.15	-	Nat. Commun. ¹⁸
5	76	3.70	4.60	-	Angew. Chem. Int. Ed. ¹⁹
6	83	4.39	5.45	-	Adv. Energy Mater. ²⁰
7	~99	5.13	6.66	~6.5	Energy Environ. Sci. ²¹
8	~99	5.08	7.01	~6.8	Angew. Chem. Int. Ed. ²²
9	58	2.49	6.20	-	Adv. Funct. Mater. ²³
10	89	3.29	5.91	-	Adv. Mater. ²⁴
11	~60	2.71	~5.00	~4.5	Nat. Sustain. ²⁵
12	~90	4.04	6.00	~6.0	J. Am. Chem. Soc. ²⁶
13	84	3.74	6.00	~6.0	Angew. Chem. Int. Ed. ²⁷

Where (~) denotes an approximate value, and (-) indicates that the corresponding information is not mentioned in the references.

Changes to the revised manuscript are shown below.

Supplementary Information:

Supplementary Fig. 22. a Spectra of the solar irradiance of AM 1.5G and solar simulator. **b** calculated photocurrent densities by integrating corresponding IPCE over AM 1.5G spectra.

2. Discrepancy introduced by the IPCE measurement process

Currently, most IPCE values are derived based on Equation R5^{13, 25, 26, 27}:

$$\text{IPCE} = 1240 \text{ (V} \cdot \text{nm)} \times I \text{ (mA cm}^{-2}\text{)} / (\lambda \text{ (nm)} \times P_{\text{light}} \text{ (mW cm}^{-2}\text{)}) \times 100\% \quad (\text{R5})$$

where 1240 represents a multiplication of Planck's constant and the light speed, I is the photocurrent density, λ is the wavelength of incident light, and P_{light} is the measured light power density at that wavelength. It is important to note that in the experimental and computational determination of IPCE, the key parameters of interest are I at various wavelengths and the incident P_{light} . The value of P_{light} can be directly measured using a light power meter. In contrast, I is typically obtained through two different experimental approaches: linear sweep voltammetry (LSV) and current-time (I -t) measurements. However, it should be emphasized that these two methods yield different current responses due to their distinct measurement principles. Specifically, LSV curve represents the variation of current density with applied voltage, whereas I -t curve reflects the temporal evolution of current density at a fixed voltage. As a result, the values of I derived from these two methods often exhibit noticeable discrepancies. The resulting I from I -t is generally lower than that observed in LSV measurements. This difference arises because LSV tests are conducted at a finite scan rate, which may not allow the system to fully reach a steady state. Consequently, the charge accumulation and migration processes at the interface may be incomplete during the scan, leading to an overestimation of I . In contrast, I -t test provides the system with sufficient time to equilibrate, resulting in a more stable and potentially lower photocurrent density. This difference is schematically demonstrated in Fig. R2, together with the I -t curves from our experimental measurements. The results show that the photocurrent density decreases after the system reaches a steady state, compared to the initial value observed immediately after illumination. This reduction may be one of the contributing factors to the relatively lower J_{int} .

Figures in response to the reviewers' comments are shown below.

Fig. R2 a Comparison schematic of J from I - t and LSV methods. b I - t curves under different wavelengths of light.

3. Effect of the number of wavelength points in IPCE measurement

In IPCE testing, the J and P_{light} need to be measured at different wavelengths, which requires adjusting the wavelength of the incident light source. Typically, the wavelength interval in IPCE measurements is set between 10 and 30 nm. A smaller wavelength interval allows for a more accurate and continuous representation of the IPCE curve. It should be noted, owing to the restriction of the grating in the device, we were only able to measure IPCE values at 10 distinct wavelengths, with an interval of approximately 25 nm. Although the obtained IPCE curve is relatively accurate, the limited number of data points in the range of 420–480 nm may lead to an overall deviation in the integration process. This could be one of the reasons for the relatively lower J_{int} observed in our IPCE calculations.

4. Influence of back-illumination and front-illumination

All of our PEC measurements were conducted under back-illumination. This experimental configuration may lead to an underestimation of the IPCE values. While back-illumination can enhance light utilization and reduce charge carrier recombination, potentially improving the overall PEC performance, the actual photon flux reaching the sample surface may be reduced due to scattering and refraction effects during the light

propagation. As a result, the measured IPCE values might be lower than the true values²⁸. This difference in illumination direction could also contribute to the observed lower J_{int} .

In summary, we have supplemented our manuscript with actual irradiance data and J_{int} derived from the IPCE. A thorough analysis of the integration method, measurement procedure, and experimental setup has been conducted to explain the discrepancies between J_{int} and J_{LSV} . We have also compared our results with those reported in recent literature, calculating J_{int} using the same method. The comparison shows that our J_{int} is generally lower than J_{LSV} values in the literature, but it still reflects the relative capability of the photoanode to convert incident photons into charge carriers. This analysis, along with the supporting data, further validates the reasonableness of our PEC results. We believe that the inclusion of these additional IPCE analyses will improve the quality of our manuscript and provide a foundation for refining our integration and measurement techniques in future studies. We are grateful for your professional insights and the opportunity to enhance the scientific integrity of our work.

Comment 2: After modification with FeOOH, the photocurrent density of the In/BiVO₄ photoanode reached approximately 6.5 mA/cm², whereas FeOOH/BiVO₄ and In/BiVO₄ alone exhibited photocurrent densities of only about 3.0 mA/cm². This performance enhancement appears consistent with the mechanism reported in previous studies (Angew. Chem. Int. Ed. 2023, e202307246; J. Am. Chem. Soc. 2021, 143, 20657–20669). For instance, in this manuscript, the enhanced PEC activity can be attributed to the presence of In on the FeOOH/In/BiVO₄ surface, which facilitated charge carrier migration of In/BiVO₄ toward FeOOH through the formation of In–O–Fe bonds.

Response: We sincerely appreciate your valuable comments, which are crucial for elucidating the mechanism behind the enhanced photocurrent density of the photoanode after cocatalyst loading. In response to your suggestion, we have conducted supplementary XPS measurements to investigate the elemental composition and

chemical state variations on the photoanode surface (**Supplementary Fig. 23**). By comparing the XPS spectra of BiVO₄, In/BiVO₄, BiVO₄/FeOOH, and In/BiVO₄/FeOOH photoanodes, we have confirmed that In³⁺ substitution facilitates the electron transfer from the BiVO₄ semiconductor to the FeOOH cocatalyst. This enhancement is likely attributed to the formation of In-O-Fe bonds, which is in line with the results reported in the literature. Below are our specific analyses.

To evaluate the individual contributions of In³⁺ substitution and FeOOH cocatalyst loading on the BiVO₄ photoanode, we have carried out comparative XPS analyses of In/BiVO₄, BiVO₄/FeOOH, and pristine BiVO₄. The results indicate that after In³⁺ substitution, the XPS peaks of Bi and V shift toward higher binding energies, demonstrating that electrons localized at the Bi and V sites are transferred to the In sites. This conclusion is further supported by XAFS and DFT calculations in the manuscript. Notably, upon loading FeOOH, the XPS peaks of Bi and V do not exhibit a shift toward higher binding energies, but instead show a slight shift toward lower binding energies by approximately 0.1-0.3 eV. This implies that electrons are transferred from FeOOH to BiVO₄.

Subsequently, we compared the XPS spectra of In/BiVO₄/FeOOH with those of In/BiVO₄ and BiVO₄/FeOOH. It was found that, compared to both In/BiVO₄ and BiVO₄/FeOOH, the Bi and V peaks in InBiVO₄/FeOOH shift toward higher binding energies. This suggests that the loading of FeOOH on the In/BiVO₄ surface promotes the transfer of electrons from In/BiVO₄ to FeOOH cocatalyst. Such an electron transfer pathway is beneficial for the separation and migration of photogenerated charge carriers. Further evidence is provided by the high-resolution Fe 2p XPS spectra. As shown in **Supplementary Fig. 23c**, the Fe 2p peak of In/BiVO₄/FeOOH shifts toward a lower binding energy compared to that of BiVO₄/FeOOH, indicating a greater degree of electron transfer from In/BiVO₄ to FeOOH. In summary, through comparative XPS analysis, we have demonstrated that the In/BiVO₄/FeOOH photoanode exhibits a more pronounced electron transfer from BiVO₄ semiconductor to the surface cocatalyst compared to the BiVO₄/FeOOH system. This electron transfer can enhance the

transport efficiency of charge carriers and improve the surface oxygen evolution reaction activity.^{24, 29, 30, 31, 32}

The In^{3+} substitution not only suppresses the recombination of photogenerated holes at HP sites, but also promotes more efficient electron transfer to FeOOH. As a result, the synergistic modification of In^{3+} substitution and FeOOH cocatalyst loading leads to a remarkable improvement in the PEC performance of the photoanode compared to single modifications. We believe that the XPS analysis of electron transfer in the $\text{InBiVO}_4/\text{FeOOH}$ photoanode provides a more comprehensive and mechanistic explanation for the observed PEC performance enhancement. We would like to express our sincere gratitude once again for your insightful and constructive comments. Below are the specific revisions made in the manuscript accordingly.

Changes to the revised manuscript are shown below.

Supplementary Information:

Supplementary Fig. 23. XPS spectrum analysis of BiVO_4 , In/BiVO_4 and $\text{BiVO}_4/\text{FeOOH}$, and $\text{In/BiVO}_4/\text{FeOOH}$ photoanodes.

XPS high-resolution spectra of **a** Bi 4f and **b** V 2p in BiVO_4 , In/BiVO_4 and $\text{BiVO}_4/\text{FeOOH}$, and $\text{In/BiVO}_4/\text{FeOOH}$ photoanodes. **c** XPS high-resolution spectra of Fe 2p in $\text{BiVO}_4/\text{FeOOH}$, and $\text{In/BiVO}_4/\text{FeOOH}$ photoanodes

Page 28: Compared to BiVO_4 , the XPS peaks of Bi and V in $\text{BiVO}_4/\text{FeOOH}$ photoanode do not shift toward higher binding energies, but instead show a slight shift toward lower binding energies by approximately 0.1-0.3 eV. This suggests that electrons are transferred from FeOOH to BiVO_4 . In contrast, the XPS peaks of Bi and V in $\text{In/BiVO}_4/\text{FeOOH}$ photoanode shift toward higher binding

energies, indicating that after loading FeOOH onto the In/BiVO₄ surface, electrons are transferred from In/BiVO₄ to FeOOH. This electron transfer direction is beneficial for the separation and migration of photogenerated charge carriers in BiVO₄^{11,12}. Furthermore, in the high-resolution Fe 2p XPS spectra, the Fe 2p peak of In/BiVO₄/FeOOH is observed to shift toward lower binding energies compared to that of BiVO₄/FeOOH, which demonstrates that a greater amount of electrons is transferred from In/BiVO₄ to FeOOH.

Comment 3: The selective substitution of Bi³⁺ with In³⁺ on the surface of BiVO₄ reduces electron–phonon coupling. Theoretically, a higher In³⁺ substitution level at surface Bi sites should further suppress electron–phonon interactions, thereby improving PEC performance. However, experimental observations are inconsistent with this theoretical expectation. The authors should provide further explanation and discussion to clarify this discrepancy.

Response: We sincerely thank you for your valuable suggestions. As you have pointed out, the PEC activity of our photoanode increases first and then decreases with the extension of the In substitution time, reaching the highest performance at a substitution time of 2 hours. This trend is mainly caused by the influence of the substitution process on the material structure. The detailed analysis is as follows:

We have added the SEM image of photoanode with a substitution time of 10 hours (In10/BiVO₄). (Please refer to **Supplementary Fig. 19**) It can be observed that the surface morphology of the photoanode is partially damaged when the substitution time is excessively long. Although, in theory, a higher surface substitution level could suppress electron–phonon interactions more effectively and thus enhance PEC performance, in practice, prolonged substitution tends to cause structural damage, which leads to a decline in performance. Actually, this structural damage is primarily due to the difference in ionic radii. Although the In³⁺ (0.80 Å) has a smaller radius than the Bi³⁺ ion (1.03 Å), the difference is still significant enough to induce structural distortion after a long substitution process³³. This ionic size mismatch is the intrinsic

reason for the observed decrease in PEC activity under excessive substitution. The phenomenon of performance degradation caused by excessive doping is also reported.

For example, the research results by Resasco et al. indicate that the photocurrent density of BiVO₄ loaded on Ta-doped TiO₂ increases first and then decreases with the increase in Ta doping concentration³⁴. Further analysis suggests that an appropriate Ta doping level can reduce the depletion width in TiO₂, thereby facilitating electron transfer from BiVO₄ to TiO₂. However, an excessive Ta doping concentration leads to an increase in defects within the nanowires, which in turn deteriorates the overall performance. Similarly, Shaddad et al. reported that excessive Zr doping in BiVO₄ materials negatively affects their photoelectrochemical activity³⁵. They suggested that this may be attributed to the reduction in film integrity caused by the difference in ionic radii under high doping conditions. These similar findings indicate that excessive doping levels can reduce the activity of the material. We believe that the analysis of the material morphology can help us better explain the impact of an excessively long substitution time on the PEC performance of the material, thereby enhancing the comprehensiveness of our manuscript. We sincerely thank you for your suggestion once again. In addition, we have added corresponding analysis and explanation in the Supplementary Information.

Changes to the revised manuscript are shown below.

Supplementary Information:

Supplementary Fig. 19. SEM images of In10/BiVO₄ photoanode.

Page 24: A comparison of the SEM images of In/BiVO₄ and In₁₀/BiVO₄ reveals that, under excessively long substitution times, the surface morphology of the photoanode exhibits some degree of damage. This may be the reason for the significant decrease in the photocurrent density observed in the In₁₀/BiVO₄ photoanode.

Comment 4: The authors suggest that the weaker electronegativity of In compared to Bi may contribute to the reduced electron–phonon coupling. It would be valuable to explore whether substituting Bi with other metals of lower electronegativity could similarly enhance the PEC performance of BiVO₄.

Response: We sincerely thank you for your valuable comments, which have significantly contributed to the improvement of our manuscript. It is worth noting that there was a minor error in the description of the electronegativity of metal ions on page 5, line 98 of the original manuscript. Specifically, we mistakenly stated that the electronegativity of In³⁺ is lower than that of Bi³⁺, whereas in fact, the electronegativity of In³⁺ is higher than that of Bi³⁺. This error does not affect the subsequent analysis or the conclusions of the paper. We sincerely apologize for this careless mistake. The conclusion that In has higher electronegativity than Bi is based on previously reported work, and in the Discussion section, we have correctly described this phenomenon³⁶.

Following the reviewer's suggestion, we have further explored whether other metal ions with higher electronegativity could also enhance the PEC performance of BiVO₄ in a similar manner. Based on the differences in electronegativity among metal ions, we selected several metal ions. However, due to time and equipment constraints, we selected Fe³⁺ for comparative experiments and obtained results similar to those observed in the In substitution. We then conducted a series of characterizations, including SEM, TEM, EDS, and XPS, on Fe³⁺-exchanged BiVO₄ (named Fe/BiVO₄) photoanode, and evaluated its PEC performance (**Supplementary Fig. 25**). The results demonstrate that our LPCE strategy can achieve surface exchange using different metal ions as exchange sources, indicating its broad applicability. Moreover, the strategy of

using highly electronegative metal ions for surface exchange to suppress hole polaron formation is also applicable to other such ions, suggesting a certain degree of generality. Thanks to your suggestion, we believe that this further investigation has effectively enhanced the quality of our manuscript. Below are the detailed characterization and testing results for Fe/BiVO₄.

The Fe/BiVO₄ photoanode was prepared using a LPCE method similar to that described in the manuscript (specifically, 0.25 g of Fe(NO₃)₃·9H₂O was used as the Fe source, and the substitution was carried out at 80 °C for 2 hours). The photoanode was characterized using SEM, TEM, XPSS, and EDS. SEM and TEM results revealed that the morphology of the photoanode remains largely unchanged after Fe³⁺ substitution (**Supplementary Fig. 25a and b**). The observed lattice spacing of 0.47 nm in the TEM image corresponds to the (110) crystallographic planes of monoclinic BiVO₄. Element mapping confirmed the uniform distribution of Bi, V, O, and Fe elements. Furthermore, XPS analysis was conducted to investigate the chemical states on the surface of the photoanode (**Supplementary Fig. 25c-e**). The high-resolution XPS spectra of Bi 4f and V 2p showed a shift of the binding energy peaks toward higher values for Fe/BiVO₄ compared to pristine BiVO₄. This shift may be attributed to the higher electronegativity of Fe³⁺ relative to Bi³⁺, which facilitates the transfer of electrons from the Bi and V sites to the Fe site. This phenomenon is consistent with the results obtained from the XPS spectra of In/BiVO₄. Notably, in the Fe 2p high-resolution spectra, two peaks corresponding to Fe³⁺ and Fe²⁺ were observed, despite the fact that Fe³⁺ was used as the substitution source. In combination with the peak shift in Bi 4f and V 2p spectra and the difference in electronegativity between Fe³⁺ and Bi³⁺, this phenomenon can be attributed to the electron transfer induced by Fe³⁺ substitution, which leads to a partial reduction in the valence state of Fe. This conclusion aligns with our findings from the characterization of In/BiVO₄ photoanodes in manuscript. The above characterization results demonstrate that our LPCE strategy is applicable to various substitution sources, enabling broader cationic exchange reactions. More importantly, the XPS results after substitution indicate that this substitution strategy promotes electron transfer toward the

substitution sites, which we believe is beneficial for suppressing the formation of hole polarons under photoexcitation.

To further verify the effectiveness of our strategy on PEC performance, we measured the PEC properties of the Fe/BiVO₄ photoanode (**Supplementary Fig. 25f**). A comparison of the LSV curves of Fe/BiVO₄ and BiVO₄ revealed that surface substitution with Fe³⁺ enhances PEC activity, achieving a current density of 2.77 mA cm⁻². The similar improvement in PEC performance through substitution with other highly electronegative elements suggests that our strategy for suppressing hole polaron formation via surface substitution is of general applicability.

Changes to the revised manuscript are shown below.

Supplementary Information:

Supplementary Fig. 25. Characterization and PEC performance of Fe/BiVO₄ photoanode.

a SEM image, **b** TEM images, **c-e** XPS high-resolution spectra of Fe/BiVO₄ photoanode. **f** LSV curves of BiVO₄ and Fe/BiVO₄ photoanodes. **g** TEM-EDS element mappings of Fe/BiVO₄ photoanode.

Page 30-31: Based on the differences in electronegativity among metal ions, we selected Fe^{3+} as the substitution source and prepared a Fe^{3+} -exchanged photoanode using an LPCE method similar to that in the manuscript (specifically, 0.25 g $\text{Fe}(\text{NO}_3)_3 \cdot 9\text{H}_2\text{O}$ was used as the Fe source, and the exchange was carried out at 80°C for 2 hours), which was named Fe/BiVO_4 . The photoanode was characterized using SEM, TEM, XPS, and EDS. The SEM and TEM results showed that the morphology of the photoanode remained largely unchanged after Fe substitution. The 0.47 nm lattice spacing observed in the TEM image corresponds to the (110) crystallographic planes of monoclinic BiVO_4 . Element mapping revealed a uniform distribution of Bi, V, O, and Fe elements. In addition, XPS analysis was conducted to investigate the chemical states on the surface of the photoanode. In the high-resolution Bi 4f and V 2p XPS spectra, the peaks of Fe/BiVO_4 shifted toward higher binding energies compared to those of pure BiVO_4 . This shift may be attributed to the higher electronegativity of Fe^{3+} relative to Bi^{3+} , which promotes electron transfer from the Bi and V sites to the Fe sites after substitution. Moreover, in the Fe 2p spectrum, two peaks corresponding to Fe^{3+} and Fe^{2+} were observed. This phenomenon can be explained by the increased electron transfer to the Fe sites, resulting in a partial reduction of the Fe valence state. The series of characterization results indicate that our LPCE strategy is applicable to various exchange sources, enabling a broader range of cation exchange reactions.

Subsequently, we evaluated the PEC performance of Fe/BiVO_4 photoanode. Comparison of the LSV curves of Fe/BiVO_4 and BiVO_4 revealed that the surface substitution with Fe^{3+} also enhances the PEC activity, achieving a photocurrent density of 2.77 mA cm^{-2} . Combining the XPS results with the PEC performance measurements, it is evident that the surface substitution with Fe^{3+} also promotes electron transfer to the substitution sites and improves the PEC performance. This suggests that our strategy of using metal ions with higher electronegativity than Bi^{3+} to modulate the hole polaron sites is universally applicable.

Comment 5: In the Figure 4h, the performance comparison of currently reported BiVO_4 -based photoanodes appears incomplete. For instance, several recent studies have demonstrated BiVO_4 -based photoanodes achieving photocurrent densities exceeding $6.5 \text{ mA} \cdot \text{cm}^{-2}$, such as those reported in *Energy Environ. Sci.*, 2022, 15,

2867–2873; *Angew. Chem. Int. Ed.*, 2025, 64(4), e202416340; and *Nat. Commun.*, 2025, 16, 2792. The inclusion of these representative works would provide a more balanced and up-to-date context.

Response: Thank you for your valuable comment regarding the completeness of the PEC performance comparison in Figure 4h. We agree with your suggestion that including more recent and representative works would provide a more balanced and up-to-date context for our results. In the revised manuscript, we have updated **Figure 4h** and **Supplementary Table 12** to incorporate the data from the following studies, as you kindly mentioned. The revised figure now presents a more comprehensive comparison, which better highlights the performance of our In/BiVO₄/FeOOH photoanode in the research landscape. The detailed modifications are listed below.

Changes to the revised manuscript are shown below.

Main Manuscript (Results-PEC Water Oxidation Performance):

Fig. 4 PEC performance for water oxidation.

h Comparison of photocurrent density, ABPE, $\eta_{\text{separation}}$ and $\eta_{\text{injection}}$ values with some representatively reported BVO₄-based photoanodes.

Page 13: In particular, the PEC performance of the In/BiVO₄/FeOOH and In/BiVO₄ photoanodes in weak alkaline electrolytes surpasses that of most reported BiVO₄-based photoanodes (Fig. 4h, 4i,

Supplementary Table 111, 12), underscoring the significant improvement in PEC performance attributed to surface In substitution.

Supplementary Information:

Supplementary Table 12. The comparison of PEC performance between In/BiVO₄/FeOOH photoanode and previously reported BiVO₄ based-photoanodes (100 mW cm⁻² AM 1.5 G solar illumination).

Sample	Photocurrent at 1.23 V _{RHE} (mA cm ⁻²)	ABPE (%)	η _{separation} (%)	η _{injection} (%)	Ref.
BiVO ₄ /Co-TCPP-FAA	5.47	1.93	92	95	15
O _v -BiVO ₄ /MIL-101	5.91	2.15	92	86	16
Co(OH) ₂ /4EtCz-Pc/BVO	6.0	1.54	73	85	17
BiVO ₄ /NiFe-LDH/Co ₃ Ge ₂ O ₅ (OH) ₄	5.15	1.85	93	87	18
D-BiVO ₄ /NiFeOOH	4.63	1.23	85	78	19
NiCo-MOF-CuCrO ₂ -BiVO ₄	5.75	0.62	86	85	20
AuCo(OH) _x /BiVO ₄	6.2	2.1	87	96	21
B(VO) _{1-δ} /NiFeOOH	5.07	1.72	79	86	22
BiVO ₄ @NiFe-LDHs/Ru	4.65	1.40	87	81	23
BiVO ₄ /HPMo:Co	5.70	1.83	96	83	24
two-step-BVO/NiFeO _x	5.54	1.85	98	92	25
Co-Sil/Mo-BiVO ₄	2.82	0.97	61	89	26
FeOOH/V-NiOOH/BVO	5.43	1.80	74	92	27
BiVO ₄ @aNiFe-MOFs	4.34	1.75	83	98	28
FeNiPO _x /BiVO ₄	6.73	~2.25	-	~95	29
Fe-NiOOH/Fe-N-BVO	7.01	1.28	100	95	30
BVO-ΔO _v /FeOOH	7.0	2.78	-	95	31
In/BiVO ₄ /FeOOH	6.46	2.19	97	99	This work

Where (~) denotes an approximate value, and (-) indicates that the corresponding information is not mentioned in the references.

Comment 6: It is recommended to supplement the performance data of the In/BiVO₄/FeOOH photoanode missing in Figure 5 to further elucidate the specific role of In³⁺.

Response:

We sincerely appreciate your valuable comments, which have greatly contributed to improving the quality of our manuscript. As you suggested, the supplementation analysis of the carrier properties and electrochemical data for the In/BiVO₄/FeOOH photoanode is crucial for elucidating the specific role of In³⁺. Following your recommendation, we have added the corresponding data of In/BiVO₄/FeOOH (please refer to **Fig. 5d-f** in manuscript and **Supplementary Fig. 29**), which allows for a comprehensive evaluation of the impact of In³⁺ substitution on the carrier behavior and catalytic activity of the photoanode. The detailed additions and analysis are as follows:

1. Carrier Behavior Analysis

From the analysis of the charge transfer resistance (R_{ct}) in **Fig. 5d** and **Table S13**, it is evident that the R_{ct} value decreases significantly from 607.6 Ω to 94.0 Ω after In³⁺ substitution. This indicates that In³⁺ doping effectively reduces interface transfer resistance and enhances the carrier transfer kinetics. Furthermore, the loading of FeOOH, which exhibits excellent carrier transport properties, further lowers the R_{ct} to 19.9 Ω , demonstrating a substantial improvement in carrier mobility and a reduction in recombination. To further explore the carrier separation and transport dynamics, we analyzed the current density distribution under chopped light conditions (**Fig. 5e**). The decay rate (τ_D) of the pristine BiVO₄ is notably short (1.3 s), which suggests severe carrier recombination and poor transport efficiency. In contrast, the τ_D of In/BiVO₄ increases significantly to 2.5 s, indicating that In³⁺ substitution enhances carrier transport and suppresses recombination. With the addition of FeOOH, which possesses excellent surface catalytic activity, the τ_D of In/BiVO₄/FeOOH is further extended to

3.1 s, reflecting improved carrier utilization efficiency. The donor densities (N_D), derived from Mott–Schottky analysis (**Fig. 5f**), also shows a clear enhancement after In^{3+} substitution. The N_D value increases from 2.12×10^{20} for BiVO_4 to 4.43×10^{20} for In/BiVO_4 . Upon loading FeOOH , the N_D is further elevated to 5.74×10^{20} . These results collectively confirm that In^{3+} substitution effectively modulates HP sites, thereby releasing more hole carriers and reducing recombination, which prolongs carrier lifetime and enhances carrier transport efficiency.

2. Surface Catalytic Activity Analysis

To assess the surface catalytic activity, we examined LSV curves under dark conditions (**Supplementary Figure 29a**). The overpotential of In/BiVO_4 (2.19 V) is not significantly lower than that of BiVO_4 (2.23 V), indicating that In^{3+} substitution has limited influence on the surface catalytic activity. However, the overpotential of $\text{In}/\text{BiVO}_4/\text{FeOOH}$ is reduced by approximately 0.4 V, which suggests that the integration of FeOOH greatly improves the utilization of surface holes during the PEC water oxidation process. Therefore, the enhancement is attributed to the synergistic effect between In-induced hole release and FeOOH 's excellent carrier transport and catalytic properties. The double-layer capacitance (C_{dl}) in the non-Faradaic region of the cyclic voltammetry (CV) curves was also evaluated to reflect the trend of electrochemically active surface area (**Supplementary Figure 29b, e-g**). As observed, In^{3+} substitution does not lead to a significant increase in C_{dl} (0.04 mF cm^{-2} to 0.09 mF cm^{-2}), whereas the C_{dl} of $\text{In}/\text{BiVO}_4/\text{FeOOH}$ (0.19 mF cm^{-2}) is approximately five times higher than that of BiVO_4 , indicating a substantial increase in the electrochemical active surface area. Finally, the Tafel slope was analyzed to evaluate the intrinsic electrochemical activity (**Supplementary Figure 29c**). The Tafel slope of In/BiVO_4 (233 mV dec^{-1}) is only slightly reduced compared to that of BiVO_4 (260 mV dec^{-1}), suggesting that In^{3+} substitution has a limited effect on enhancing the surface catalytic activity. In contrast, the loading of FeOOH significantly lowers the Tafel slope to 76 mV dec^{-1} , which demonstrates a marked improvement in the intrinsic electrochemical activity of the photoanode.

In summary, by incorporating the carrier and electrochemical data of In/BiVO₄/FeOOH, we have conducted a comparative analysis with those of BiVO₄ and In/BiVO₄, providing a comprehensive understanding of the role of In³⁺ in carrier dynamics and catalytic performance. Combined with our DFT calculations, electronic structure characterizations, and carrier property analysis, it is clear that In³⁺ substitution effectively suppresses the formation of HPs, thereby releasing more hole carriers, reducing carrier recombination, and extending carrier lifetime. It is important to note, In³⁺ substitution does not significantly enhance the intrinsic catalytic activity of the surface. Instead, the FeOOH co-catalyst not only improves carrier transport but also exhibits outstanding catalytic activity, efficiently promoting the oxygen evolution reaction (OER) on the surface. The synergistic effect between In³⁺ suppresses the formation of HPs and releases holes and FeOOH's catalytic and transport capabilities leads to a substantial improvement in the overall PEC performance. We believe the addition of In/BiVO₄/FeOOH into the carrier property analysis section significantly aids in clarifying the role of In³⁺ and enhances the scientific quality of the manuscript. We are once again grateful for your expert and insightful suggestions.

Changes to the revised manuscript are shown below.

Main Manuscript (Results-Analysis of HPs and carrier properties):

Fig. 5. **d** EIS spectroscopy, **e** chopped chronoamperometry curves and normalized transient current-time plots, **f** M-S spectroscopy of BiVO₄, In/BiVO₄ and In/BiVO₄/FeOOH photoanodes.

Page 16-17: Following the suppression of HP formation and the subsequent release of additional photogenerated holes, the carrier properties of the photoanodes were significantly modified. These modifications were analyzed using electrochemical testing and spectroscopic characterization.

Furthermore, the carrier properties and catalytic performance of the In/BiVO₄/FeOOH photoanode were also compared to further elucidate the role of In³⁺ substitution. The LSV curves under dark conditions, electrochemical double-layer capacitance (C_{dl}), and Tafel slopes were employed to reveal the changes in the catalytic activity of the photoanodes (Supplementary Fig. 29). The results indicate that the surface In³⁺ substitution does not significantly enhance the kinetics of the water oxidation reaction, whereas the FeOOH co-catalyst, due to its excellent carrier transport properties and catalytic activity, significantly improves the catalytic performance of the photoanode.

The influence of In³⁺ on the carrier properties of the photoanode was further investigated using electrochemical testing. Electrochemical impedance spectroscopy (EIS) results (Fig. 5d, Supplementary Table 11) show that the charge transfer resistance (R_{ct}) of In/BiVO₄ is 94.0 Ω , markedly lower than that of pristine BiVO₄ (607.6 Ω). This finding suggests that In³⁺ substitution significantly reduces interfacial charge transfer resistance, thereby improving carrier transport kinetics. Furthermore, the loading of FeOOH further reduces the R_{ct} to 19.9 Ω , demonstrating a significant improvement in carrier mobility and a reduction in recombination. By analyzing the current density distribution under chopped light and integrating the data, the decay rate (τ_D) of transient photocurrent was determined, which reflects carrier separation and transport dynamics (Fig. 5e). The τ_D for pristine BiVO₄ is notably short (1.3 s), indicative of substantial carrier recombination and poor transport capability. Conversely, the τ_D for the In/BiVO₄ photoanode increased significantly to 2.5 s, demonstrating that In³⁺ substitution effectively enhances carrier transport capacity and inhibits recombination. With the addition of FeOOH, the τ_D of In/BiVO₄/FeOOH was extended to 3.1 s, reflecting improved carrier transport dynamics. The positive slopes observed in the Mott-Schottky analysis (Fig. 5f) indicate that all photoanodes exhibit n-type semiconductor characteristics⁴¹. Notably, the In/BiVO₄ photoanode exhibits a lower slope than pristine BiVO₄, suggesting an increase in carrier density. The calculated donor densities (N_D) are consistent with this observation, revealing BiVO₄(2.12×10^{20}) < In/BiVO₄(4.43×10^{20}), confirming that In³⁺ substitution significantly enhances carrier density⁴². Upon loading with FeOOH, the N_D is further elevated to 5.74×10^{20} . To further investigate the role of In³⁺ substitution in extending carrier lifetime, time-resolved photoluminescence (TRPL) spectroscopy was performed (Fig. 5g). The results revealed that the carrier lifetime of the In/BiVO₄ photoanode (2.33 ns) is significantly enhanced compared to that of the pristine BiVO₄ photoanode (1.81 ns), further corroborating that In³⁺

substitution suppresses surface carrier recombination and prolongs carrier lifetime⁴³.

Collectively, the tests and analyses of carrier characteristics indicate that In³⁺ substitution effectively inhibit surface carrier recombination, extend carrier lifetime, and promote carrier transport. This improvement is attributed to the precise modulation of HP sites (Fig. 5h). Moreover, FeOOH, due to its excellent catalytic activity, can significantly enhance the surface water oxidation capability of the material. It works synergistically with In³⁺ modification to further promote carrier transport at the surface, thereby improving the PEC performance. In conclusion, through the precise modulation of specific sites to suppress electron-phonon coupling and inhibiting the formation of surface HPs, we effectively released a greater number of surface hole carriers, resulting in increased carrier concentration and lifetime. This improvement is critical for improving the overall carrier utilization efficiency.

Supplementary Information:

Supplementary Table 13. Summarized parameters of EIS for the photoanodes at 1.3 V_{Ag/AgCl}.

Sample	R _s (Ω)	R _{ct} (Ω)	C _{ct} (μF/cm ²)
BiVO ₄	50.8	607.6	9.8
In/BiVO ₄	48.3	94.0	16.1
In/BiVO ₄ /FeOOH	42.7	19.9	4.3

Supplementary Fig. 29. Electrochemical testing of photoanodes.

a LSV curves under dark conditions, **b** Double layer capacitance, **c** Tafel slope curves, **d** EIS spectra and **e-g** Cyclic voltammetry (CV) curves tested at scan rates of 20 to 100 mV s^{-1} of BiVO_4 , In/BiVO_4 and $\text{In/BiVO}_4/\text{FeOOH}$ photoanodes.

Page 35-36: The LSV curves under dark conditions show that the overpotential of In/BiVO_4 (2.19 V) is not significantly lower than that of BiVO_4 (2.23 V), while the overpotential of $\text{In/BiVO}_4/\text{FeOOH}$ is notably reduced by approximately 0.4 V. This indicates that the In^{3+} substitution does not significantly enhance the surface catalytic activity, whereas the loading of FeOOH significantly improves the catalytic activity of the photoanode. The double-layer capacitance (C_{dl}) of the photoanodes, calculated from the CV curves in the non-Faradaic region, reveals that the C_{dl} of In/BiVO_4 (0.09 mF cm^{-2}) is somewhat increased compared to BiVO_4 (0.04 mF cm^{-2}), but the C_{dl} of FeOOH is significantly enhanced to about 5 times that of BiVO_4 . A larger C_{dl} implies a greater electrochemical active surface area, suggesting that the FeOOH cocatalyst markedly increases the electrochemical active surface area of the photoanode. The Tafel slope is used to reveal the intrinsic electrochemical activity of the photoanode. It can be observed that the Tafel slope of the In/BiVO_4 photoanode (233 mV dec^{-1}) does not show a significant decrease compared to BiVO_4 (260 mV dec^{-1}), indicating that the In^{3+} surface substitution has limited effect

on improving the electrochemical activity. In contrast, the Tafel slope of In/BiVO₄/FeOOH is reduced to 76 mV dec⁻¹, demonstrating that FeOOH significantly enhances the intrinsic electrochemical activity of the photoanode. The above electrochemical analysis indicates that In substitution does not effectively improve the catalytic activity. As stated in the manuscript, the In³⁺ substitution primarily enhances the PEC performance of the photoanode by modulating the HP sites and releasing more holes. Meanwhile, FeOOH, due to its favorable carrier transport properties and excellent catalytic activity, improves the surface water oxidation capability of the material. It works synergistically with the In³⁺ substitution strategy to jointly enhance the PEC performance of the photoanode.

Responses to Reviewer #4:

Recommendation: Reject

The manuscript addresses the role of polarons in BiVO₄ photoanodes, particularly focusing on hole polarons. While the topic is of potential interest to the community working on photoelectrochemical catalysis, the current work suffers from significant conceptual, methodological, and presentation issues that preclude its publication in Nature Communications. My major concerns are as follows:

Authors: We sincerely appreciate the reviewer's in-depth evaluation and acknowledge the main concerns raised regarding our concept, methodology, and presentation. We recognize the need for significant revisions and are committed to addressing all the issues you have pointed out in a comprehensive manner. We believe that these modifications will greatly strengthen our manuscript and provide a more solid foundation for the topic. In particular, we will focus on a more in-depth and extensive discussion of the role mechanism of polaron and provide direct evidence for the LPCE strategy. To address the existing questions, we will also supplement our work with additional experiments and DFT calculations. We are grateful for the valuable feedback provided by the reviewer and hope that our revisions clearly respond to the concerns raised. We also look forward to any further comments or suggestions. Below are our detailed responses to your comments.

Comment 1: The hole polarons (HPs) in BiVO₄ are widely regarded as small polarons with relatively low activation energies compared to small electron polarons (EPs), which are often considered quasi-free carriers. The manuscript exclusively emphasizes HPs without any discussion or investigation of EPs. The lack of a balanced treatment weakens the overall scientific rationale.

Response: We sincerely appreciate your professional comments on our manuscript. As you rightly pointed out, our discussion on EPs was relatively limited, which affected the overall balance of the manuscript. We apologize for not adequately distinguishing and summarizing the roles of EPs and HPs, and for focusing predominantly on the regulation of HP sites without a broader discussion on EP sites. Based on your valuable suggestions, we have extensively revised and supplemented both the manuscript and the Supplementary Information. These revisions include the physical properties of EPs and HPs, their impact on the PEC system, and how our regulation strategy affects both types of polarons. We hope that these additions and corrections will improve the quality of our manuscript, especially in the theoretical section. Once again, thank you for your positive evaluation and constructive suggestions. Below are the specific revisions we have made:

1. Physical properties of EPs and HPs

As you have highlighted, compared to HPs, EPs are generally considered to possess lower activation energy. Therefore, it is essential to first discuss the physical properties of EPs and HPs in order to clarify the distinctions and connections between them. The concept of polarons was introduced by Solomon Pekar in 1946, referring to the quasiparticle formed by an excess charge carrier (electron or hole) localized in a potential well and accompanied by a local lattice distortion³⁷. Broadly speaking, polarons can be categorized into small and large polarons. In semiconductor oxides such as BiVO₄ and TiO₂, the properties are predominantly influenced by small polarons, and thus we have focused our discussion on small polarons in the response and in the manuscript. Although small polarons can be generated through various mechanisms (including doping, defects, surface functionalization, or photoexcitation), in many

semiconductor oxides they are primarily formed via photoexcitation and tend to localize at different sites, resulting in the formation of EPs and HPs, respectively^{38, 39, 40, 41}. In BiVO₄, relevant studies have indicated that EPs and HPs localize at VO₄ and BiO₈ sites, respectively, accompanied by local lattice expansion and contraction^{7, 12, 42}.

2. Impact of EPs and HPs on PEC Systems

EPs and HPs are localized charge carriers formed by photoexcitation, and their distinct properties result in different impacts on PEC systems. Literature reports indicate that EPs typically follow a hopping conduction mechanism, in which they move from one localized site to another through thermally activated hopping across an energy barrier^{8, 9, 43, 44, 45}. This mechanism leads to a slow charge transport rate in PEC systems, which is often considered a limiting factor for overall performance. The slow charge transport rate may result in severe carrier recombination in the bulk, thus hindering the efficiency of charge utilization. In recent years, several studies have aimed to enhance the charge transport rate by promoting EP hopping. For example, Wu et al. investigated the small polaron hopping mechanism in P-doped BiVO₄ photoanodes and found that the substitution of P for V mediates the regulation of carrier density, trap states, and small polaron hopping, thereby enhancing charge transport dynamics⁴³. Wang et al. utilized an unconventional substitution of Ti⁴⁺ for V⁵⁺, which effectively reduced the polaron hopping barrier, promoting the separation and transfer of bulk charge carriers⁴⁴. However, although these approaches are effective in alleviating bulk recombination, they mainly focus on improving electron mobility, with limited attention given to the enhancement of hole utilization efficiency. Therefore, targeted improvement of the BiO₈ hole polaron units is expected to effectively enhance the hole utilization efficiency in BiVO₄. Indeed, HP sites, particularly those at the surface, significantly influence the release of holes, which severely limits the performance of photoanodes, as surface oxidation reactions primarily rely on holes. While some work has addressed the regulation of bulk BiO₈ sites, the targeted regulation of surface HP sites to promote the release of surface hole polarons remains a critical challenge⁴⁶.

3. Impact of our regulation strategy on HPs and EPs

Our proposed LPCE strategy specifically targets the surface HP sites in the photoanode by in situ substitution of Bi^{3+} with In^{3+} , which has a higher electronegativity. This substitution suppresses the formation of surface HPs, thereby releasing more holes to participate in surface reactions. Although our strategy is focused on surface HP sites, it is indeed necessary to also discuss the surface EP sites in order to comprehensively support our conclusions. To this end, we have supplemented the manuscript with DFT calculations on the formation energy of EP sites, in comparison with the formation energy of HP sites (**Supplementary Fig. 2**). We selected the same VO_4 sites and introduced additional electrons to simulate the localization of EPs in BiVO_4 -DFT and In/BiVO_4 -DFT models. The polaron formation energy (E_p) of the was calculated based on the following equation⁴⁷:

$$E_p = E_s + E_e \quad (\text{R6})$$

where E_s is the strain energy contribution and E_e is the electronic energy. The comparison of E_p reveals that in In/BiVO_4 -DFT model, the E_p of HPs increases significantly (-0.24 eV to -0.05 eV), whereas the E_p of eps remains largely unchanged (-0.59 eV to -0.51 eV, **Fig. 2c** in manuscript **and Supplementary Fig. 2c**). This difference in E_p indicates that our substitution strategy primarily affects the BiO_8 site and has minimal impact on the VO_4 site. The reason is, although we describe the EP and HP sites as VO_4 and BiO_8 , respectively, the holes are more localized at the Bi atoms, while the electrons are more localized at the V atoms. Given the relatively large distance between Bi and V atoms, the regulation of Bi sites has a limited effect on V sites. We have further supported this observation with a schematic illustration of the Bi-V (3.84 Å) and In-V (3.82 Å) bond lengths (**Supplementary Fig. 2d**). In addition, we analyzed the changes in charge density at the VO_4 sites after optical excitation by combining differential charge density with Bader charge analysis (**Supplementary Fig. 2e**). The results show that the charge density at the VO_4 sites in both models exhibits the same trend before and after light excitation: compared to other non-localized sites in the respective models, the VO_4 sites show a higher charge density. However, there is no significant difference in the amount of charge between

the two models. This result indicates that our models can reflect the generation of EPs, and also suggests that there is almost no difference in the formation of EPs between the two models before and after optical excitation, which contrasts with the changes observed in the charge density at the BiO₈ and InO₈ sites in the manuscript. This further confirms that our In-substitution strategy has a minimal effect on the EP sites and mainly contributes to the suppression of HPs formation. In addition, we have also compared the positions of HPs and EPs within the band gap, as well as the local bond length changes at the BiO₈ and VO₄ sites before and after photoexcitation. A detailed discussion on these aspects can be found in the part of Responses to Reviewer #4-Comment 3-Response and Responses to Reviewer #4-Comment 4-Response below.

Overall, based on your valuable comments, we have made significant revisions to the discussion section on EPs and HPs. We have supplemented the manuscript with a series of DFT calculations, including formation energy, DOS, differential charge density, Bader charge analysis, and bond length modeling, in an effort to comprehensively address the issues present in our original manuscript. We believe that these modifications will substantially strengthen the manuscript and provide a more solid theoretical foundation. Thank you again for your insightful feedback.

Changes to the revised manuscript are shown below.

Main Manuscript (Results-DFT-guided Strategies for the Suppression of HPs):

Fig. 2: DFT calculations analyze the suppressive effect of In substitution.

Charge density difference diagram of **a** BiVO₄-DFT and **b** In/BiVO₄-DFT with HP calculated by DFT. **c** The calculated polaronic stability of BiVO₄-DFT and In/BiVO₄-DFT, where E_p , E_s and E_e represent the contributions of polaron formation energy, lattice distortion energy, and electronic energy, respectively. PDOS of **d** BiVO₄-DFT and **e** In/BiVO₄-DFT calculated by DFT with HP. **f** Bi-O radial distribution functions in BiVO₄-DFT and In-O radial distribution functions in In/BiVO₄-DFT with and without HP.

Page 5-6: To further confirm the suppression of polaron formation, we compared the bond lengths of the In/BiVO₄-DFT and BiVO₄-DFT models before and after photoexcitation (Fig. 2f and Supplementary Fig. 3). It was observed that the Bi-O bond length significantly decreases from 2.45 Å to 2.40 Å, whereas the In-O bond length remains at 2.25 Å, further supporting the notion that polarons are less likely to form at InO₈ sites²⁵. Correspondingly, we also investigated the effect of In substitution on EP behavior. Differential charge density, Bader charge analysis, E_p (EP), DOS, and bond length variation data collectively indicate that In substitution has negligible impact on the properties of Eps (Supplementary Fig. 2, 3 and Supplementary Table 1).

Supplementary Information:

Supplementary Fig. 2. DFT calculations of electron polaron (EP) properties

EP and HP localization sites in **a** BiVO₄-DFT and **b** In/BiVO₄-DFT. **c** The calculated polaronic stability, **d** schematic diagrams of Bi-V and In-V bond lengths, **e** charge density difference diagram with EP, **f**, **g** DOS, and **h** a comparison of EP positions in the band gap for BiVO₄-DFT and In/BiVO₄-DFT.

Page 6: There is no significant change in the formation energy of EPs between BiVO₄-DFT and In/BiVO₄-DFT models, indicating that the substitution of In for Bi does not significantly suppress the formation of EPs. The reason for this is that both Bi-V (3.84 Å) and In-V (3.82 Å) bonds are relatively long, making it difficult for the modulation of the Bi site to influence the V site. The differential charge density and Bader charge analysis confirm the presence of EPs, which are localized at the selected VO₄ site and exhibit higher charge density compared to other VO₄ sites. In the DOS, electronic states close to CBM can also be observed, and these states are mainly contributed by V and O atoms. Moreover, it is found that the EP electronic states in BiVO₄-DFT and In/BiVO₄-DFT models are located at 0.36 eV and 0.34 eV below the CBM, respectively, showing no notable difference. Taken together with the results presented in the manuscript, these findings suggest that the substitution of In for Bi can suppress the formation of hole polarons (HPs), but has little effect on the formation of EPs.

Comment 2: The manuscript contains numerous typos and grammatical errors that must be carefully revised. The current language issues significantly hinder readability.

Response: We sincerely appreciate your thorough reading and insightful comments on our manuscript. You are absolutely correct in noting that the typographical and grammatical errors have indeed impaired the clarity and readability of the text. In response, we have carefully revised and proofread the entire document, correcting all noticeable spelling, grammatical, and lexical errors to ensure the accuracy and professionalism of the language. The modifications are not listed here but have been highlighted in **green** in the revised version. We are deeply grateful for your time and attention to detail in reviewing our work.

Comment 3: Electron and hole polarons in BiVO₄ are distinct species, and their different energetic positions within the bandgap need to be clearly discussed. The current Figure 1 is misleading, as it does not differentiate the locations of EPs and HPs. The authors should provide convincing evidence that HPs, rather than EPs, dominate charge recombination in BiVO₄.

Response: Thank you for your professional comments on the physical properties of polarons. We consider this to be a crucial suggestion, as EPs and HPs, being distinct types of polarons, indeed require a clearer discussion of their positions within the band gap. In response, we have added DFT calculations to more explicitly address their locations and combined these with previously reported work to highlight the importance of our strategy for suppressing hole polarons in PEC systems. It is worth noting that in ***Responses to Reviewer #4-Comment 1-Response***, we have already discussed the physical properties of EPs and HPs, their impact on the PEC system, and how our regulation strategy affects both types of polarons. Therefore, in the following, we will focus specifically on the positions of EPs and HPs within the band gap and how they influence the utilization efficiency of charge carriers.

1. Energy positions of EPs and HPs in the band gap

Firstly, we acknowledge that the original Fig. 1 in manuscript may not have clearly conveyed the positions of EPs and HPs. We apologize for any potential confusion this may have caused and have revised **Fig. 1** in the revised manuscript to provide a more intuitive and clear schematic representation. It is important to note that the positions of EPs and HPs in the band gap have already been reported in the literature, and our schematic in the upper left corner of **Fig. 1** is consistent with these findings. Specifically, it shows that the electronic state of the EP is located below the conduction band (CB), while the electronic state of the HP is located above the valence band (VB). For example, Valentin et al. reported based on first-principles electronic structure calculations that during simulated photoexcitation in anatase TiO₂, the self-trapped excited electrons mainly reside on lattice Ti atoms, while the holes are localized on lattice O atoms⁴⁸. The energy positions of the EP and HP within the band gap are 0.73 eV below the CB and 1.83 eV above the VB, respectively. Similarly, Viktor et al. found in BiVO₄ supercells that the EP electronic states are located 1.09 eV below the CB, and the HP electronic states are located 0.78 eV above the VB⁷. These results confirm that our revised schematic accurately reflects the typical positions of polarons in the band gap.

In addition, we have supplemented DOS plots to further discuss the positions of different polaron models in the band gap of our system (Please refer to **Supplementary Fig. 2f and g**). By comparing **Supplementary Fig. 2f and g**, it can be observed that under simulated photoexcitation, both models exhibit occupied electronic states near the conduction band minimum (CBM), which correspond to the formation of EPs due to excess electrons. The Projected density of states (PDOS) reveals that these electronic states are mainly contributed by V and O atoms, indicating that the EPs are localized at VO₄ sites, and more specifically at V atoms. This finding is analogous to our previous conclusion regarding HPs, which are localized at BiO₈ sites. Furthermore, the corresponding differential charge density and Bader charge analysis also support the localization of both EPs and HPs at their respective formation sites in our models (**Supplementary Fig. 2e**). Next, we further discuss the specific energy positions of Eps (**Supplementary Fig. 2h**). It is important to note that since our strategy effectively suppresses the formation of HPs (the electronic states observed in the In/BiVO₄-DFT

model are due to excess electrons delocalizing over other Bi and O atoms, **Fig. 2e and f** in manuscript), the discussion of the HP electronic state position in the band gap is not practically distinct. Therefore, we mainly compare the EP positions in the band gap. The EP states in the BiVO₄-DFT and In/BiVO₄-DFT models are located at 0.36 eV and 0.34 eV below the CBM, respectively, with no significant difference. This indicates that the In substitution has a minimal impact on the energy position of EPs in the band gap.

2. Impact of HPs and EPs on charge recombination in BiVO₄

As you pointed out in Responses to Reviewer #4-Comment 1, our previous discussion on the influence of HPs and EPs on charge recombination in BiVO₄ was insufficient, particularly in terms of the role of EPs. Following your suggestions, we have added DFT calculations on EPs and provided a comprehensive discussion on the physical properties of both EPs and HPs, their effects on PEC performance, and how our strategy influences them (Please refer to Responses to Reviewer #4-Comment 1-Response). In general, the hopping mechanism of EPs leads to bulk charge recombination, which limits the performance of PEC systems. Previous studies have demonstrated that doping or defect engineering can facilitate the hopping of EPs, thereby promoting bulk charge transport^{8, 43, 44, 49}. However, since the surface of photoanodes mainly involves hole-participating oxidation reactions, and HPs are more restrictive in terms of hole release, the regulation of surface HP sites to release more holes is of significant value in improving the efficiency of photoanodes. A more detailed discussion on the impact of HPs and EPs on charge recombination in BiVO₄ can be found in the second point of Responses to Reviewer #4-Comment 1-Response.

Changes to the revised manuscript are shown below.

Main Manuscript (Results–Theoretical design for HPs suppression):

Fig. 1: Schematic diagram of the HPs suppression mechanism.

Fig. 1 illustrates the formation and suppression mechanism of HPs. In the BiVO₄ photoanode, the self-trapping of photogenerated carriers leads to the formation of EPs (VO₄ sites) and HPs (BiO₈ sites), with HPs significantly limiting the utilization of surface hole carriers. By selectively substituting Bi³⁺ with In³⁺, the formation energy of polarons is increased, thereby suppressing the generation of HPs and releasing more hole carriers.

Main Manuscript (Results-DFT-guided Strategies for the Suppression of HPs):

Fig. 2: DFT calculations analyze the suppressive effect of In substitution.

PDOS of **d** BiVO₄-DFT and **e** In/BiVO₄-DFT calculated by DFT with HP.

Page 5-6: Subsequently, we analyzed the electronic states using the Projected density of states (PDOS) method (Fig. 2d, e). The results indicate that in BiVO₄-DFT model, excess holes occupy the O and Bi sites, forming localized HPs. In contrast, in In/BiVO₄-DFT, there is no hole localization at the In sites. Instead, excess holes tend to accumulate at other Bi and O sites, suggesting that In sites are less susceptible to polaron formation. To further confirm the suppression of polaron formation, we compared the bond lengths of the In/BiVO₄-DFT and BiVO₄-DFT models before and after photoexcitation (Fig. 2f and Supplementary Fig. 3). It was observed that the Bi-O bond length significantly decreases from 2.45 Å to 2.40 Å, whereas the In-O bond length remains at 2.25 Å, further supporting the notion that polarons are less likely to form at InO₈ sites²⁵. Correspondingly, we also investigated the effect of In substitution on EP behavior. Differential charge density, Bader charge analysis, E_p(EP), DOS, and bond length variation data collectively indicate that In substitution has negligible impact on the properties of Eps (Supplementary Fig. 2, 3 and Supplementary Table 1).

Supplementary Information:

Supplementary Fig. 2. DFT calculations of electron polaron (EP) properties

f, g DOS, and **h** a comparison of EP positions in the band gap for BiVO₄-DFT and In/BiVO₄-DFT.

Page 6: In the DOS, electronic states close to CBM can also be observed, and these states are mainly contributed by V and O atoms. Moreover, it is found that the EP electronic states in BiVO₄-DFT and In/BiVO₄-DFT models are located at 0.36 eV and 0.34 eV below the CBM, respectively, showing no notable difference. Taken together with the results presented in the manuscript, these findings suggest that the substitution of In for Bi can suppress the formation of hole polarons (HPs),

but has little effect on the formation of EPs.

Comment 4: Small polaron formation typically induces lattice expansion to accommodate the localized carrier. However, the theoretical results in this work indicate a reduced Bi–O bond length, which appears contradictory. The authors should provide a clear explanation of this discrepancy.

Response: We sincerely thank you for the professional question regarding the relationship between polaron formation and bond length changes. We agree with your point that a more explicit explanation of the bond length expansion or contraction during polaron formation is necessary. In response, we have supplemented DFT calculations to provide a more detailed discussion of the relationship between the formation of electron and hole polarons and the corresponding changes in metal-oxygen (M-O) bond lengths (**Supplementary Fig. 3**). In general, the formation of EP is accompanied by lattice expansion, resulting in an increase in M-O bond length, whereas the formation of HP is associated with lattice contraction, leading to a decrease in M-O bond length. The specific analysis is as follows:

Firstly, to better distinguish the bond length changes during the formation of EP and HP, we have added the radial distribution function of V-O bonds. As shown in **Supplementary Fig. 3a**, in contrast to the observed decrease in Bi-O bond length under photoexcitation, the V-O bond length increases after photoexcitation. Moreover, it is evident that the substitution of In has a significant inhibitory effect on the reduction of In-O bond length under photoexcitation, but no obvious influence on the increase of nearby V-O bond lengths. Subsequently, we present the changes in M-O bond lengths (including V-O, Bi-O, and In-O) before and after photoexcitation in both BiVO₄-DFT and In/BiVO₄-DFT models. From the **Supplementary Fig. 3b**, it can be seen that in both models, the V-O bond length increases by approximately 0.1 Å after photoexcitation. Since the V-O site is the location where EP form, this result suggests that the formation of EP is associated with an increase in M-O bond length. For the HP formation site, in BiVO₄-DFT model, the Bi-O bond length decreases by about 0.05 Å

after photoexcitation, indicating that HP formation is accompanied by a reduction in M-O bond length. However, in In/BiVO₄-DFT model, the In-O bond shows almost no shortening, suggesting that the formation of In-O site effectively suppress the formation of HP. This is consistent with the conclusions in our manuscript.

Finally, we summarize the relationship between the two types of polaron formation and the corresponding bond length variations at their respective sites, as shown in **Supplementary Fig. 3c**. Specifically, the formation of EP induces local lattice expansion, while the formation of HP induces local lattice contraction. Therefore, our theoretical result showing a decrease in Bi-O bond length is in line with the mechanism of hole polaron formation.

This relationship between polaron formation and bond length variation has been reported in previous literature. For example, Wiktor et al. observed that electron polarons localize around vanadium atoms, reducing the oxidation state of V and increasing the V-O bond length from approximately 1.72 Å to 1.81 Å⁷. Hole polarons, on the other hand, are mainly distributed among one bismuth and eight oxygen atoms, with the charge localization causing the Bi-O bond length to decrease from an average of 2.48 Å to 2.42 Å. Similarly, Liu et al. noted that when hole polarons localize in BiO₈ unit, the Bi-O bond length shortens by about 0.03 Å¹⁰. These findings are consistent with our conclusions.

We hope that the supplementary DFT calculations and the analysis based on the referenced literature can provide a satisfactory explanation for your concern.

Changes to the revised manuscript are shown below.

Supplementary Information:

Supplementary Fig. 3. Relationship between polarons and bond length changes.

a V-O radial distribution functions and **b** variations in bond length within the local model in BiVO₄-DFT and In/BiVO₄-DFT with and without the hole polaron. **c** Schematic illustration of polaron formation.

Page 7: As shown in Supplementary Fig. 2 a, the V-O bond length is observed to increase after photoexcitation. In combination with Figure 2 f from the manuscript, it can be seen that the substitution of In significantly suppresses the shortening of the In-O bond under photoexcitation, but has little effect on the elongation of the V-O bond. A comparison of the models before and after photoexcitation reveals that in both BiVO₄-DFT and In/BiVO₄-DFT models, the V-O bond length increases by approximately 0.1 Å after photoexcitation. This can be attributed to the local lattice expansion induced by the formation of the electron polaron (EP). In contrast, the hole polaron (HP) site exhibits different behavior. In BiVO₄-DFT model, the Bi-O bond length decreases by about 0.05 Å after photoexcitation, while in In/BiVO₄-DFT model, the In-O bond shows almost no shortening. This indicates that the formation of the In-O site effectively suppresses the generation of hole polarons. Schematic illustration of polaron formation demonstrate the relationship between HP and EP formation and bond length changes. Specifically, the formation of EP requires local lattice expansion, while the formation of HP is associated with local lattice contraction.

Comment 5: The claim that In^{3+} substitutes Bi^{3+} rather than V^{5+} lacks direct proof.

Rigorous evidence combining DFT simulations with experimental characterization is required to substantiate this critical point.

Response: We sincerely appreciate your valuable comments. The evidence for the selective substitution of In^{3+} for Bi^{3+} is of crucial importance to our manuscript. We fully agree with your suggestion that a comprehensive analysis combining DFT calculations and experimental characterizations is essential to conclusively demonstrate the selectivity of this substitution. Based on your feedback, we have supplemented DFT calculations and additional experimental measurements, and integrated them with existing data to systematically analyze and confirm the mechanism of this selective substitution. We believe that the further in-depth analysis of the selective substitution mechanism in combination with DFT calculations will significantly enhance the quality of our manuscript. The following are the specific revisions we have made.

1. DFT calculations of formation energy

To investigate the mechanism of selective substitution, we performed DFT calculations of the formation energy for In substituting at different sites in the BiVO_4 lattice (**Supplementary Fig. 18** and **Supplementary Table 9**). Specifically, we constructed two substitutional models: In/ BiVO_4 -DFT(Bi): one In atom replacing a Bi atom in the BiVO_4 supercell; In/ BiVO_4 -DFT(V): one In atom replacing a V atom in the same supercell. The substitutional formation energy (E_{sub}) was calculated using the following equation:

$$E_{\text{sub}} = E(\text{sub. BiVO}_4) - E(\text{BiVO}_4) + E_{\text{sub. atom}} - E_{\text{In}} \quad (\text{R7})$$

where $E(\text{sub. BiVO}_4)$ is the total energy of the substituted BiVO_4 ; $E(\text{BiVO}_4)$ is the total energy of the pristine BiVO_4 ; $E_{\text{sub. atom}}$ is the chemical potential of the substitutional atom; E_{In} is the chemical potential of In atom. The results show that the E_{sub} for In substituting at the Bi site is 0.03 eV, while that for In substituting at the V site is 2.54 eV. The significantly lower formation energy for the Bi site substitution indicates that In^{3+} preferentially substitutes Bi^{3+} from an energetic standpoint. Therefore, our strategy

of selective substitution at the Bi site is thermodynamically favorable, supporting the feasibility of the selective substitution process.

2. Experimental characterization

To confirm the elemental composition and the site selectivity of In incorporation into the BiVO₄ photoanode, we conducted a series of experimental characterizations, including XPS, etch-XPS, ICP, AC HAADF-STEM, and XAFS. These methods provided multi-faceted evidence for the selective substitution of In³⁺ at Bi³⁺ sites.

(1) XPS Analysis

We compared the cationic composition before and after In³⁺ substitution by normalizing the XPS peak intensities of Bi, V, and In. The results show a significant decrease in Bi content, only a slight decrease in V content, and a significant increase in In content. This elemental trend confirms that In selectively substitutes Bi rather than V (**Fig. 3b**, **Supplementary Figure 14a-c** and **Supplementary Table 5**).

(2) Etch-XPS Analysis

To further investigate the depth-dependent element distribution, we performed etch-XPS analysis. The results reveal that with increasing etching depth, Bi content significantly increases, V content slightly increases, and In content gradually decreases. This indicates that the substitution occurs predominantly at the surface, and that In preferentially occupies Bi sites, supporting the selective substitution mechanism (**Fig. 3c** and **Supplementary Figure 14d-f**).

(3) ICP Analysis

We also carried out ICP analysis to quantify the elemental composition. As shown in **Supplementary Table 8**, after In³⁺ substitution, the content of Bi is significantly reduced (from 51.62% to 50.93%), while the content of V remains almost unchanged (from 48.38% to 48.36%), and the content of In increases to 0.71%. In contrast, the Bi content decreases by 0.69%, whereas the V content decreases by only 0.02%, which strongly demonstrates that our LPCE strategy enables selective substitution of Bi by In.

(4) AC HAADF-STEM images

To analyze the atomic-scale structure, we employed AC HAADF-STEM. The image in **Figure 3a** clearly shows distinct dark spots, which correspond to the In atoms substituting Bi atoms. This is attributed to the lower atomic number (49) of In compared to Bi (83), resulting in weaker electron scattering and resulting in a darker appearance in the image. The intensity profile along the dashed line in **Figure 3a** reveals high-intensity peaks corresponding to Bi atoms, and lower-intensity features associated with In atoms. It should be noted that V atoms, with a much lower atomic number (23), contribute minimally to the HAADF signal, so the observed intensity reduction is primarily due to the substitution of Bi by In. This is further supported by **Supplementary Figure 13**, which shows the AC HAADF-STEM image of the (110) plane. In this image, the Bi atoms appear brightest, the In atoms are moderately bright, and the V atoms are the darkest, clearly demonstrating that the dark spots in **Figure 3a** are In atoms substituting Bi atoms. This confirms that In successfully enters the BiVO₄ lattice and occupies Bi sites, as achieved through our LPCE method.

(5) XAFS Analysis

To investigate the local coordination environment of In, we performed XAFS analysis, including XANES and EXAFS, and compared the results with the coordination environment of Bi in pristine BiVO₄. (**Fig. 3d-g**, **Supplementary Fig. 17** and **Supplementary Table 10**) The EXAFS fitting results indicate that the In-O coordination number in In/BiVO₄ is similar to that of Bi-O in BiVO₄, suggesting that In³⁺ forms a coordination environment analogous to Bi³⁺ (InO₈ site). This is a direct and strong evidence for the selective substitution of Bi by In. Specifically, the In-O bond length, obtained from EXAFS fitting, is approximately 2.159 Å, which is close to the Bi-O bond length in BiVO₄ (2.199). This further confirms that In³⁺ substitutes for Bi³⁺, forming an eight-coordinate environment with oxygen atoms, rather than a four-coordinate structure as would be expected if In substituted V. Hence, the coordination number and bond length data from XAFS analysis strongly support the selective substitution of Bi by In.

3. Summary

In summary, we have systematically analyzed and verified the selective substitution of In^{3+} for Bi^{3+} using a combination of experimental and computational approaches: XPS, etch-XPS, and ICP analysis collectively confirms a significant decrease in Bi content and indicate that In preferentially substitutes for Bi rather than V. AC HAADF-STEM imaging offers direct atomic-scale evidence, showing that In atoms occupy the original Bi sites. XAFS measurements demonstrate that In forms an InO_8 coordination site, analogous to the BiO_8 site in BiVO_4 , further supporting the selectivity of In substitution at the Bi site. DFT calculations show that substitution of In at the Bi site is energetically more favorable than at the V site, offering theoretical insight into the mechanism of selective substitution. These complementary experimental and computational results consistently support the selective substitution of In^{3+} at the Bi^{3+} site. We believe that this multifaceted approach not only strengthens the scientific foundation of our work, but also deepens the understanding of the structure-property relationship in the modified BiVO_4 system. Once again, thank you for your valuable comments. The revisions made based on your suggestions have significantly improved the quality of our manuscript. The following are the modified sections in the main manuscript and the Supplementary Information.

Changes to the revised manuscript are shown below.

Main Manuscript (Results-Synthesis and Structural Characterization):

Fig. 3: Material characterization.

a AC HAADF-STEM images of In/BiVO₄ photoanode. **b** XPS high-resolution spectroscopy of Bi 4f and V 2p in BiVO₄ and In/BiVO₄ photoanodes. **c** etch-XPS spectroscopy of In/BiVO₄ photoanode. **d** In K-edge XANES spectroscopy of In foil, In₂O₃ and In/BiVO₄. **e** Fourier transformed k^3 -weighted $\chi(k)$ function of the EXAFS spectroscopy for In K-edge. **f** corresponding EXAFS fitting curve for In/BiVO₄. **g** WT-EXANES of In foil, In₂O₃ and In/BiVO₄. **h** LPCE process and the carrier transfer of different sites diagram.

Page 8-10: To accurately determine the presence of In³⁺, Spherical aberration corrected high-angle annular dark-field scanning transmission electron microscopy (AC HAADF-STEM) was employed. The isolated dark dots in Fig. 3a indicate the atomic dispersion of In atoms, rather than the formation of clusters or nanoparticles. Supplementary Fig. 13 presents an image of the (110) crystal plane and a schematic of atomic structure. These results indicate that surface In³⁺ is incorporated into the BiVO₄ lattice by substituting Bi³⁺, while the overall lattice structure remains intact, confirming the successful achievement of selective and uniform substitution through the LPCE method.

X-ray photoelectron spectroscopy (XPS) was employed to investigate the elemental states and chemical composition of the surface region. Quantitative elemental analyses via EDS and XPS revealed differences in the elemental composition between the bulk and surface regions, indicating that LPCE initially occurs at the solid-liquid interface before extending inward (Supplementary Table 4, 5). Furthermore, the significant reduction in surface Bi content suggests that In preferentially substitutes Bi rather than V. The high-resolution Bi 4f spectrum exhibited symmetric peaks at 164.34 eV and 159.04 eV, corresponding to Bi 4f_{5/2} and Bi 4f_{7/2} of Bi³⁺, respectively (Fig. 3b). Similarly, the high-resolution V 2p XPS spectrum revealed two symmetric peaks at 524.33 eV and 516.63 eV, corresponding to V 2p_{1/2} and V 2p_{3/2} of V⁵⁺ (Fig. 3b)²⁸. Notably, the binding energies of the Bi 4f and V 2p peaks in In/BiVO₄ are shifted to higher values compared to pristine BiVO₄, suggesting an electron transfer from Bi and V sites to In sites after LPCE. Moreover, the potential effects induced by local strain and oxygen vacancies were systematically discussed and excluded (Supplementary Fig. 15). The In 3d_{3/2} and In 3d_{5/2} peaks in the In 3d spectroscopy confirm the presence of In³⁺, indicating that the valence state of In in the samples is consistent with that of the In source (In(NO₃)₃), which remained constant throughout LPCE process (Supplementary Fig. 14b)²⁹. Further analysis of elemental composition changes and electron transfer was conducted using etching combined with XPS (etch-XPS). The results indicate that as the etching depth increases, the In content decreases while the Bi content significantly increases, with only a moderate increase in V content, suggesting that In primarily substitutes Bi sites (Fig. 3c). Additionally, the binding energy shifts further confirm the electron transfer from Bi and V sites to In sites (Supplementary Fig. 14d-f).

To further elucidate the electronic state and local coordination environment of In, X-ray absorption fine (XAFS) spectroscopy was employed. As shown in Fig. 3d, the K-edge X-ray absorption near-edge structure (XANES) spectroscopy of In/BiVO₄ closely resembles that of In₂O₃, implying that the oxidation state of In in In/BiVO₄ is predominantly trivalent. Notably, the absorption edge of In/BiVO₄ shifts toward lower binding energies compared to that of In₂O₃, consistent with the XPS results. This shift indicates a decrease in the oxidation state of In species, attributed to the higher electronegativity of In compared to Bi, resulting in electron transfer to the InO₈ sites formed via In substitution. Extended X-ray absorption fine structure (EXAFS) analysis

was carried out to elucidate the local coordination environments of Bi and In species. The local coordination environment of Bi species in BiVO₄ was carefully analyzed using the Bi L-edge X-ray absorption spectrum (Supplementary Fig. 17). The results revealed that in BiVO₄, the Bi species predominantly exhibit Bi-O coordination in the first coordination shell. By fitting the EXAFS data, a coordination number of approximately 7.4 for Bi-O was obtained, indicating the presence of BiO₈ sites (Supplementary Table 10)^{30,31,32}. As shown in Fig. 3e, the R-space transformation of the EXAFS spectrum reveals a pronounced peak at 1.62 Å, corresponding to In-O bonding. The fitted EXAFS results show that in In/BiVO₄, the In-O bond length in the first coordination shell is 2.159 Å, with a coordination number of approximately 7.72, which is significantly different from that in In₂O₃ (6). The very similar coordination numbers between the Bi species in BiVO₄ and the In species in In/BiVO₄ suggest the presence of InO₈ sites in In/BiVO₄, analogous to BiO₈. This observation provides strong structural evidence in support of the selective replacement of Bi sites via the LPCE strategy. Additionally, wavelet transform (WT) simulations were carried out to analyze the radial distance resolution in K-space. As shown in Fig. 3g1-g3, the WT intensity maxima corresponding to In-O coordination near 5.0 Å⁻¹ are well-resolved between 1.0 and 2.0 Å, with no significant In-In coordination observed, indicating that In is predominantly coordinated with O atoms in the sample.

The ICP results also support the conclusion of the selective substitution of In for Bi (Supplementary Table 8). Moreover, we conducted DFT calculations to reveal the underlying mechanism of the selective substitution (Supplementary Fig. 18 and Supplementary Table 9). The significantly lower formation energy for In substituting the Bi site (0.03 eV) compared to that for substituting the V site (2.54 eV) indicates that the substitution of In for Bi is thermodynamically favorable, enabling selective substitution. Furthermore, DFT calculations elucidated the electron transfer behavior induced by the In substitution (Supplementary Fig. 5, Supplementary Table 3). The differential charge density and Bader charge analysis on the (110) crystal plane reveal electron accumulation at the In sites, and these results are in good agreement with the XPS and synchrotron radiation results. Collectively, the analysis demonstrates that the InO₈ sites are formed after LPCE, which facilitates electron transfer from Bi and V sites to In sites. This selective regulation effectively modulates the electronic structure of specific sites and provides an accurate experimental framework

consistent with theoretical predictions, thereby elucidating the role of HP suppression in enhancing PEC performance.

Supplementary Information:

Supplementary Fig. 13. AC HAADF-STEM images and schematic atomic structure of In/BiVO₄ photoanode.

Supplementary Fig. 14. XPS spectrum analysis of BiVO₄, In/BiVO₄ and In/BiVO₄/FeOOH photoanodes.

a XPS full spectrum of BiVO₄, In/BiVO₄ and In/BiVO₄/FeOOH photoanodes. **b** XPS high-resolution spectra of In 3d in In/BiVO₄ photoanode. **c** XPS high-resolution spectra of O 1s in BiVO₄ and In/BiVO₄ photoanodes. Etch-XPS high-resolution spectra of **d** Bi 4f, **e** V 2p and **f** In 3d in In/BiVO₄ photoanode.

Page 18: The XPS full spectrum confirmed the presence of Bi, V, O and In in In/BiVO₄, aligning

with the EDS results. The XPS high-resolution spectra of O 1s displayed asymmetric peaks indicative of lattice oxygen (O_L), oxygen defects (O_V) and chemisorbed or dissociated oxygen species (O_C), respectively (Supplementary Table 6)8. These results indicate that the milder LPCE conditions did not result in a deficiency of O.

With increasing etching depth, the peaks in the etch-XPS high-resolution spectra of Bi and V both shift toward lower binding energies, indicating that the Bi and V sites lose more electrons as the In content increases. In the etch-XPS high-resolution spectra of In 3d, the peaks corresponding to In $3d_{3/2}$ and In $3d_{5/2}$ shift toward higher binding energies, while the peak corresponding to Bi $4d_{5/2}$ shifts toward lower binding energies. This suggests that, with the increase in In content, the In sites gain more electrons. The results of these three spectra collectively indicate that the substitution of In facilitates the transfer of electrons from the Bi and V sites to the In sites.

Supplementary Fig. 17. Bi L-edge XAFS data of Bi foil, Bi_2O_3 and $BiVO_4$

a Bi L-edge XANES spectroscopy of Bi foil, Bi_2O_3 and $BiVO_4$. **b** Fourier transformed k^3 -weighted $\chi(k)$ function of the EXAFS spectroscopy for Bi L-edge. **c** k^3 -weighted EXAFS signal in k -space for Bi L-edge. **d, e** Corresponding EXAFS fitting curve for $BiVO_4$ and Bi_2O_3 . **f** Fourier transformed

k^3 -weighted $\chi(k)$ function of the EXAFS spectroscopy of In/BiVO₄ and BiVO₄. g WT-EXANES of Bi foil, Bi₂O₃ and BiVO₄.

Page 22: From the Fourier-transformed EXAFS spectrum of BiVO₄, a spectral peak corresponding to the first coordination shell can be observed, which exhibits features similar to those in Bi₂O₃, but distinct from those in Bi foil. This indicates that Bi-O coordination is likely present in the first coordination shell of BiVO₄. Moreover, the wavelet transform plots also reveal a Bi-O bonding feature in the first shell that is analogous to that in Bi₂O₃. Subsequently, we performed a fitting analysis on the EXAFS data of BiVO₄ to further determine the Bi-O coordination number. The results show that the Bi-O coordination number in BiVO₄ is approximately 7.4 (Supplementary Table 10).

Supplementary Fig. 18. DFT calculations of the substitutional formation energy (E_{sub}) for different substitutional sites

Page 23: In/BiVO₄-DFT(Bi): one In atom replacing a Bi atom in the BiVO₄ supercell; In/BiVO₄-DFT(V): one In atom replacing a V atom in the same supercell. E_{sub} for In substituting at the Bi site is 0.03 eV, while that for In substituting at the V site is 2.54 eV. The significantly lower formation energy for the Bi site substitution indicates that In³⁺ preferentially substitutes Bi³⁺ from an energetic standpoint.

Supplementary Table 5. The cation elements ratio of BiVO₄ and In/BiVO₄ electrodes by XPS. (Normalized according to the ratio of cations).

Sample	Bi (at. %)	V (at. %)	In (at. %)
BiVO ₄	60.2	39.8	-
In/BiVO ₄	45.2	31.6	23.2

Supplementary Table 8. The cation elements ratio (at. %) of BiVO₄ and In/BiVO₄ electrodes obtained from ICP analysis (Normalized according to the ratio of cations).

Sample	Bi (at. %)	V (at. %)	In (at. %)
BiVO ₄	50.16	49.84	-
In/BiVO ₄	49.68	49.68	0.64

Supplementary Table 9. Substitutional formation energy (E_{sub}) at different substitutional sites

Sample	E (sub. BiVO ₄)	E (BiVO ₄)	$E_{\text{sub. atom}}$	E_{In}	E_{sub}
In/BiVO ₄ -DFT(V)	-314.27	-323.17	-3.83	-2.58	2.54
In/BiVO ₄ -DFT(Bi)	-321.89	-323.17	-8.94	-2.58	0.03

Supplementary Table 10. Structural parameters of In/BiVO₄ and BiVO₄ extracted from the EXAFS fitting.

Sample	shell	CN	R (Å)	σ^2	ΔE_0	R factor
In ₂ O ₃	In-O	6*	2.165	0.004	3.33	0.006
In/BiVO ₄	In-O	7.72±0.9	2.159	0.015	-5.59	0.005
Bi ₂ O ₃	Bi-O	3*	2.133	0.005	-5.81	0.011
BiVO ₄	Bi-O	5.08±1.1	2.199	0.010	-0.18	0.004
		2.32±1.1	2.525		-0.04	

Comment 6: In lines 308–309, the authors argue that phonon-assisted recombination of photo-generated carriers is enhanced at higher temperatures. This interpretation is inconsistent with the reported results, where one would instead expect recombination to be suppressed at elevated temperatures. This contradiction requires clarification.

Response: We would like to express our sincere gratitude for your valuable comments on the presentation of our temperature-dependent photoluminescence (Td-PL) spectroscopy section. We apologize for any ambiguities or inconsistencies that may have resulted from the initial formulation. In response, we have revised and elaborated the data analysis part of the Td-PL spectra to improve clarity and completeness. Additionally, we have summarized the discussion from two key perspectives: the relationship between carrier recombination and temperature, and the distinctions and connections between our results and those reported in the literature. The revised content is as follows:

1. Relationship between photogenerated carrier recombination and temperature in Td-PL spectroscopy

In the manuscript, we stated that the intensity of the Td-PL peak decreases with increasing temperature, indicating that phonon-assisted recombination of photogenerated carriers is suppressed at low temperatures (**Fig.5a** and **Supplementary Fig. 27b-d**). We sincerely apologize for the insufficient elaboration of this section in the original manuscript, which may have led to potential misunderstandings. In fact, lower temperatures can suppress the key phonon-assisted recombination process, thereby inhibiting the recombination of photogenerated carriers and leading to a stronger PL emission peak⁵⁰. However, as the temperature increases, the intensity of lattice vibrations also increases, which enhances the lattice relaxation of the luminescent centers⁵¹. This, in turn, promotes nonradiative transitions and ultimately results in a decrease in the PL peak intensity. Furthermore, the analysis of the Huang–Rhys factor (S), derived from fitting the temperature-dependent full-width at half-maximum (FWHM) of the PL peaks, provides additional evidence for the suppression of polaron formation due to the In surface substitution.

2. Distinctions and connections between our results and literature reports

Our Td-PL results indicate that In surface substitution can suppress polaron formation by inhibiting electron–phonon coupling, thereby releasing more surface

holes. The conclusion from our Td-PL measurements that carrier recombination is enhanced at higher temperatures is not in contradiction with the temperature-dependent behaviors observed in the literature. In fact, although elevated temperatures generally promote charge transport rates, the associated intensified thermal motion also enhances electron–phonon coupling, which facilitates nonradiative carrier recombination and shortens carrier lifetimes^{52, 53, 54}. Therefore, how to regulate the temperature dependence of carrier lifetimes in order to achieve higher carrier utilization efficiency remains one of the key issues in the field of energy conversion, particularly in photovoltaics and photothermal catalysis^{55, 56, 57}.

In summary, we have revised the relevant sections of the manuscript from both of the aforementioned aspects to provide a clearer and more rigorous analysis of the Td-PL data.

Changes to the revised manuscript are shown below.

Main Manuscript (Results-Suppression of HPs and Carrier Property Analysis):

Page 14: Additionally, temperature-dependent photoluminescence (Td-PL) spectroscopy was employed to probe the underlying mechanism of electron-phonon coupling in the suppression of HPs. It was observed that the intensity of the PL peak decreased with increasing temperature, suggesting that the phonon-assisted recombination of photo-generated carriers is suppressed at lower temperatures (Fig. 5b, Supplementary Fig. 27b)³⁷. The electron-phonon coupling strength was quantitatively evaluated using the Huang-Rhys factor (S), which was extracted by fitting the temperature-dependent full width at half maximum (FWHM) of the PL peak using Equation 1^{38,39}.

$$\text{FWHM}=2.36\sqrt{S}E_{\text{phonon}}\sqrt{\coth\frac{E_{\text{phonon}}}{2k_{\text{B}}T}} \quad (1)$$

where S and E_{phonon} represent the electron-phonon coupling strength and the phonon energy, respectively. The S for In/BiVO₄ was calculated to be 48.9, whereas for BiVO₄, it was considerably higher at 96.5 (Supplementary Fig. 27). The significantly reduced S value for In/BiVO₄ indicates a marked suppression of carrier-phonon coupling.

Comment 7: The Td-PL results presented in the manuscript may suggest reduced polaron hopping, but they do not provide direct evidence that the observed effects stem from HPs rather than EPs. Additional experimental data are necessary to support the claim of enhanced HP hopping in contrast to EP hopping.

Response: We sincerely appreciate your professional suggestions, which are highly valuable for our interpretation and analysis of polarons. We agree with your opinion that additional experimental data should be added to further analyze and explain the suppression of polaron formation. However, it should be noted that due to current instrumental limitations, it is difficult to experimentally separate the signals of electrons and holes directly^{58, 59}. Nevertheless, theoretical calculations may provide detailed information on the electronic or hole carrier polaron structures, particularly regarding their local characteristics and mobility^{8, 9, 10, 43, 44}. Additionally, it is worth mentioning that the analysis of polaron hopping activation energy (E_h) provides further support for our strategy for suppressing HP formation^{43, 44, 60}. In summary, from the theoretical perspective, we have added DFT calculations including polaron formation energy, differential charge density, Bader charge analysis, bond length variations, and DOS to reveal the impact of our strategy on the properties of EP. From the experimental perspective, we have also added the measurement of E_h to confirm that our strategy has little effect on EP but mainly suppresses HP formation to release more holes (please refer to **Supplementary Fig. 28**). The specific revisions are as follows:

1. Supplementary DFT calculations and analysis

Based on your valuable comments in **Reviewer #4-Comment 1** and **Reviewer #4-Comment 3**, we have added DFT calculations for EP to analyze the impact of our substitution strategy on EP formation. The results show that In substitution leads to only a slight increase in the formation energy of EP, but a significant increase in the formation energy of HP, indicating that HP formation is effectively suppressed. Moreover, the analysis of differential charge density, Bader charge, bond length changes, and DOS all strongly support the view that our In substitution has a minimal effect on EP, but significantly inhibits HP formation. You can find the detailed

supplementary analysis in Responses to Reviewer #4-Comment 1-Response and Responses to Reviewer #4-Comment 3-Response.

2. Experimental measurement of polaron hopping activation energy

EP typically transfers from one polarized site to another through a hopping mechanism, and the energy required for this process is known as the polaron hopping activation energy (E_h). This activation energy can be derived from EIS measurements at different temperatures. We have measured EIS curves at various temperatures and obtained temperature-dependent electronic conductivity curves (Supplementary Fig. 28a and b). The results show that the conductivity of both photoanodes increases significantly with rising temperature, which confirms the hopping conduction mechanism of EP. Subsequently, we derived the E_h from the $\ln(\sigma T)-1/T$ plot. The results indicate that the E_h of In/BiVO₄ (529 meV) is almost the same as that of BiVO₄ (543 meV), suggesting that the EP hopping is not significantly promoted. In combination with DFT calculations, EPR results, Td-PL fitting, and SPV measurements, it can be concluded that our In substitution strategy mainly targets HP sites and effectively suppresses hole polaron formation. This is also the reason for the observed changes in the EPR spectra and the reduction in the S factor obtained from Td-PL fitting. In contrast, our strategy has little effect on EP sites and does not significantly promote the hopping behavior of EP. This is because the Bi sites we modulated are associated with HP formation and do not have a significant impact on the EP sites (V sites).

In general, we have achieved selective substitution of Bi by In through the LPCE strategy, modulated the electronic structure of HP sites, and effectively suppressed the formation of hole polarons, thereby releasing more holes. The supplementary DFT and experimental data further confirm that our strategy has little influence on EP. We believe that this additional analysis of EP enhances the scientific rigor and comprehensiveness of our results. Thank you once again for your insightful comments.

Changes to the revised manuscript are shown below.

Main Manuscript (Results-Suppression of HPs and Carrier Property Analysis):

Page 14: It should be noted that, from an experimental perspective, it is currently difficult to directly distinguish between HP and EP. However, the polaron hopping activation energy (E_h) indirectly reveals that In substitution has little effect on the hopping of EPs, indicating that our strategy primarily regulates HP sites and suppresses the formation of HPs (Supplementary Fig. 28).

Supplementary Information:

Supplementary Fig. 28. Temperature-dependent conductance of photoanodes.

The temperature-dependent conductance of **a** BiVO₄ and **b** In/BiVO₄ photoanodes. **c** The $\ln(\sigma T)$ - $1/T$ plots of BiVO₄ and In/BiVO₄ photoanodes.

Page 34: The conductivity of BiVO₄ and In/BiVO₄ photoanodes increases significantly with rising temperature, which confirms the hopping conduction mechanism of EP. The E_h of In/BiVO₄ (529 meV) is almost the same as that of BiVO₄ (543 meV), suggesting that the EP hopping is not significantly promoted.

References

1. Shulman R, *et al.* Determination of the iron-sulfur distances in rubredoxin by x-ray absorption spectroscopy. *Proceedings of the National Academy of Sciences* **72**, 4003-4007 (1975).
2. Ravel B, Newville M. ATHENA, ARTEMIS, HEPHAESTUS: data analysis for X-ray absorption spectroscopy using IFEFFIT. *Synchrotron Radiation* **12**, 537-541 (2005).
3. Pattengale B, Ludwig J, Huang J. Atomic insight into the W-doping effect on carrier dynamics and photoelectrochemical properties of BiVO₄ photoanodes. *The Journal of Physical Chemistry C* **120**, 1421-1427 (2016).
4. Cerrato E, Paganini MC, Giamello E. Photoactivity under visible light of defective ZnO investigated by EPR spectroscopy and photoluminescence. *J Photoch Photobio A* **397**, 112531 (2020).

5. Cooper JK, *et al.* Role of hydrogen in defining the N-Type character of BiVO₄ photoanodes. *Chemistry of Materials* **28**, 5761-5771 (2016).
6. Venkatesan R, *et al.* Dielectric behavior, conduction and EPR active centres in BiVO₄ nanoparticles. *Journal of Physics and Chemistry of Solids* **74**, 1695-1702 (2013).
7. Wiktor J, Ambrosio F, Pasquarello A. Role of polarons in water splitting: The case of BiVO₄. *ACS Energy Letters* **3**, 1693-1697 (2018).
8. Zhang WR, *et al.* Unconventional relation between charge transport and photocurrent via boosting small polaron hopping for photoelectrochemical water splitting. *ACS Energy Letters* **3**, 2232-2239 (2018).
9. Wu F, Ping Y. Combining Landau–Zener theory and kinetic Monte Carlo sampling for small polaron mobility of doped BiVO₄ from first-principles. *Journal of Materials Chemistry A* **6**, 20025-20036 (2018).
10. Liu T, Cui M, Dupuis M. Hole polaron transport in bismuth vanadate BiVO₄ from hybrid density functional theory. *The Journal of Physical Chemistry C* **124**, 23038-23044 (2020).
11. Cooper JK, *et al.* Electronic structure of monoclinic BiVO₄. *Chemistry of Materials* **26**, 5365-5373 (2014).
12. Kweon KE, Hwang GS. Structural phase-dependent hole localization and transport in bismuth vanadate. *Physical Review B* **87**, 205202 (2013).
13. Liu B, *et al.* A standalone bismuth vanadate-silicon artificial leaf achieving 8.4% efficiency for hydrogen production. *Nature Communications* **16**, 2792 (2025).
14. Zhao R, *et al.* Dual Mo-doping in BiVO₄/FeCoNiO_x photoanode enables near-theoretical photocurrent density via synergistic bulk-surface engineering for solar water splitting. *Advanced Science* **12**, e09037 (2025).
15. Liu B, *et al.* A BiVO₄ photoanode with a VO_x layer bearing oxygen vacancies offers improved charge transfer and oxygen evolution kinetics in photoelectrochemical water splitting. *Angewandte Chemie International Edition* **62**, 202217346 (2023).
16. Cui J, *et al.* 2D bismuthene as a functional interlayer between BiVO₄ and NiFeOOH for enhanced oxygen-evolution photoanodes. *Advanced Functional Materials* **32**, 2207136 (2022).
17. Zhang B, *et al.* Unveiling the activity and stability origin of BiVO₄ photoanodes with FeNi oxyhydroxides for oxygen evolution. *Angewandte Chemie International Edition* **59**, 18990-18995 (2020).

18. Jian J, *et al.* Embedding laser generated nanocrystals in BiVO₄ photoanode for efficient photoelectrochemical water splitting. *Nature Communications* **10**, 2609 (2019).
19. Gao RT, Wang L. Stable cocatalyst-free BiVO₄ photoanodes with passivated surface states for photocorrosion inhibition. *Angewandte Chemie International Edition* **59**, 23094-23099 (2020).
20. Gao RT, *et al.* Pt-Induced defects curing on BiVO₄ photoanodes for near-Threshold charge separation. *Advanced Energy Materials* **11**, 2102384 (2021).
21. Yang JW, *et al.* High-efficiency unbiased water splitting with photoanodes harnessing polycarbazole hole transport layers. *Energy & Environmental Science* **17**, 2541-2553 (2024).
22. Yang J, *et al.* Fe-N co-doped BiVO₄ photoanode with record photocurrent for water oxidation. *Angewandte Chemie International Edition* **64**, 202416340 (2025).
23. Chen X, *et al.* Tailoring carrier dynamics of BiVO₄ photoanode via dual incorporation of Au and Co(OH)_x cooperative modification for photoelectrochemical water splitting. *Advanced Functional Materials* **35**, 2416091 (2024).
24. Xin Y, *et al.* Enhanced photocatalytic efficiency through oxygen vacancy-driven molecular epitaxial growth of metal-organic frameworks on BiVO₄. *Advanced Materials* **37**, 2417589 (2025).
25. Gao R-T, *et al.* Photoelectrochemical production of disinfectants from seawater. *Nature Sustainability* **8**, 672-681 (2025).
26. Zhang X, *et al.* Engineering single-atomic Ni-N₄-O sites on semiconductor photoanodes for high-performance photoelectrochemical water splitting. *Journal of the American Chemical Society* **143**, 20657-20669 (2021).
27. Luo W, *et al.* Phthalocyanine grafting strategy induces strong intrinsic electric fields and molecule-edge carrier transport pathways for photoelectrochemical water splitting. *Angewandte Chemie International Edition* **64**, 202504589 (2025).
28. Jin B, *et al.* A two-photon tandem black phosphorus quantum dot-sensitized BiVO₄ photoanode for solar water splitting. *Energy & Environmental Science* **15**, 672-679 (2022).
29. Saad AM, *et al.* Boosting water oxidation kinetics of BiVO₄ through a metal-organic co-catalyst enriched with phosphate groups (Co,Fe-NTMP): Insights from LMCT mechanism and DFT study. *Applied Catalysis B: Environment and Energy* **370**, 125163 (2025).
30. Lin J, *et al.* Nitrogen-doped cobalt-iron oxide cocatalyst boosting photoelectrochemical water splitting of BiVO₄ photoanodes. *Applied Catalysis B: Environmental* **320**, 121947 (2023).

31. Zhang B, *et al.* Nitrogen-incorporation activates NiFeO_x catalysts for efficiently boosting oxygen evolution activity and stability of BiVO₄ photoanodes. *Nature Communications* **12**, 6969 (2021).
32. Zhang Z, Huang X, Zhang B, Bi Y. High-performance and stable BiVO₄ photoanodes for solar water splitting via phosphorus–oxygen bonded FeNi catalysts. *Energy & Environmental Science* **15**, 2867-2873 (2022).
33. Zhong X, *et al.* In³⁺-doped BiVO₄ photoanodes with passivated surface states for photoelectrochemical water oxidation. *Journal of Materials Chemistry A* **6**, 10456-10465 (2018).
34. Resasco J, *et al.* TiO₂/BiVO₄ nanowire heterostructure photoanodes based on type II band alignment. *ACS central science* **2**, 80-88 (2016).
35. Shaddad MN, *et al.* Cooperative catalytic effect of ZrO₂ and α-Fe₂O₃ nanoparticles on BiVO₄ photoanodes for enhanced photoelectrochemical water splitting. *ChemSusChem* **9**, 2779-2783 (2016).
36. Xue KLaD. Estimation of electronegativity values of elements in different valence states. *The Journal of Physical Chemistry A* **110**, 11332-11337 (2006).
37. Pekar S. Local quantum states of electrons in an ideal ion crystal. *Zhurnal Eksperimentalnoi I Teoreticheskoi Fiziki* **16**, 341-348 (1946).
38. Reticcioli M, *et al.* Interplay between adsorbates and polarons: CO on rutile TiO₂ (110). *Physical Review Letters* **122**, 016805 (2019).
39. Tanner AJ, *et al.* Photoexcitation of bulk polarons in rutile TiO₂. *Physical Review B* **103**, L121402 (2021).
40. Tanner AJ, *et al.* Polaron-adsorbate coupling at the TiO₂ (110)-carboxylate interface. *The journal of physical chemistry letters* **12**, 3571-3576 (2021).
41. Reticcioli M, *et al.* Polaron-driven surface reconstructions. *Physical Review X* **7**, 031053 (2017).
42. Wiktor J, Pasquarello A. Electron and hole polarons at the BiVO₄–water interface. *ACS applied materials & interfaces* **11**, 18423-18426 (2019).
43. Wu H, *et al.* Low-bias photoelectrochemical water splitting via mediating trap states and small polaron hopping. *Nature Communications* **13**, 6231 (2022).
44. Wang J, *et al.* Unconventional substitution for BiVO₄ to enhance photoelectrocatalytic

- performance by accelerating polaron hopping. *ACS Applied Materials & Interfaces* **15**, 14359-14368 (2023).
45. Qiu W, *et al.* Freeing the polarons to facilitate charge transport in BiVO₄ from oxygen vacancies with an oxidative 2D precursor. *Angewandte Chemie International Edition* **58**, 19087-19095 (2019).
 46. Lu Y, *et al.* Boosting charge transport in BiVO₄ photoanode for solar water oxidation. *Advanced Materials* **34**, 2108178 (2022).
 47. Sun L, *et al.* Disentangling the role of small polarons and oxygen vacancies in CeO₂. *Physical Review B* **95**, 245101 (2017).
 48. Di Valentin C, Selloni A. Bulk and surface polarons in photoexcited anatase TiO₂. *The journal of physical chemistry letters* **2**, 2223-2228 (2011).
 49. Wu H, *et al.* Polaron-mediated transport in BiVO₄ photoanodes for solar water oxidation. *ACS Energy Letters* **8**, 2177-2184 (2023).
 50. Wang H, *et al.* Giant electron–hole interactions in confined layered structures for molecular oxygen activation. *Journal of the American Chemical Society* **139**, 4737-4742 (2017).
 51. Li H, *et al.* Hole polaron-mediated suppression of electron–hole recombination triggers efficient photocatalytic nitrogen fixation. *Advanced Materials* **36**, 2408778 (2024).
 52. Zhang L, *et al.* Dynamics of photoexcited small polarons in transition-metal oxides. *The journal of physical chemistry letters* **12**, 2191-2198 (2021).
 53. Wright AD, *et al.* Electron–phonon coupling in hybrid lead halide perovskites. *Nature communications* **7**, 11755 (2016).
 54. Zhou X, *et al.* Temperature dependence of electron–phonon interactions in gold films rationalized by time-domain ab initio analysis. *The Journal of Physical Chemistry C* **121**, 17488-17497 (2017).
 55. Li W, *et al.* Time-domain ab initio analysis rationalizes the unusual temperature dependence of charge carrier relaxation in lead halide perovskite. *ACS Energy Letters* **3**, 2713-2720 (2018).
 56. Su H, *et al.* Hollow nanoreactor with MoS₂ encapsulated in ZnIn₂S₄: Spatially oriented distribution of MoS₂ improves photothermal hydrogen production activity. *Chemical Engineering Journal* **520**, 166165 (2025).
 57. Li Z, *et al.* Temperature-dependent thermal behavior of BTP-4F-12-based organic solar cells. *Nano Energy* **140**, 111043 (2025).

58. Nie W, *et al.* Light-activated photocurrent degradation and self-healing in perovskite solar cells. *Nature Communications* **7**, 11574 (2016).
59. Liu C, *et al.* Direct spectroscopic observation of the hole polaron in lead halide perovskites. *The journal of physical chemistry letters* **11**, 6256-6261 (2020).
60. He B, *et al.* Enhanced bulk and interfacial charge transfer in Fe:VOPO₄ modified Mo:BiVO₄ photoanodes for photoelectrochemical water splitting. *eScience* **5**, 100242 (2025).

Responses to the referees' comments

Dear Reviewers:

We sincerely thank you for your valuable and insightful comments on our manuscript titled “Leveraging surface hole polaron site tuning to enhance carrier separation in BiVO₄ photoanodes” Your thoughtful feedback has been instrumental in helping us refine the presentation, clarify the scientific arguments, and improve the overall quality of the work. We have carefully considered each of your comments and made corresponding revisions to the manuscript, which have significantly enhanced its accuracy and readability. We hope that the revised version meets your expectations and addresses the concerns raised.

Thank you once again for your time, effort, and constructive guidance.

Sincerely,

Fang He

Responses to Reviewer #1:

Reviewer's comments:

The authors have satisfactorily answered all of my comments and concerns.

Authors:

Thank you for your positive feedback and are pleased that our responses addressed your concerns satisfactorily. Your insightful comments and concerns have been fully incorporated into the revised manuscript. Your constructive suggestions have significantly enhanced the quality of this manuscript. We sincerely appreciate the time and expertise you dedicated to the review process.

Responses to Reviewer #2:

Reviewer's comments:

All issues addressed by this reviewer were well revised and explained.

Authors:

We sincerely thank you for your recognition of the revisions made in response to your comments. It is encouraging to know that the changes have been well addressed and clearly explained. Your constructive suggestions were of great value to us, and we are truly appreciative of your time and effort in reviewing our manuscript.

Responses to Reviewer #3:

Reviewer's comments:

The authors provide detailed and satisfactory responses to the comments raised, which makes the manuscript more concise and clear. Therefore, I am in favor of its publication.

Authors:

We sincerely thank you for your positive assessment and support for the publication of our work. The revisions made in response to your comments have significantly enhanced the manuscript's clarity and overall readability. Your constructive input has been extremely helpful in refining the manuscript. We are deeply grateful for your professional review and valuable feedback.

Responses to Reviewer #4:

Reviewer's comments:

I appreciate the authors' extensive effort in revising the manuscript and providing additional calculations and characterizations. However, after carefully evaluating the rebuttal and revised version, I find that the major scientific concerns I raised remain largely unresolved.

1. Most of the authors' new explanations, formation energies, charge density maps, Bader charge analyses, DOS, and bond-length changes, are purely theoretical. While

these calculations are useful for conceptual support, they do not constitute direct experimental evidence for the proposed suppression of hole polarons (HPs) or the claimed enhancement of HP-mediated processes. The central mechanistic claim therefore remains insufficiently validated.

2. The rebuttal repeatedly states that separating hole vs. electron polarons experimentally is “difficult,” yet the manuscript continues to make strong claims about HP-dominated recombination and HP-specific suppression. The added EIS-derived activation energies do not experimentally isolate HP dynamics; they primarily reflect electron hopping. Thus, the key mechanistic assignment still lacks direct experimental grounding.

3. In BiVO_4 , small electron polarons typically dominate bulk transport limitations, whereas hole polarons localize strongly at Bi-O motifs and are known to be less mobile. This fundamental understanding is well documented. The manuscript’s central narrative that HPs are the primary limitation and require selective suppression does not convincingly introduce new physics. The responses emphasize literature consistency rather than presenting new experimental breakthroughs.

4. Although the authors provide extensive DFT evidence that In substitution increases HP formation energy, the manuscript still lacks direct observation of hole release from HP sites. As a result, the proposed “HP suppression” as the primary enhancement mechanism remains speculative.

5. While the topic is interesting and relevant to the BiVO_4 and polaron communities, the mechanistic claims are still not supported by sufficiently strong and direct experimental evidence. The revised manuscript appears to expand theoretical detail rather than resolve the scientific gaps.

In summary, I appreciate the authors’ effort, but the concerns regarding conceptual solidity and experimental validation remain unresolved. I therefore maintain my previous recommendation.

Authors:

We sincerely appreciate your thorough reading of our manuscript and your thoughtful comments. We fully understand the concerns you have raised regarding the conceptual reliability and experimental validation of our work, which mainly stem from two aspects: **(1)** the direct experimental evidence for the suppression of hole polaron (HP) formation, and **(2)** the experimental evidence for the efficient release of surface holes and the inhibition of surface recombination. In response, we have added additional experiments in the revised manuscript to address these issues. These include the use of **in-situ irradiation XPS (ISI-XPS)** and **femtosecond transient absorption spectroscopy (fs-TAS)** to further directly demonstrate the suppression of HP formation, as well as **the ratio of transient to steady-state photocurrent (i_0/i)** and **open-circuit potential (OCP)** measurements to further confirm the efficient release of surface holes and the inhibition of surface recombination.

We have carefully reviewed your concerns and hope that the revisions and additions we have made will help clarify and strengthen the key aspects of our study. We would be grateful if you could re-evaluate the updated version and consider whether the current presentation adequately resolves the identified issues. We remain open to further modifications based on your valuable feedback. Below is our point-by-point response to your comments.

Comment 1: Most of the authors' new explanations, formation energies, charge density maps, Bader charge analyses, Density of States, and bond-length changes, are purely theoretical. While these calculations are useful for conceptual support, they do not constitute direct experimental evidence for the proposed suppression of hole polarons (HPs) or the claimed enhancement of HP-mediated processes. The central mechanistic claim therefore remains insufficiently validated.

Response: We sincerely thank you for the insightful comments on the current experimental validation in our revised manuscript. We are pleased to see that you are satisfied with our response to the theoretical calculation issues (including formation

energy, charge density maps, Bader charge analysis, and density of states) raised in your first round of comments, which are essential for understanding the underlying mechanisms and providing conceptual support. Furthermore, we fully agree with your perspective that a comprehensive and in-depth experimental analysis is still necessary to clarify our core mechanistic claims.

Perhaps our description was not direct enough, which has led to your continued doubts, even though we have previously provided various methods to prove direct or indirect evidence for the suppression of HP formation. Experimental evidence including in-situ irradiation electron paramagnetic resonance (EPR) spectroscopy, temperature-dependent photoluminescence (Td-PL) spectroscopy, surface photovoltage (SPV) spectrum, electrochemical measurements, and time-resolved photoluminescence (TRPL) spectroscopy are used to prove the suppression of HP formation. These are supported by theoretical analysis from density functional theory (DFT) calculations, and the results are mutually consistent. However, we believe that simply emphasizing or highlighting the existing data is insufficient to dispel your doubts or further improve our manuscript. To further comprehensively and deeply demonstrate the suppression of HP formation, we have added **additional ISI-XPS (Fig. R5)** and **fs-TAS (Fig. R7)** measurements. The ISI-XPS measurements can directly analyze the formation of polarons at different polarized sites and provide complementary insights to the Td-PL results¹. The fs-TAS can directly observe the polaron formation behavior, thereby providing further direct experimental evidence for the suppression of HP formation². Combining these results with those in the manuscript, we have summarized and systematically analyzed the direct experimental evidence for HP suppression, which includes the following four aspects:

1. In-situ irradiation EPR spectra

We conducted in-situ irradiation EPR measurements under irradiated and non-irradiated conditions, which serve as direct experimental evidence for the generation and suppression of polarons. Specifically, in the dark, both BiVO₄ and In/BiVO₄ photoanodes exhibit an EPR signal at $g=2.003$, which is attributed to free electrons (**Fig.**

R1a). Under illumination, additional EPR signals are observed in the BiVO₄ sample (**Fig. R1b**). These include peaks at $g = 1.945$ and $g = 1.960$, attributed to the self-trapping of photogenerated electrons, and a peak at $g = 2.039$, attributed to the self-trapping of photogenerated holes. These spectral features are clear evidence of the formation of both EPs and HPs. In contrast, the EPR signals in In/BiVO₄ are significantly reduced, indicating that the generation of HPs is effectively suppressed under illumination. The comparison of EPR results before and after irradiation provides a clear experimental demonstration of the suppression of HP formation. The use of EPR to analyze polaron behavior is a commonly accepted experimental approach in the literature, with relevant studies covering a variety of metal oxides such as ZnO, TiO₂ and KTaO₃. For example, Cerrato et al. clearly detected trapped electron and trapped hole centers formed via photogenerated carriers in ZnO, using EPR spectroscopy (**Fig. R2a**)³. Yang et al. identified electron polaron self-trapping in the rutile-phase TiO₂ through EPR measurements (**Fig. R2b**)⁴. Li et al. demonstrated, by observing the EPR spectral changes under irradiation, the capture of photogenerated holes by surface lattice oxygen in KTaO₃, leading to polaron formation (**Fig. R2c and d**)⁵. Similarly, Laguta et al. identified two types of shallow hole centers via EPR under irradiation (**Fig. R2e**)⁶. In the same context, EPR has been widely used for the direct characterization of polarons, and it can be conveniently combined with other characterization techniques^{7, 8, 9, 10}.

Fig. R1 EPR spectroscopy of BiVO₄ and In/BiVO₄ photoanodes under **a** dark and **b** illumination conditions.

Fig. R2 **a** EPR spectra of ZnO. Ref. (3): *Journal of Photochemistry and Photobiology A: Chemistry* 397 (2020): 112531. **b** Photoinduced EPR spectrum from a TiO₂ (rutile) crystal. Ref. (4): *Physical Review B* 87, 125201 (2013). EPR spectra of KTaO₃ in the **c** light and **d** dark state. Ref. (5): *Advanced Materials* 36, 2408778 (2024). **e** EPR spectra of KTaO₃. Ref. (6): *Journal of Applied Physics* 93, 6056-6064 (2003). **f** Differential continuous wave spectra of TiO₂ nano-objects obtained under monochromatic excitation. Ref. (7): *The Journal of Physical Chemistry C* 111, 14597-14601 (2007).

2. Td-PL spectra

Td-PL is a powerful tool for revealing the state of polarons by evaluating the strength of electron-phonon coupling. A stronger electron-phonon coupling implies a stronger self-trapping of polarons, and the coupling strength can be quantitatively assessed using the Huang-Rhys factor¹¹. Specifically, we fitted the full width at half maximum (FWHM) of the Td-PL spectra using Equation R1, and thus obtained the S values^{12, 13}.

$$\text{FWHM} = 2.36 \sqrt{S} E_{\text{phonon}} \sqrt{\coth \frac{E_{\text{phonon}}}{2k_b T}} \quad (\text{R1})$$

The results show that the S value for In/BiVO₄ is 48.9, while that for BiVO₄ is significantly higher at 96.5 (**Fig. R3c** and **d**). This suggests that In/BiVO₄ exhibits

weaker polaron localization behavior. Although both electron and hole polarons exist in the photoanode, our strategy focuses on precisely modulating the surface hole-polaron sites, with minimal impact on the electron-polaron sites. Therefore, the decrease in electron-phonon coupling strength is mainly attributed to the suppression of HP formation. We consider this result as one of the direct experimental evidences for HP suppression. The use of Td-PL to analyze polaron dynamics is also a widely adopted method in the current field. For example, Li et al. demonstrated the enhanced electron-phonon coupling in KTaO_3 using Td-PL spectroscopy and the S factor, thereby confirming the formation of HP (Fig. R4a)⁵. Similarly, McCall et al. analyzed the electron-phonon coupling via Td-PL spectroscopy and proposed a model in which electron-phonon coupling induces small polarons and facilitates the trapping of charge carriers (Fig. R4b)¹⁴. A number of studies have also employed Td-PL spectroscopy to investigate electron-phonon coupling and polaron behavior, indicating that this technique is widely recognized and utilized in the field (Fig. R4c-e)^{15, 16, 17}.

Fig. R3 Td-PL spectroscopy of **a** BiVO_4 and **b** In/BiVO_4 photoanodes. Temperature dependence of the full width at half maximum (FWHM) of **c** BiVO_4 and **d** In/BiVO_4 photoanode with the Huang-Rhys factor (S) and phonon energy (E_{phonon}) extracted from fitting the data.

Fig. R4 a The temperature-dependence PL spectra and FWHM of the STE emission as a function of temperature of KTaO_3 nanosheets. Ref. (5): *Advanced Materials* 36, 2408778 (2024). **b** The temperature-dependence PL spectra and FWHM vs temperature of $\text{Cs}_3\text{Sb}_2\text{I}_9$ and $\text{Rb}_3\text{Sb}_2\text{I}_9$. Ref. (14): *Chemistry of Materials* 29, 4129-4145 (2017). **c** PL spectra at different temperatures from 50 to 373 K, temperature-dependent peak energy and the fitting results of $\text{ZnCuInS}/\text{ZnSe}/\text{ZnS}$ QDs with different particle sizes. Ref. (15): *The Journal of Physical Chemistry C* 117, 19288-19294 (2013). **d** Temperature-dependent PL spectra analysis. of $\alpha\text{-HgI}_2$. Ref. (16): *Journal of Luminescence* 237, 118161 (2021). **e** Temperature dependence of the PL of the sample TiO_2 (500°C). Ref. (17): *Journal of Applied Physics* 108, 113502 (2010).

3. ISI-XPS

EPs and HPs are generated through the self-trapping of photogenerated carriers during photoexcitation, which makes it challenging to directly resolve their localized sites. Fortunately, in BiVO₄ material, electrons and holes localize at different sites (V sites and Bi sites), allowing us to analyze the characteristics of EPs and HPs using ISI-XPS^{18, 19}. In this work, by comparing the XPS spectra before and after irradiation, we distinguished the strength of electron-phonon coupling at different atomic sites. It was found that after In doping, the In site does not exhibit significant electron-phonon coupling, while the electron-phonon coupling at the Bi site is slightly weakened, and the electron-phonon coupling at the V site remains largely unchanged (**Fig. R5**). Combined with the results from Td-PL, we were able to further differentiate the changes in electron-phonon coupling at different polaron sites. The specific results are as follows:

XPS is a well-established method for determining the atomic states on the surface. In the state of photoexcited polarons, the electron-phonon interaction leads to changes in the peak width of XPS signals¹. Here, we compared the ISI-XPS spectra of BiVO₄ and In/BiVO₄ under both dark and illuminated conditions. We observed a significant increase in the FWHM of the Bi 4f and V 2p spectra in BiVO₄ after photoexcitation. On one hand, this indicates the formation of polarons associated with electron-phonon coupling; on the other hand, it confirms that polarons are formed at both the Bi and V sites. In combination with previous studies and our results, this can be attributed to HPs at the Bi site and EPs at the V site, respectively (**Fig. R5a and b**). Notably, in the In/BiVO₄ photoanode, no significant electron-phonon coupling is observed at the In site, the electron-phonon coupling at the Bi site is slightly weakened, and the electron-phonon coupling at the V site remains largely unchanged (**Fig. R5c-e**). This suggests that the introduction of In suppresses the formation of HPs. By combining the quantitative analysis of electron-phonon coupling strength from Td-PL and the qualitative analysis of different sites from ISI-XPS, we jointly confirm that the formation of HPs is suppressed, while the formation of EPs remains largely unchanged.

It should be noted that the formation of polarons can be caused by various factors, such as optical excitation, doping, and defects, all of which may lead to charge self-trapping and the formation of polarons. However, in this manuscript, we mainly focus on the polarons generated through carrier self-trapping during the process of optical excitation in metal oxides. Therefore, an in-situ illumination environment, provided by ISI-XPS, is essential for the analysis of these polarons. A similar approach has been widely adopted in the literature. For example, Maslakov et al. utilized ISI-XPS to investigate localized charge induced by irradiation in CeO₂ thin films, leading to the reduction of Ce⁴⁺ to Ce³⁺, and observed an increased broadening of the Ce 5s and Ce 4s XPS peaks (**Fig. R6a**)²⁰. This localized charge defect is essentially a type of polaron. Li et al. used ISI-XPS to observe the electron transfer direction in a ZnIn₂S₄ and carbon nanofibers heterostructure under optical excitation, and found shifts in the XPS peak positions, as well as changes in the peak area and broadening (**Fig. R6b**)²¹. The increased broadening of the Zn and In XPS peaks may result from electrons being transferred to the carbon nanofibers, leaving behind hole carriers in ZnIn₂S₄. Similarly, Li et al. employed ISI-XPS to analyze the effect of optical excitation on electron transfer, revealing an increase in the peak area of Cu XPS spectra, which is attributed to Cu ions accepting more electrons (**Fig. R6c**)²². This electron accumulation is fundamentally similar to the localization of electrons in electron polarons. In addition to these methods, the use of XPS to analyze electron-phonon coupling, and thus infer polaron behavior, has also been extensively reported. Chaudhuri et al. observed a strong peak corresponding to V⁵⁺ and a shoulder peak corresponding to V⁴⁺ in the V 2p XPS spectra of BiVO₄, which may be due to the formation of polaronic states and associated local lattice distortions (**Fig. R6d**)²³. Wang et al. observed in the XPS spectra of NiNb₂O₆ with both columbite and rutile structures the Ni²⁺ peaks and a shoulder peak on the low-binding-energy side, which is typically indicative of the formation of EPs (**Fig. R6e**)²⁴.

In summary, we have utilized ISI-XPS to analyze the broadening of polaronic sites, thereby revealing the impact of In substitution on the electron-phonon coupling strength

and polaron formation at different sites. Combined with the results from Td-PL, this provides strong evidence that the formation of HPs is suppressed.

Fig. R5 ISI-XPS high-resolution spectra of **a** Bi 4f and **b** V 2p in BiVO₄ photoanode. ISI-XPS high-resolution spectra of **c** Bi 4f, **d** V 2p and **e** In 3d in In/BiVO₄ photoanode. **f** Difference in binding energy of elements at different polaron sites before and after irradiation.

Fig. R6 a XPS data of CeO₂ thin films and bulk samples before and after ¹²⁹Xe²³⁺ irradiation. Ref. (20): Applied Surface Science 448, 154-162 (2018). **b** The main parameters of XPS images in ZnIn₂S₄ and carbon nanofibers. Ref. (21): Applied Catalysis B: Environment and Energy 356, 124223 (2024). **c** The ISI-XPS spectra of Cu 2p in CuPcS/NMF-LDHs. Ref. (22): Journal of Materials Chemistry A 12, 13168-13180 (2024). **d** V2p_{3/2} XPS spectra of BiVO₄ sample. Ref. (23): J Physical Review B 97, 195150 (2018). **e** Ni 2p XPS spectra of columbite and rutile NiNb₂O₆, as well as the corresponding enlarged Ni 2p_{3/2} spectra. Ref. (24): Journal of Materials Chemistry C 8, 16107-16112 (2020).

4. fs-TAS

Finally, we employed fs-TAS to provide information on polaron dynamics and formation times in a time-resolved manner. In fs-TAS, the system is first excited by a femtosecond pump pulse, and then the optical absorption in the UV-visible region is monitored as a function of time delay using a probe pulse¹⁰. This allows us to reflect the time evolution between different ground and excited states after photoexcitation. In the kinetic model of BiVO₄, the formation of HPs is associated with a very short decay time constant, while the formation of EPs corresponds to a longer decay time constant, which is accompanied by a carrier recombination process². This enables us to distinguish the dynamics of HPs and EPs and analyze the formation behavior of HPs through kinetic fitting. The specific results are as follows:

Both BiVO₄ and In/BiVO₄ samples exhibit a negative absorption band between 400-450 nm, corresponding to ground-state bleaching (GSB), and a positive absorption band between 450-500 nm, corresponding to excited-state absorption (ESA) due to photo-induced carrier excitation (**Fig. R7a** and **b**)²⁵. At the same time, both samples show a similar absorption decay process, which gradually decreases after 500 fs, corresponding to the carrier recombination process (**Fig. R7c** and **b**). Notably, compared to BiVO₄, the stronger positive absorption signal in In/BiVO₄ indicates a higher carrier concentration and more efficient carrier separation²⁶.

The self-trapping process of polarons is a transient phenomenon occurring on a very short timescale. Therefore, analyzing the behavior of charge carriers across different timescales through kinetic fitting is crucial for revealing the polaron formation mechanism. We fitted the kinetic decay curves at 515 nm using a triple-exponential model. As shown in **Fig. R7e**, both BiVO₄ and In/BiVO₄ exhibit three distinct decay lifetimes. Notably, although the shortest time constant τ_1 of In/BiVO₄ (0.59 ps) is significantly longer than that of BiVO₄ (0.59 ps), both are still below 5 ps. Such ultrafast lifetime components are typically associated with the formation of HPs^{2, 5, 27, 28}. For instance, Ravensbergen et al. analyzed the carrier dynamics in BiVO₄ over femtosecond to microsecond timescales and developed a model, identifying hole trapping occurring on timescales less than 5 ps, which is indicative of HP formation (**Fig. R8a**)². In contrast, the formation and recombination of EPs occur on longer timescales (greater than 40 ps). Similarly, Grigioni et al. observed analogous lifetime components related to hole trapping in both BiVO₄ and WO₃/BiVO₄ at approximately 6 ps, which are essentially consistent with HPs (**Fig. R8b**)²⁷. Likewise, Li et al. observed τ_1 and τ_2 at around 50 ps and 1900 ps, respectively, in the dynamics of atomically ordered KTaO₃, corresponding to relaxation caused by defects and subsequent carrier recombination. However, in atomically disordered KTaO₃, an ultrafast τ_0 process was observed, which can be attributed to the direct trapping of photo-generated holes by lattice oxygen to form HPs (**Fig. R8c**)⁵. Additionally, Zhu et al. identified a component on the order of about 1 ps in fs-TAS kinetic fitting for NiO and Cu-doped NiO, which can be attributed to the formation of polarons through the interaction of injected holes with the lattice (**Fig. R8d**)²⁸. These studies collectively suggest that the ultrafast timescale dynamics can be associated with HP formation. Moreover, longer lifetimes indicate a slower kinetic process, and for HPs, this implies that the formation process is suppressed. Therefore, the significantly longer τ_1 observed in In/BiVO₄ indicates that HP formation is effectively suppressed compared to BiVO₄.

For the relatively longer timescale dynamics, τ_2 and τ_3 can be attributed to electron transfer and carrier recombination processes, respectively. Notably, In/BiVO₄ exhibits longer τ_2 and τ_3 compared to BiVO₄, suggesting a slower rate of carrier recombination and an extended carrier lifetime, which is beneficial for the participation of active carriers in the catalytic reaction (**Fig. R7e**)²⁹. This phenomenon is supported by numerous related reports. For example, Wang et al. observed a significantly increased lifetime component τ_3 in BiVO₄ loaded with CuS and NiFeCoO_x, indicating that these co-catalysts effectively prolong the lifetime of reactive carriers (**Fig. R9a-c**)³⁰. Xin et al. found a pronounced increase in the fs-TAS lifetime of BiVO₄ with MOF loading and a high density of oxygen vacancies (τ_1 increased from 54.21 ps to 71.77 ps, and τ_2 from 701.2 ps to 1479.82 ps), suggesting that MOF loading can passivate vacancies and extend the lifetime of photo-generated holes (**Fig. R9d**)³¹. Chen et al. observed that TiO₂/BiVO₄-0.5 ($\tau_1 = 143.7$ ps, $\tau_2 = 701.2$ ps) exhibits significantly longer delays than TiO₂/BiVO₄-4 ($\tau_1 = 4.3$ ps, $\tau_2 = 99.9$ ps), indicating that excessive TiO₂ content promotes carrier recombination and reduces the efficiency of electron transfer (**Fig. R9e**)³². Lin et al. observed a slower recombination rate ($\tau_2 = 121.44$ ps) and a longer average lifetime ($\tau_{\text{avg.}} = 117.67$ ps) for active charges in the polymer gradient heterojunction compared to other monolithic or stacked heterojunctions, demonstrating its superior spatial charge separation and transfer efficiency, which leads to the generation of more photo-induced electrons and holes and enhances electrochemical reactions (**Fig. R9f**)³³.

Based on the fs-TAS-related studies previously reported in the literature, our fs-TAS data clearly reveal the changes in ultrafast lifetime components, indicating an extended τ_1 corresponding to the HP formation process and confirming its suppression. The data also show a decrease in carrier recombination and an extension in carrier lifetime for the longer lifetime components. This further implies that surface holes in In/BiVO₄ can be released more effectively.

Fig. R7 Transient adsorption spectra of **a** BiVO₄ and **b** In/BiVO₄ photoanodes. Transient absorption spectra at a selection of delay times for **c** BiVO₄ and **d** In/BiVO₄ photoanodes. **e** The decay kinetics monitored at 515 nm of BiVO₄ and In/BiVO₄ photoanodes.

Fig. R8 **a** fs-TAS spectra and the carrier dynamics model of BiVO₄. Ref. (2): The Journal of Physical Chemistry C 118, 27793-27800 (2014). **b** TAS decay monitored, fitting parameters and proposed charge carrier transitions of BiVO₄ and WO₃/BiVO₄ heterojunction. Ref. (27): The Journal of Physical Chemistry C 119, 20792-20800 (2015). **c** Representative kinetic traces along with their global fitting results of the fs-TA spectra of KTaO₃ nanosheets. Ref. (5): Advanced Materials 36, 2408778 (2024). **d** TAS spectra and kinetic traces for NiO and Cu-doped NiO. Ref. (28): The Journal of Physical Chemistry C 125, 16049-16058 (2021).

Fig. R9 a-c Decay kinetics monitored at 480 nm of BiVO₄, BiVO₄/CuS and BiVO₄/CuS/NiFeCoO_x. Ref. (30): *Angewandte Chemie International Edition* 64, 202507259 (2025). **d** Normalized decay kinetic curves probed at 700 nm. Ref. (31): *Advanced Materials* 37, 2417589 (2025). **e** TA kinetics with different chemical additives. Ref. (32): *Journal of the American Chemical Society* 146, 9163-9171 (2024). **f** The kinetics of the fs-TA absorption bands. Ref. (33): *Advanced Materials* 37, 2415608 (2025).

In summary, we acknowledge that the current study does not provide direct observation of the polaron formation process or the dynamic behavior of hole carriers injected into the electrolyte. These phenomena are too microscopic and transient to be captured by existing experimental tools, and the development of more advanced instrumentation is required. However, we believe that the combination of experimental and theoretical evidence now offers a convincing and well-supported narrative. The direct experimental evidence from EPR, Td-PL, ISI-XPS, and FS-TAS, along with indirect experimental data such as SPV, TRPL, and electrochemical tests, and theoretical support from DFT calculations, together provide a multi-faceted and comprehensive demonstration of the suppression of HP formation. We appreciate your interest in this scientific issue, and we believe that the supplementary revisions made

to the manuscript in response to your suggestions will significantly enhance the completeness and depth of our analysis.

Changes to the revised manuscript are shown below.

Main Manuscript (Abstract):

Page 2: The electron paramagnetic resonance (EPR), temperature-dependent photoluminescence (Td-PL) spectroscopy, in situ irradiation XPS (ISI-XPS), and femtosecond time-resolved absorption spectroscopy (fs-TAS) all confirm the suppression of hole polaron formation.

Main Manuscript (Introduction):

Page 3: The suppression of HP formation has been experimentally verified using electron paramagnetic resonance (EPR), temperature-dependent photoluminescence (Td-PL) spectra, in situ irradiation XPS (ISI-XPS), and femtosecond transient absorption spectroscopy (fs-TAS).

Main Manuscript (Results- Suppression of HPs and Carrier Property Analysis):

Fig. 5: Analysis of HPs and carrier properties.

c ISI-XPS high-resolution spectra of In 3d in In/BiVO₄ photoanode. **d** Transient absorption spectra of In/BiVO₄ photoanode. **e** The decay kinetics monitored at 515 nm of BiVO₄ and In/BiVO₄ photoanodes.

Page 14-16: To reveal the changes in the strength of electron–phonon coupling at different polaron sites and thereby confirm the selective suppression of HPs, we conducted in situ irradiation XPS (ISI-XPS) measurements. During the self-trapping of photoexcited carriers to form polarons, the electron-phonon coupling leads to an increase in the peak broadening of XPS⁴⁰. In the case of the BiVO₄ photoanode, the Bi 4f and V 2p spectra exhibit a significant increase in FWHM under illumination, indicating the localization of both HPs and EPs at the Bi and V sites, respectively (Supplementary Fig. 28). In contrast, in the In/BiVO₄ photoanode, the In site shows no significant increase in FWHM, the Bi site exhibits a slight increase, and the V site still shows a clear broadening (Fig. 5c and Supplementary Fig. 28). This indicates that the In substitution weakens the electron-phonon coupling at HP sites, thus suppressing HP formation, while having a limited effect on EPs. Finally, we employed fs-TAS to provide time-resolved information on polaron dynamics and formation times. Both BiVO₄ and In/BiVO₄ show a negative absorption band in the 400-450 nm range, corresponding to ground-state bleaching (GSB), and a positive absorption band in the 450-500 nm range, corresponding to excited-state absorption (ESA) due to photo-induced carrier excitation (Fig. 5d, Supplementary Fig. 29)⁴¹. At the same time, both samples display a similar absorption decay process, which gradually decreases after 500 fs, corresponding to the carrier recombination process. Notably, compared to BiVO₄, the stronger positive absorption signal in In/BiVO₄ indicates a higher carrier concentration and more efficient carrier separation⁵. Subsequently, the kinetic decay curves at 515 nm were fitted using a three-exponential model. As shown in Fig. 5e, both BiVO₄ and In/BiVO₄ exhibit three distinct decay time constants. The shortest time constant (τ_1) is attributed to the formation of HPs, while τ_2 and τ_3 are associated with electron transfer and carrier recombination, respectively^{42, 43}. Notably, the τ_1 of In/BiVO₄ is 1.09 ps, significantly longer than that of BiVO₄ (0.59 ps). This suggests that the self-trapping process of HPs is slowed down, indicating that HP formation is effectively suppressed. In addition, the longer τ_2 and τ_3 in In/BiVO₄ suggest a reduced carrier recombination rate and an extended carrier lifetime, which is beneficial for more active carriers to participate in the catalytic reaction⁴⁴. In addition, the polaron hopping activation energy (E_h) indirectly indicates that In substitution has little effect on the hopping of EPs, suggesting that our strategy mainly modulates the HP sites and suppresses the formation of HPs (Supplementary Fig. 30). SPV spectroscopy results further

support this conclusion, with a markedly enhanced SPV signal indicating an increased accumulation of positive charges at the surface of the In/BiVO₄ photoanode (Fig. 5f)⁴⁵. This is attributed to the suppression of polaron formation, which facilitates the release of more hole carriers, resulting in a hole-enriched surface. The combined results from EPR, Td-PL, and SPV spectroscopy demonstrate that the surface substitution of In inhibits the formation of HPs by reducing electron-phonon coupling, thereby releasing more photogenerated holes. These findings are in good agreement with the predictions from the DFT calculations.

Main Manuscript (Discussion):

Page 19: EPR, Td-PL, ISI-XPS and fs-TAS provided clear evidence of the suppression of HPs, resulting in the release of more photogenerated hole carriers and a concomitant decrease in carrier recombination.

Main Manuscript (Methods- Material characterization):

Page 22: In-situ irradiation XPS measurements were performed using a ThermoFisher ESCALAB 250Xi X-ray photoelectron spectrometer. The system utilized a monochromatic Al-K α X-ray source (1486.6 eV) for excitation, operating under ambient conditions with illumination provided by a 300 W xenon lamp (PLS-SXE300E, Beijing Perfectlight, China) emitting light in the wavelength range of 320-780 nm. Prior to conducting the irradiation experiments, XPS spectra of all elemental components in the samples were collected in the dark as reference data. The femtosecond transient absorption spectroscopy (fs-TAS) is performed using the Ultrafast Helios system, which is coupled with a Coherent Astrella laser (>7 mJ, 800 nm, <100 fs, 1 kHz) and an OPerA-Solo OPA (240-2600 nm tunable), enabling high-sensitivity transient absorption measurements in the 320–1600 nm range with sub-14 fs time resolution.

Supplementary Information:

Supplementary Fig. 28. ISI-XPS analysis.

ISI-XPS high-resolution spectra of **a** Bi 4f and **b** V 2p in BiVO₄ photoanode. ISI-XPS high-resolution spectra of **c** Bi 4f and **d** V 2p in In/BiVO₄ photoanode. **e** Difference in binding energy of elements at different polaron sites before and after irradiation.

Supplementary Fig. 29. Fs-TAS analysis.

a Transient absorption spectra of BiVO₄ photoanode. Transient absorption spectra at a selection of delay times for **b** BiVO₄ and **c** In/BiVO₄ photoanodes.

Comment 2: The rebuttal repeatedly states that separating hole vs. electron polarons experimentally is “difficult,” yet the manuscript continues to make strong claims about HP-dominated recombination and HP-specific suppression. The added EIS-derived activation energies do not experimentally isolate HP dynamics; they primarily reflect

electron hopping. Thus, the key mechanistic assignment still lacks direct experimental grounding.

Response: We sincerely appreciate your valuable comments on our conclusions regarding the HP-dominated recombination mechanism and the selective suppression of HPs. We fully agree with your point that the activation energies derived from temperature-dependent EIS (Td-EIS) measurements cannot experimentally isolate the dynamics of HPs. However, while EIS is not capable of directly differentiating the dynamic behavior of EP and HP, it has been extensively employed in the literature to characterize the hopping of bulk EP. For example, He et al. utilized Td-EIS to observe a significantly reduced small polaron hopping activation energy in Fe:VOPO₄ modified Mo:BiVO₄, indicating a notable improvement in bulk charge transport (**Fig. R10a**)³⁴. Zhang et al. demonstrated the reduction in polaron hopping energy through temperature-dependent conductivity measurements (**Fig. R10b**)³⁵. Wu et al. investigated the promoting effect of P doping on polaron hopping in BiVO₄ using Td-EIS, thereby revealing its role in enhancing charge transport (**Fig. R10c**)³⁶. In addition, Rettie et al. studied polaron hopping behavior in W-doped BiVO₄ and Ti-doped Fe₂O₃ using similar approaches, as did Tang et al. in BiVO₄ with varying concentrations of oxygen vacancies, all of which highlighted the impact of polarons on bulk charge transport (**Fig. R10d-f**)^{37, 38, 39}. Therefore, our temperature-dependent EIS data provide indirect but consistent evidence that the EP sites in the In-doped BiVO₄ photoanode remain largely unaffected. Given that our modulation strategy targets the surface rather than the bulk, and specifically targets HP sites (Bi sites) rather than EP sites (V sites), this result provides an additional perspective to exclude the influence of the liquid phase cation exchange (LPCE) strategy on EP sites, thereby supporting the conclusion that our strategy mainly affects HP sites. Since surface reactions at the photoanode are predominantly hole-driven, we intentionally modulated the surface HP sites to release more hole carriers. This is consistent with the results from EPR, Td-PL, ISI-XPS, fs-TAS, SPV, and electrochemical tests, all of which collectively support the mechanism of HP formation suppression.

Fig. R10 a-f Activation energy fitting curves used in the reported works to analyze polaron hopping behavior. **a** Ref. (34): *EScience*, 2025, 5(1): 100242. **b** Ref. (35): *ACS Energy Letters*, 2018, 3(9): 2232-2239. **c** Ref. (36): *Nature Communications*, 2022, 13(1): 6231. **d** Ref. (37): *Applied Physics Letters*, 2015, 106(2). **e** Ref. (38): *The Journal of Physical Chemistry Letters*, 2016, 7(3): 471-479. **f** Ref. (39): *Advanced Functional Materials*, 2022, 32(18): 2110284.

It is worth noting that we fully acknowledge your point that direct experimental evidence is essential to clearly demonstrate the suppression of HP formation. In response, we have supplemented the study with ISI-XPS and fs-TAS measurements, which are capable of directly proving the suppression of HP formation. By integrating these results with the EPR and Td-PL data already presented in the manuscript, we are able to directly show that HP formation is significantly suppressed. A detailed discussion of these results can be found in the part of **Responses to Reviewer #4-Comment 1-Response** for your kind reference.

In addition, our DFT calculations provide theoretical support, while XPS, X-ray Absorption Fine Structure (XAFS), and a series of carrier property measurements offer indirect experimental evidence further confirming the suppression of HP formation.

Below, we will briefly discuss the theoretical evidence and the indirect experimental evidence.

1. Theoretical analysis

In the theoretical aspect, we used DFT to simulate the model of HP generation under light excitation (Fig. R11). The differential charge density combined with Bader analysis shows a strong accumulation of holes at the Bi sites, indicating the formation of HPs (Fig. R11a and b). In contrast, no significant hole localization is observed at the In substitution sites, proving that HP formation is suppressed at the In sites. A comparison of $E_p(\text{HP})$ further supports this conclusion (Fig. R11c). For the BiVO_4 -DFT model, the formation energy of hole polarons $E_p(\text{HP})$ is -0.24 eV, suggesting that holes tend to localize and form HPs at the Bi sites. However, the $E_p(\text{HP})$ for the In/BiVO_4 -DFT model (-0.05 eV) is significantly higher, indicating that HP formation is effectively suppressed at the In sites. The bond length changes also confirm this trend: the In-O bond shows no significant change after light excitation, while the Bi-O bond shortens by approximately 0.1 Å, implying the formation of HPs at the Bi sites and their suppression at the In sites (Fig. R11d).

Fig. R11 Charge density difference diagram of **a** BiVO_4 -DFT and **b** In/BiVO_4 -DFT with HP calculated by DFT. **c** The calculated polaronic stability of BiVO_4 -DFT and In/BiVO_4 -DFT, where E_p , E_s and E_c represent the contributions of polaron formation energy, lattice distortion energy, and

electronic energy, respectively. **d** Bi-O radial distribution functions in BiVO₄-DFT and In-O radial distribution functions in In/BiVO₄-DFT with and without HP.

2. Indirect experimental evidence

The indirect experimental evidence supporting the suppression of HP formation can be categorized into two main aspects. On one hand, XPS and XAFS results indicate that after In substitution, electrons in the photoanode transfer to the In sites, leading to a higher electron density at the In sites even in the absence of light excitation. This is beneficial for inhibiting HP generation under illumination, as HPs are essentially formed through hole self-trapping. The accumulation of more electrons at the In sites weakens the tendency for hole aggregation, thus reducing HP formation. On the other hand, carrier property measurements demonstrate that surface carriers are released more rapidly, and surface recombination is suppressed. Since surface recombination is mainly mediated by HPs, these observations indirectly support the suppression of HP formation, which in turn facilitates more efficient hole carrier release.

To elaborate, our XPS results show that after In substitution, the spectral peaks of Bi and V shift toward higher binding energies, indicating electron loss at the Bi and V sites (**Fig. R12a**). From the etch-XPS spectra, we observe that the binding energy of the In element shifts toward higher values with increasing etching depth, while the binding energies of Bi and V shift toward lower values, suggesting that In gradually loses electrons, and Bi and V gradually gain electrons as the etching depth increases (**Fig. R12b-d**). This implies that electrons transfer from Bi and V sites to In sites at the surface, which is further supported by the XAFS results. Specifically, the In K-edge XANES spectrum of In/BiVO₄ shows that the absorption edge shifts toward lower binding energies compared to that of In₂O₃ (**Fig. R12e**). This shift indicates a reduction in the valence state of In species, which can be attributed to the greater electronegativity of In relative to Bi, resulting in electron transfer to the InO₈ sites formed by In substitution.

In terms of carrier behavior, the SPV spectrum shows that, compared to BiVO₄ photoanodes, the In/BiVO₄ photoanode exhibits a significantly enhanced SPV signal, indicating a marked increase in surface positive charge (Fig. R12f). This suggests that hole carriers accumulate more on the surface, which can be attributed to the suppression of HP formation, thereby promoting hole release. The TRPL measurement reveals that the carrier lifetime of In/BiVO₄ (2.33 ns) is clearly longer than that of BiVO₄ (1.81 ns) (Fig. R12g). This further confirms that the suppression of HP formation at the surface leads to efficient hole release, thus reducing surface recombination and extending carrier lifetime.

Fig. R12 a XPS high-resolution spectroscopy of Bi 4f and V 2p in BiVO₄ and In/BiVO₄ photoanodes. Etch-XPS high-resolution spectra of b Bi 4f, c V 2p and d In 3d in In/BiVO₄ photoanode. e In K-edge XANES spectroscopy of In foil, In₂O₃ and In/BiVO₄. f SPV spectroscopy, g TRPL spectroscopy of BiVO₄ and In/BiVO₄ photoanodes.

In summary, we have adopted a comprehensive set of experimental techniques, offering both direct and indirect evidence from multiple angles to substantiate the

suppression of hole polaron formation. These include direct experimental evidence from EPR, Td-PL ISI-XPS, and fs-TAS, indirect evidence from XPS, XAFS, and carrier property tests, as well as theoretical support from DFT calculations. Collectively, these data strongly support the conclusion that In modification significantly suppresses the formation of HPs. We have also revised the mechanistic statements in the manuscript to be more precise and clearer in accordance with your suggestion. We sincerely appreciate your valuable feedback and believe that the supplement and revision of the HP suppression mechanism will greatly enhance the scientific rigor and accuracy of our manuscript.

Comment 3: In BiVO₄, small electron polarons typically dominate bulk transport limitations, whereas hole polarons localize strongly at Bi–O motifs and are known to be less mobile. This fundamental understanding is well documented. The manuscript’s central narrative that HPs are the primary limitation and require selective suppression does not convincingly introduce new physics. The responses emphasize literature consistency rather than presenting new experimental breakthroughs.

Response: We deeply appreciate your careful reading and the valuable comments that have helped us improve the clarity and scientific impact of our manuscript. We fully agree with your observation that in BiVO₄, EPs dominate the bulk charge transport, while HPs are localized within the Bi-O framework and exhibit poor mobility, a fundamental physical phenomenon that has been extensively studied and widely accepted in the literature. However, the innovation of our work does not lie in rediscovering the formation mechanism of polarons, but rather in proposing a surface-specific modulation strategy that enables selective modulation of HP sites, thereby effectively suppressing HP formation, enhancing the release of hole carriers, and reducing surface carrier recombination, which ultimately improves the photoelectrochemical (PEC) performance of the photoanode. Specifically, we employed DFT calculations to reveal that In³⁺ substitution for Bi³⁺ can suppress the formation of HPs in BiVO₄. We achieved selective substitution at the surface through

the LPCE method, and confirmed this via characterization techniques such as Spherical aberration corrected high-angle annular dark-field scanning transmission electron microscopy (AC HAADF-STEM) and XAFS. Subsequently, we used EPR, TD-PL, ISI-XPS, and fs-TAS to provide further evidence for the suppression of HP formation. Finally, we demonstrated the efficient release of hole carriers and the inhibition of surface recombination through SPV, TRPL, and a series of electrochemical measurements.

Our proposed strategy differs from previous approaches that focus on enhancing the hopping of electron polarons to improve bulk carrier transport. Instead, we target the surface HP sites to promote the release of surface hole carriers. This strategy holds significant practical importance for suppressing surface carrier recombination and enhancing the PEC performance of BiVO₄, and it provides a new perspective for optimizing the performance of photoanode materials. In accordance with your suggestions, we have revised the manuscript to clarify the methodological innovation and practical significance of this work. We hope that the revised version more accurately reflects the scientific contribution of our study. Below, we summarize the importance of our modulation strategy in PEC systems, as well as its distinctions and connections with previously reported studies.

1. The importance of surface HP site modulation in PEC systems

First, to elucidate the roles of HPs and EPs in the PEC system, it is necessary to briefly describe the operating principles and the challenges faced by PEC systems and photoanodes. As shown in Fig. R13, a typical PEC system consists of a photoanode (n-type semiconductor), a photocathode (p-type semiconductor), and an external circuit connecting them. The core processes include light absorption (carrier generation, I), charge separation (carrier transport, II), and charge transfer at the electrode/electrolyte interface (carrier injection, III). Specifically, the photoelectrode absorbs light, exciting electrons from the VB to CB and generating corresponding holes in VB. The excited electrons are then transported through the external circuit to the photocathode to participate in the reduction reaction, while the holes remain at the photoanode surface

to participate in the oxidation reaction. During the generation, separation, and injection of these charge carriers, carrier recombination can occur (IV), which negatively impacts the overall utilization efficiency. Notably, in water splitting systems, the photoanode undergoes the more complex four-electron oxygen evolution reaction (OER), which is widely considered to be the key limiting factor for the overall water-splitting performance. Therefore, it is essential to tailor the photoanode to improve its photoelectrochemical conversion efficiency.

Fig. R13 Schematic of the working process of PEC System. In the schematic, I represents the light absorption process, II represents the charge carrier separation process, III represents the charge carrier injection process, and IV represents the charge carrier recombination process.

For the photoanode, the processes of carrier generation, separation, and injection are each constrained by different factors (**Fig. R14**). The generation of carriers is limited by the bandgap of the photoanode, which determines the range of light absorption. The separation of carriers is hindered by severe bulk recombination, primarily due to the slow charge transport within the material. Recent studies have indicated that in metal oxides such as BiVO₄, the bulk transport process is mainly mediated by EP hopping, and its efficiency is limited by the relatively high activation energy associated with this process. As for the surface injection process, the main limiting factor is surface recombination. It is worth noting that, under the influence of an internal electric field, the photogenerated electron-hole pairs in the photoanode are

effectively separated. Consequently, in the actual operation of a PEC system, a large number of hole carriers accumulate at the photoanode surface rather than electrons. Unlike the bulk EP, which limits carrier transport efficiency, the HP at the photoanode surface causes the holes to become highly localized, resulting in significant surface recombination and severely restricting the injection of holes into the electrolyte. Therefore, we have designed a strategy targeting the key factor that limits surface carrier injection-HPs-by constructing surface InO_8 sites to suppress the formation of HPs and thereby release more hole carriers for participation in surface reactions. This approach differs from those that aim to enhance bulk charge transport efficiency by modulating EP behavior.

Fig. R14 Schematic of the working process of Photoanode.

2. The distinctions and connections between our proposed strategy and previously reported studies

In recent years, the influence of polarons on PEC systems has attracted widespread attention. Numerous studies have been reported on the regulation of EP sites in BiVO_4 photoanodes. These approaches have promoted EP hopping through doping and defect engineering, significantly improving the bulk charge transport efficiency and reducing bulk recombination. For example, Wu et al. enhanced EP hopping by hydrogenation treatment, which improved electron transport dynamics and enabled efficient migration of electrons to the conductive substrate⁴⁰. In their another study, the doping of P at V

sites simultaneously modulated the carrier density, trap states, and EP hopping behavior, achieving a remarkably high charge separation efficiency under low potential³⁶. Qiu et al. fabricated BiVO₄ photoanodes using an oxygen-rich precursor, effectively reducing the overall concentration of O vacancies and promoting EP transport, thereby achieving high electron transport efficiency⁴¹. Many other relevant studies have also been reported, such as the Ti-doped BiVO₄ by Wang et al., the Mo-doped BiVO₄ by Zhang et al., and the Fe:VOPO₄ grafted on Mo:BiVO₄ by He et al.^{34, 35, 42}. These works highlight the role of EP in bulk charge transport, demonstrating that promoting EP hopping can enhance the efficiency of bulk charge transport. The relevant content has been presented in Responses to Reviewer #4-Comment 2-Response (Fig. R10) for your kind reference.

However, it is worth noting that while bulk charge transport has a significant impact on PEC performance, the regulation of surface charge injection efficiency is equally critical. As mentioned earlier, a large number of holes accumulate at the surface of the photoanode to participate in surface oxidation reactions. The presence of HP at the surface can hinder the release of these holes, which has a substantial negative effect on the photoanode. Therefore, it is necessary to regulate HP sites in order to release more holes. In this regard, Lu et al. improved the charge transport properties at HP sites by creating Bi vacancies, thereby promoting the charge transfer efficiency⁴³. Unfortunately, the regulation of surface HP sites still remains to be further explored. In our work, the proposed surface-selective substitution strategy for suppressing HP formation not only effectively enhances the PEC performance of the photoanode but also provides a new perspective for the modulation of surface HP sites.

In summary, based on the distinct mechanisms of EP and HP in photoanodes, we designed and implemented a strategy for selective modulation of surface HP. Various characterization techniques (EPR, Td-PI, ISI-XPS, fs-TAS) confirmed the suppression of HP formation. Finally, SPV and other characterizations revealed the enhanced release of surface holes. Unlike strategies aimed at promoting EP hopping in the bulk phase, our approach is a surface-selective modification strategy targeting the slow release and

severe surface recombination of hole carriers caused by surface HP. This strategy offers a new direction for the more efficient utilization of surface charge carriers in PEC systems.

Comment 4: Although the authors provide extensive DFT evidence that In substitution increases HP formation energy, the manuscript still lacks direct observation of hole release from HP sites. As a result, the proposed “HP suppression” as the primary enhancement mechanism remains speculative.

Response: We sincerely appreciate your thoughtful and constructive comments on our work. We fully understand the concern regarding the proposed “HP suppression” as the mechanism for performance enhancement, which indeed requires further substantiation. As the reviewer rightly pointed out, the direct observation of hole release from polaronic sites is essential for clarifying the impact of HP suppression on the carrier dynamics. It is important to note that a range of experimental techniques is currently available for in-depth investigation of carrier characteristics, including the intuitive analysis of hole release and carrier recombination behaviors. In the manuscript, we have already employed SPV, TRPL, and a range of electrochemical tests to analyze the hole release process, providing a relatively comprehensive demonstration of how the suppression of hole polarons influences surface hole mobility. In response to your suggestion, we have further supplemented the data with the ratio of transient to steady-state photocurrent (i_0/i) derived from chopped-light current-time curves, as well as open-circuit potential (OCP) measurements, to further comprehensively reveal the changes in carrier characteristics within the photoanode. Moreover, we have additionally provided direct evidence demonstrating the suppression of HP formation, further supporting our view on the efficient release of holes at the HP sites. The relevant content has been presented in ***Responses to Reviewer #4-Comment 1-Response*** for your kind reference. Below, we will integrate these newly added characterizations with the existing data in the manuscript to provide a thorough analysis of the impact of the HP suppression strategy on hole release in the photoanode.

First, the SPV spectra provide a direct experimental demonstration of how the suppression of HP formation affects carrier release. The magnitude of surface photovoltage is not only determined by the number of electron-hole pairs generated from photon absorption, but also by the diffusion of these carriers to surface states^{44, 45}. For a photoanode, a stronger positive SPV signal indicates a greater number of holes diffusing to the surface and generating a larger contact potential difference^{46, 47}. In our SPV results, the In/BiVO₄ photoanode shows a significantly enhanced SPV signal compared to BiVO₄ photoanode, indicating a substantial increase in surface positive charge (**Fig. R15a**). This notable enhancement in surface photovoltage suggests an increased accumulation of hole carriers on the surface, which is attributed to the suppression of HP formation and the consequent facilitation of hole release. The SPV results thus provide a direct experimental confirmation that the suppression of HP formation promotes the release of hole carriers. The TRPL measurement reveals the impact of HP suppression on carrier lifetime (**Fig. R15b**)⁴⁸. Our results show that the carrier lifetime of the In/BiVO₄ photoanode (2.33 ns) is significantly longer than that of BiVO₄ photoanode (1.81 ns). This further confirms that the suppression of HP formation effectively facilitates the release of holes, thereby inhibiting surface recombination and extending the carrier lifetime.

In terms of electrochemical testing, we analyzed the changes in carrier characteristics using carrier injection efficiency (η_{inj}), chopped-light I-t curve as well as the open-circuit potential (OCP) test, which were supplemented in response to your suggestion. η_{inj} is derived from LSV curves measured with and without a sacrificial agent and reflects the percentage of carriers injected into the electrolyte. For a photoanode, this metric directly indicates the efficiency of surface hole release^{49, 50}. The result in **Fig. R15c** shows that η_{inj} of In/BiVO₄ photoanode reaches 73%, which is 2.1 times higher than that of BiVO₄ (35%). This result demonstrates that the surface hole carriers are released more efficiently in the modified photoanode. The transient photocurrent decay time (τ_D) and the ratio of transient to steady-state photocurrent (i_0/i), derived from the I-t curves under chopped light, reflect the separation and transport

capabilities of the carriers and help analyze recombination phenomena. The τ_D of the original BiVO₄ is significantly short (1.3 s), indicating a high degree of carrier recombination (**Fig. R15d**). In contrast, the τ_D of In/BiVO₄ increases to 2.5 s, nearly doubling, which suggests that the suppression of HP formation improves carrier transport and effectively reduces surface recombination. The i_0/i ratio is a measure of the release and transport of hole carriers (**Fig. R15e**). The i_0/i value of In/BiVO₄ is 0.67, a clear improvement over the original BiVO₄ (0.48), further indicating that surface holes are released more efficiently and surface recombination is significantly suppressed. OCP under illuminated and dark conditions is used to evaluate the driving force of the photoanode for water oxidation, reflecting the ability of holes to reach the surface and participate in the reaction. Compared to BiVO₄ (0.19 V), the OCP of In/BiVO₄ increases to 0.22 V, indicating an improvement in the hole transport and injection capability of the photoanode (**Fig. R15f**). This suggests that the surface hole carriers are released more effectively and surface recombination is significantly reduced.

Finally, the dark LSV curves, Tafel slope, and double-layer capacitance results all indicate that the surface substitution with In does not significantly enhance the OER activity (**Fig. R15g-i**). This excludes the possibility that the improvement in PEC performance is due to enhanced surface catalytic activity. Combined with the results of Td-EIS, we can further exclude the influence of the LPCE strategy on the EP sites (**Fig. R15j-l**). Taken together, our series of characterizations and tests comprehensively demonstrate that the enhancement in PEC performance is attributed to the suppression of HP formation, which in turn improves multiple aspects of carrier behavior.

Fig. R15 **a** SPV spectroscopy, **b** TRPL spectroscopy, **c** Carrier injection efficiency, **d** Chopped chronoamperometry curves and normalized transient current-time plots, **e** Ratio of transient to steady-state photocurrent, **f** OCP, **g** LSV curves under dark conditions, **h** Tafel slope curves, and **i** Double layer capacitance of BiVO₄ and In/BiVO₄ photoanodes. Td-EIS of **j** BiVO₄ and **k** In/BiVO₄ photoanodes. **l** The $\ln(\sigma T)$ - $1/T$ plots of BiVO₄ and In/BiVO₄ photoanodes. (Except for the data in Figures e and f, which are newly supplemented, all other data can be found in the manuscript and the supporting information)

Changes to the revised manuscript are shown below.

Main Manuscript (Abstract):

Page 17-18: Meanwhile, the increase in the ratio of steady-state to transient photocurrent density (from 0.48 to 0.67) after In substitution further indicates the suppression of surface recombination and the efficient release of surface holes (Supplementary Fig. 32a). In addition, the higher open-circuit potential (OCP) of the In/BiVO₄ photoanode also demonstrates an enhanced hole transport and injection capability (Supplementary Fig. 32b).

Supplementary Information:

Supplementary Fig. 32. Photoelectrochemical testing of photoanodes.

a The Ratio of transient to steady-state photocurrent and **b** OCP of BiVO₄ and In/BiVO₄ photoanodes.

Page 35: The i_0/i value of In/BiVO₄ is 0.67, a clear improvement over the original BiVO₄ (0.48), indicating that surface holes are released more efficiently and surface recombination is significantly suppressed. Compared to BiVO₄ (0.19 V), the OCP of In/BiVO₄ increases to 0.22 V, indicating an improvement in the hole transport and injection capability of the photoanode. This suggests that the surface hole carriers are released more effectively and surface recombination is significantly reduced.

Comment 5: While the topic is interesting and relevant to the BiVO₄ and polaron communities, the mechanistic claims are still not supported by sufficiently strong and direct experimental evidence. The revised manuscript appears to expand theoretical detail rather than resolve the scientific gaps.

Response: We sincerely appreciate your thoughtful and constructive feedback, which has helped us improve the scientific rigor of our work. In response to the concern regarding the experimental support for the proposed mechanism, we have conducted and included new and direct experimental evidence in the revised manuscript to better substantiate our mechanistic claims. Specifically, we have added the following key experimental results:

1. Direct evidence for the suppression of HP formation

We utilized ISI-XPS to investigate the changes in the strength of electron-phonon coupling at different polaron sites, allowing us to qualitatively analyze the localization behavior of EPs and HPs. Combined with the quantitative results from Td-PL, we were able to directly reveal the changes in the formation of HPs and EPs. The results showed that the formation of EPs is not significantly affected, while the formation of HPs is clearly suppressed. This directly supports our proposed concept of selective suppression of HP formation. To gain a deeper understanding of the polaron dynamics and formation times, we introduced fs-TAS to probe the ultrafast carrier dynamics on the surface. The results indicated that, after In doping, the decay lifetime τ_1 corresponding to the HP trapping process is significantly increased (from 0.59 ps to 1.09 ps), which directly demonstrates that the HP formation process is suppressed.

These experimental results have been presented in *Responses to Reviewer #4-Comment 1-Response* for your kind reference. Although the direct experimental detection of HPs is still technically challenging, the set of complementary experimental techniques employed in our manuscript (including EPR, Td-PL, ISI-XPS, and fs-TAS) all consistently confirm the suppression of HP formation (**Fig. R1-R9**).

2. Further evidence for the rapid release of surface holes

To further confirm the enhanced surface hole injection and utilization, we added two additional electrochemical tests in the revised manuscript: i_0/i and OCP. The results of these two experiments indicate that surface recombination is significantly reduced on In/BiVO₄, and surface holes can be released more efficiently. Moreover, the ultrafast

carrier dynamics analysis from fs-TAS also indicates that In/BiVO₄ exhibits a longer carrier lifetime and suppressed carrier recombination. These two experimental results have been presented in Responses to Reviewer #4-Comment 4-Response for your kind reference.

By combining the two additional electrochemical tests with the existing data in the manuscript (SPV, TRPL, and carrier injection efficiency), we conducted a comprehensive analysis of the impact of HP suppression on hole carrier release, and demonstrated the rapid release of surface holes and the suppression of surface carrier recombination (Fig. R15).

In summary, we would like to emphasize that the two aspects of direct experimental evidence included in the revised manuscript substantially reinforce the mechanistic basis of our study. We hope that the updated version adequately addresses your concern regarding the absence of such evidence. We sincerely appreciate your insightful and valuable suggestions, which have greatly contributed to the improvement of our manuscript.

References

1. Citrin PH, Eisenberger P, Hamann DR. Phonon broadening of X-ray photoemission linewidths. *Physical Review Letters* **33**, 965-969 (1974).
2. Ravensbergen J, *et al.* Unraveling the carrier dynamics of BiVO₄: a semtosecond to microsecond transient absorption study. *The Journal of Physical Chemistry C* **118**, 27793-27800 (2014).
3. Cerrato E, Paganini MC, Giamello E. Photoactivity under visible light of defective ZnO investigated by EPR spectroscopy and photoluminescence. *Journal of Photochemistry and Photobiology A: Chemistry* **397**, 112531 (2020).
4. Yang S, *et al.* Intrinsic small polarons in rutile TiO₂. *Physical Review B* **87**, 125201 (2013).
5. Li H, *et al.* Hole polaron-mediated suppression of electron–hole recombination triggers efficient photocatalytic nitrogen fixation. *Advanced Materials* **36**, 2408778 (2024).
6. Laguta V, *et al.* Light-induced defects in KTaO₃. *Journal of Applied Physics* **93**, 6056-6064 (2003).

7. Dimitrijevic NM, *et al.* Effect of size and shape of nanocrystalline TiO₂ on photogenerated charges. An EPR study. *The Journal of Physical Chemistry C* **111**, 14597-14601 (2007).
8. Di Valentin C, *et al.* Density functional theory and electron paramagnetic resonance study on the effect of N-F codoping of TiO₂. *Chemistry of Materials* **20**, 3706-3714 (2008).
9. Niklas J, Poluektov OG. Charge transfer processes in OPV materials as revealed by EPR spectroscopy. *Advanced Energy Materials* **7**, 1602226 (2017).
10. Franchini C, *et al.* Polarons in materials. *Nature Reviews Materials* **6**, 560-586 (2021).
11. Zhang Y. Applications of Huang–Rhys theory in semiconductor optical spectroscopy. *Journal of Semiconductors* **40**, 091102 (2019).
12. Moreno M, M. T. Barriuso, J. A. Aramburu. The Huang–Rhys factor $S(a_{1g})$ for transition-metal impurities: a microscopic insight. *Journal of Physics: Condensed Matter* **4**, 9481-9488 (1992).
13. Luo J, *et al.* Efficient and stable emission of warm-white light from lead-free halide double perovskites. *Nature* **563**, 541-545 (2018).
14. McCall KM, *et al.* Strong electron–phonon coupling and self-trapped excitons in the defect halide perovskites A₃M₂I₉ (A= Cs, Rb; M= Bi, Sb). *Chemistry of Materials* **29**, 4129-4145 (2017).
15. Liu W, *et al.* Temperature-dependent photoluminescence of ZnCuInS/ZnSe/ZnS quantum dots. *The Journal of Physical Chemistry C* **117**, 19288-19294 (2013).
16. Lin Z, Zheng W, Huang F. Narrow band emission from layered α -HgI₂ micro-/nano-sheets with high Huang-Rhys factor. *Journal of Luminescence* **237**, 118161 (2021).
17. Preclíková J, *et al.* Nanocrystalline titanium dioxide films: Influence of ambient conditions on surface- and volume-related photoluminescence. *Journal of Applied Physics* **108**, 113502 (2010).
18. Liu T, Cui M, Dupuis M. Hole polaron transport in bismuth vanadate BiVO₄ from hybrid density functional theory. *The Journal of Physical Chemistry C* **124**, 23038-23044 (2020).
19. Wiktor J, Ambrosio F, Pasquarello A. Role of polarons in water splitting: The case of BiVO₄. *ACS Energy Letters* **3**, 1693-1697 (2018).
20. Maslakov KI, *et al.* XPS study of ion irradiated and unirradiated CeO₂ bulk and thin film samples. *Applied Surface Science* **448**, 154-162 (2018).

21. Li Y, *et al.* Revealing electron numbers-binding energy relationships in heterojunctions via in-situ irradiated XPS. *Applied Catalysis B: Environment and Energy* **356**, 124223 (2024).
22. Li Y, *et al.* Molecular-polaron-coupling-enhanced photocatalytic CO₂ reduction on copper phthalocyanine/NiMgFe layered double hydroxide nanocomposites. *Journal of Materials Chemistry A* **12**, 13168-13180 (2024).
23. Chaudhuri A, *et al.* Direct observation of anisotropic small-hole polarons in an orthorhombic structure of BiVO₄ films. *Physical Review B* **97**, 195150 (2018).
24. Wang J, *et al.* Effect of cation arrangement on polaron formation and colossal permittivity in NiNb₂O₆. *Journal of Materials Chemistry C* **8**, 16107-16112 (2020).
25. Vdović S, *et al.* Excited state dynamics of β-carotene studied by means of transient absorption spectroscopy and multivariate curve resolution alternating least-squares analysis. *Physical Chemistry Chemical Physics* **15**, 20026 (2013).
26. Zhang C, *et al.* Tailoring non-covalent interaction via single atom to boost interfacial charge transfer toward photoelectrochemical water oxidation. *Advanced Materials* **37**, 2410632 (2025).
27. Grigioni I, *et al.* Dynamics of photogenerated charge carriers in WO₃/BiVO₄ heterojunction photoanodes. *The Journal of Physical Chemistry C* **119**, 20792-20800 (2015).
28. Zhu K, *et al.* Unraveling the mechanisms of beneficial Cu-doping of NiO-based photocathodes. *The Journal of Physical Chemistry C* **125**, 16049-16058 (2021).
29. Jing J, *et al.* Construction of interfacial electric field via dual-porphyrin heterostructure boosting photocatalytic hydrogen evolution. *Advanced Materials* **34**, 2106807 (2021).
30. Wang J, *et al.* Photothermal CuS as a hole transfer layer on BiVO₄ photoanode for efficient solar water oxidation. *Angewandte Chemie International Edition* **64**, 202507259 (2025).
31. Xin Y, *et al.* Enhanced photocatalytic efficiency through oxygen vacancy-driven molecular epitaxial growth of metal-organic frameworks on BiVO₄. *Advanced Materials* **37**, 2417589 (2025).
32. Chen C, *et al.* Efficient photoreduction of CO₂ to CO with 100% selectivity by slowing down electron transport. *Journal of the American Chemical Society* **146**, 9163-9171 (2024).
33. Lin Y, *et al.* Organic gradient homojunction via D-A engineering enables photoelectric/photothermal dual-assisted catalysis toward full spectrum light-coupled low-temperature seawater batteries. *Advanced Materials* **37**, 2415608 (2025).
34. He B, *et al.* Enhanced bulk and interfacial charge transfer in Fe:VOPO₄ modified Mo:BiVO₄

- photoanodes for photoelectrochemical water splitting. *EScience* **5**, 100242 (2025).
35. Zhang WR, *et al.* Unconventional relation between charge transport and photocurrent via boosting small polaron hopping for photoelectrochemical water splitting. *ACS Energy Letters* **3**, 2232-2239 (2018).
 36. Wu H, *et al.* Low-bias photoelectrochemical water splitting via mediating trap states and small polaron hopping. *Nature Communications* **13**, 6231 (2022).
 37. Tang S, *et al.* Harvesting of infrared part of sunlight to enhance polaron transport and solar water splitting. *Advanced Functional Materials* **32**, 2110284 (2022).
 38. Rettie AJ, *et al.* Unravelling small-polaron transport in metal oxide photoelectrodes. *The Journal of Physical Chemistry Letters* **7**, 471-419 (2016).
 39. Rettie AJE, *et al.* Anisotropic small-polaron hopping in W:BiVO₄ single crystals. *Applied Physics Letters* **106**, 022106 (2015).
 40. Wu H, *et al.* Polaron-mediated transport in BiVO₄ photoanodes for solar water oxidation. *ACS Energy Letters* **8**, 2177-2184 (2023).
 41. Qiu W, *et al.* Freeing the polarons to facilitate charge transport in BiVO₄ from oxygen vacancies with an oxidative 2D precursor. *Angewandte Chemie International Edition* **58**, 19087-19095 (2019).
 42. Wang J, *et al.* Unconventional substitution for BiVO₄ to enhance photoelectrocatalytic performance by accelerating polaron hopping. *ACS Applied Materials & Interfaces* **15**, 14359-14368 (2023).
 43. Lu Y, *et al.* Boosting charge transport in BiVO₄ photoanode for solar water oxidation. *Advanced Materials* **34**, 2108178 (2022).
 44. Zhang L, *et al.* N-TiO₂-coated polyester filters for visible light—photocatalytic removal of gaseous toluene under static and dynamic flow conditions. *Journal of Environmental Chemical Engineering* **4**, 357-364 (2016).
 45. Fang Y, *et al.* High-efficiency oxygen evolution on γ -Fe₂O₃ catalysts with BiVO₄ photoabsorbers and TpAQ hole transport layers for photoelectrochemical water splitting. *Journal of Colloid and Interface Science* **689**, 137213 (2025).
 46. Xie M, *et al.* Long-lived, visible-light-excited charge carriers of TiO₂/BiVO₄ nanocomposites and their unexpected photoactivity for water splitting. *Advanced Energy Materials* **4**, 1300995 (2014).

47. Zhang J, *et al.* Novel AuPd bimetallic alloy decorated 2D BiVO₄ nanosheets with enhanced photocatalytic performance under visible light irradiation. *Applied Catalysis B: Environmental* **204**, 385-393 (2017).
48. Miao Y, *et al.* Surface active oxygen engineering of photoanodes to boost photoelectrochemical water and alcohol oxidation coupled with hydrogen production. *Applied Catalysis B: Environmental* **323**, 122147 (2023).
49. Luo W, *et al.* Phthalocyanine grafting strategy induces strong intrinsic electric fields and molecule-edge carrier transport pathways for photoelectrochemical water splitting. *Angewandte Chemie International Edition* **64**, 202504589 (2025).
50. Tian K, *et al.* Interfacial engineering-assisted energy level modulation enhances the photoelectrochemical water oxidation performance of bismuth vanadate photoanodes. *Advanced Energy Materials* **15**, 2404477 (2024).